

# On scalar products in higher rank quantum separation of variables

**Jean Michel Maillet⋆, Giuliano Niccoli† and Louis Vignoli‡**

Univ Lyon, Ens de Lyon, Univ Claude Bernard, CNRS,
Laboratoire de Physique, UMR 5672, F-69342 Lyon, France

⋆ maillet@ens-lyon.fr † giuliano.niccoli@ens-lyon.fr ‡ louis.vignoli@ens-lyon.fr

## Abstract

Using the framework of the quantum separation of variables (SoV) for higher rank quantum integrable lattice models [1], we introduce some foundations to go beyond the obtained complete transfer matrix spectrum description, and open the way to the computation of matrix elements of local operators. This first amounts to obtain simple expressions for scalar products of the so-called separate states, that are transfer matrix eigenstates or some simple generalization of them. In the higher rank case, left and right SoV bases are expected to be *pseudo-orthogonal*, that is for a given SoV co-vector $\langle \underline{h} |$, there could be more than one non-vanishing overlap $\langle \underline{h} | \underline{k} \rangle$ with the vectors $| \underline{k} \rangle$ of the chosen right SoV basis. For simplicity, we describe our method to get these pseudo-orthogonality overlaps in the fundamental representations of the $\mathcal{Y}(gl_3)$ lattice model with $N$ sites, a case of rank 2. The non-zero couplings between the co-vector and vector SoV bases are exactly characterized. While the corresponding *SoV-measure* stays reasonably simple and of possible practical use, we address the problem of constructing left and right SoV bases which do satisfy standard orthogonality (by standard we mean $\langle \underline{h} | \underline{k} \rangle \propto \delta_{\underline{h},\underline{k}}$). In our approach, the SoV bases are constructed by using families of conserved charges. This gives us a large freedom in the SoV bases construction, and allows us to look for the choice of a family of conserved charges which leads to orthogonal co-vector/vector SoV bases. We first define such a choice in the case of twist matrices having simple spectrum and zero determinant. Then, we generalize the associated family of conserved charges and orthogonal SoV bases to generic simple spectrum and invertible twist matrices. Under this choice of conserved charges, and of the associated orthogonal SoV bases, the scalar products of separate states simplify considerably and take a form similar to the $\mathcal{Y}(gl_2)$ rank one case.

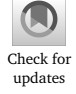

# 1  Introduction

The quantum separation of variables (SoV) has been introduced by Sklyanin [2–6] in the framework of the quantum inverse scattering method [7–15]. It enables to analyze the transfer matrix (and Hamiltonian) spectrum using the Yang-Baxter commutation relations. It does not rely on any ansatz, which makes explicit its advantage w.r.t. Bethe Ansatz methods [8,16–22]. This method has been first systematically developed in the class of the rank one integrable quantum models [23–54] proving its wide range of application. The completeness of the transfer matrix spectrum characterization in the SoV approach for compact representations has been clearly addressed and proven in [33–35,37–55]. In this rank one case, the SoV approach has also been shown to lead to simple determinant formulae for scalar products of the so-called separate states [37,39–45,48,53,54]. Those include the transfer matrix eigenstates and their generalizations with factorized but otherwise arbitrary wave functions in the SoV basis. In several important cases, the form factors of local or quasi-local operators have been computed in terms of determinants, while in [50,52,56] a rewriting of the determinants giving the scalar product formulae has been obtained paving the way for the direct analysis of form factors and correlation functions in the homogeneous and thermodynamic limits.

    Our aim is to extend these achievements to the higher rank cases. Let us comment that scalar product formulae and matrix elements of local operators have been already computed

in the literature [57–70] for the higher rank case in the nested algebraic Bethe ansatz (NABA) framework [71–74] and that more recently have appeared interesting works analyzing these problems in SoV related frameworks [75, 76].

Sklyanin has also pioneered the SoV approach in the higher rank case[1], in the particular example of rank two [6]. Sklyanin's beautiful SoV construction involves the identification of a $B$-operator, whose eigenco-vector basis is meant to separate the spectral problem of the transfer matrix. The other fundamental elements of the Sklyanin's construction [6] are the identification of an $A$-operator, whose role is that of generating the shift operator on the $B$-spectrum, together with the identification of an operator quantum spectral curve equation involving the transfer matrices, the $B$-operator and the $A$-operator. These operator equations should separate the transfer matrix spectrum when computed in the zeroes of the $B$-operator. However, in [6] the SoV construction has been developed just using the $gl_3$ Yang-Baxter commutation relations without introducing any specific representations of the algebra. Only more recently, the SoV analysis for higher rank has been revived. For the fundamental representations of $gl_3$ Yang-Baxter algebra, in [77] the spectrum of the Sklyanin's $B$-operator has been conjectured together with its diagonalizability for some classes of twisted boundary conditions on the basis of an exact analysis of quantum chains of small sizes. Moreover in [77], the Sklyanin's $B$-operator has been used to conjecture a formula for the transfer matrix eigenvectors bypassing the traditional nested Bethe Ansatz procedure and consistent with small chains verification[2]. Then, in [79] the separation of variables approach has been initiated for non-compact representations of the $gl_3$ Yang-Baxter algebra determining the eigenfunctions of the Sklyanin's $B$-operator. While these findings are quite interesting, the complete implementation of the Sklyanin's SoV program for higher rank seems more involved as, at least for fundamental representations, the proposed $A$-operator acts as shift only on part of the $B$-spectrum which leaves unproven the separate relations in this SoV framework. This phenomenon has been already anticipated by Sklyanin in [6] and it occurs when the spectrum of the $B$-operator zeroes partially coincides with that of the poles of operators appearing in the commutation relations between $A$-operator and $B$-operator and/or in the operator quantum spectral curve equation, see [1] for further discussions.

In [1] we have overcome these difficulties by developing a new SoV approach which relies only on the abelian algebra of conserved charges of the given quantum integrable model. In our SoV approach the SoV co-vectors/vectors bases are generated by the action of appropriate sets of conserved charges on some reference co-vector/vector, hence bypassing the construction of the Sklyanin's $A$ and $B$ operators.

In its most general form, our construction uses a family of commuting conserved charges say $T(\lambda)$, $\lambda \in \mathbb{C}$ (typically the transfer matrix, its fused versions or the Baxter $Q$-operator in most of the cases considered, but in principle more general situations could occur) acting on some Hilbert space $\mathcal{H}$ ($\mathcal{H}^*$ being its dual) of the considered model. Such a family is said to be SoV bases generating if there exist a co-vector $\langle L| \in \mathcal{H}^*$ (resp. a vector $|R\rangle \in \mathcal{H}$) and sets of commuting conserved charges constructed from $T(\lambda)$, $T_{h_a}^{(a)}$ (resp. $\tilde{T}_{k_a}^{(a)}$) where $a = 1, \ldots, N$ and $h_a, k_a = 0, \ldots, d_a - 1$ with $d = \prod_{a=1}^{N} d_a$ the dimension of the Hilbert spaces $\mathcal{H}$ and $\mathcal{H}^*$, such that the set of co-vectors,

$$\langle h_1, \ldots, h_N| = \langle L| \prod_{a=1}^{N} T_{h_a}^{(a)} \, , \qquad (1.1)$$

---

[1]See also [25, 51] for some interesting analysis toward the SoV description of higher rank cases.

[2]This conjecture has been then proven in the ABA framework in [70]. These observations and conjectures have also been extended to the super-symmetric case in [78].

forms a basis of $\mathcal{H}^*$ and the set of vectors,

$$|k_1,\dots,k_N\rangle = \prod_{a=1}^{N} \tilde{T}_{h_a}^{(a)}|R\rangle \,, \tag{1.2}$$

forms a basis of $\mathcal{H}$. It follows immediately, by construction, that whenever such bases exist, any common eigenvector $|t\rangle$ (resp. eigenco-vector $\langle t|$) of the family $T(\lambda)$ with eigenvalue $t(\lambda)$ is also a common eigenvector (resp. eigenco-vector) of the commuting sets of conserved charges $T_{h_a}^{(a)}$ (resp. $\tilde{T}_{k_a}^{(a)}$) with eigenvalues $t_{h_a}^{(a)}$ (resp. $\tilde{t}_{k_a}^{(a)}$). Hence the corresponding wave functions in the coordinates $h_i$ (resp. $k_i$) factorize as

$$\Psi_t(h_1,...,h_N) \equiv \langle h_1,...,h_N|t\rangle = \langle L|t\rangle \prod_{i=1}^{N} t_{h_i}^{(i)} \,, \tag{1.3}$$

and similarly,

$$\tilde{\Psi}_t(k_1,...,k_N) \equiv \langle t|k_1,...,k_N\rangle = \langle t|R\rangle \prod_{i=1}^{N} \tilde{t}_{k_i}^{(i)} \,. \tag{1.4}$$

This also means that the eigenvectors coordinates in such SoV bases are completely determined from the eigenvalues of the commuting conserved charges used to construct those bases. Hence, the very existence of such bases implies the simplicity of the spectrum of the family $T(\lambda)$ since the coordinates (wave function) of any eigenvector are completely determined by the corresponding eigenvalue. This in turn implies that the above sets of conserved charges $T_{h_a}^{(a)}$ and $\tilde{T}_{k_a}^{(a)}$ are both basis of the vector space $\mathcal{C}_{T(\lambda)}$ of operators commuting with the family of operators $T(\lambda)$. Hence the *linear* action of the operator $T(\lambda)$ on such bases can be computed in a close form as for any values of $h_1,...,h_N$ (resp. $k_1,...,k_N$), the product $T_{h_a}^{(a)} \cdot T(\lambda)$ (resp. $T(\lambda) \cdot \tilde{T}_{k_a}^{(a)}$) is also a conserved charge commuting with $T(\lambda)$. Hence it is an element of $\mathcal{C}_{T(\lambda)}$ that can be decomposed *linearly* on the basis generated by $T_{h_a}^{(a)}$ (resp. $\tilde{T}_{k_a}^{(a)}$).

To make this more explicitly, let us introduce compact notations we will be using all along this paper, namely, $\underline{h} = (h_1,...,h_N)$ and similarly $\underline{k} = (k_1,...,k_N)$, and accordingly, $T_{\underline{h}} = \prod_{a=1}^{N} T_{h_a}^{(a)}$, $\tilde{T}_{\underline{k}} = \prod_{a=1}^{N} \tilde{T}_{k_a}^{(a)}$, and also $|k_1,...,k_N\rangle = |\underline{k}\rangle$, $\langle h_1,...,h_N| = \langle \underline{h}|$ for the two sets defining the right and left SoV bases[3] , then there exist scalar complex coefficients $N_{\underline{h}}^{\underline{l}}(\lambda)$ and $N_{\underline{h},\underline{k}}^{\underline{l}}$ such that[4]:

$$T_{\underline{h}} \cdot T(\lambda) = \sum_{\underline{l}} N_{\underline{h}}^{\underline{l}}(\lambda)\, T_{\underline{l}} \,, \tag{1.5}$$

and,

$$T_{\underline{h}} \cdot \tilde{T}_{\underline{k}} = \sum_{\underline{l}} N_{\underline{h},\underline{k}}^{\underline{l}}\, T_{\underline{l}} \,. \tag{1.6}$$

---

[3]Using such compact notations it should not be forgotten that these vectors $|\underline{k}\rangle$ and co-vectors $\langle \underline{h}|$ defining SoV bases are depending respectively on the chosen sets of conserved charges $\tilde{T}_{k_a}^{(a)}$ and $T_{h_a}^{(a)}$ and on the reference vector $|R\rangle$ and co-vector $\langle L|$. Hence in the following such compact notations will be used only after such choices have been defined.

[4]Let us stress here that these complex coefficients which can be interpreted as the structure constants of the associative and commutative algebra of the conserved charges, are depending directly on the choice of the two sets of commuting conserved charges $T_{h_a}^{(a)}$ and $\tilde{T}_{k_a}^{(a)}$. Hence changing those sets, eventually in a non-linear way, as sums of products of commuting conserved charges are still commuting conserved charges, will modify these structure constants accordingly.

Similarly one can define two other sets of complex coefficients, namely $C_{\underline{\mathbf{h}},\underline{\mathbf{k}}}^{\underline{\mathbf{l}}}$ and $\tilde{C}_{\underline{\mathbf{h}},\underline{\mathbf{k}}}^{\underline{\mathbf{l}}}$ such that:

$$T_{\underline{\mathbf{h}}} \cdot T_{\underline{\mathbf{k}}} = \sum_{\underline{\mathbf{l}}} C_{\underline{\mathbf{h}},\underline{\mathbf{k}}}^{\underline{\mathbf{l}}} \, T_{\underline{\mathbf{l}}} \,, \tag{1.7}$$

and,

$$\tilde{T}_{\underline{\mathbf{h}}} \cdot \tilde{T}_{\underline{\mathbf{k}}} = \sum_{\underline{\mathbf{l}}} \tilde{C}_{\underline{\mathbf{h}},\underline{\mathbf{k}}}^{\underline{\mathbf{l}}} \, \tilde{T}_{\underline{\mathbf{l}}} \,. \tag{1.8}$$

The knowledge of these relations together with the action of the complete family of conserved charges $T(\lambda)$ on our SoV bases has been shown to completely characterize the common spectrum of all the above commuting conserved charges. Particular realizations of this situation include the case where the $T_{h_a}^{(a)}$ are powers of the transfer matrix evaluated in the inhomogeneity parameters as $T(\xi_a)^{h_a}$, or are given as the fused transfer matrices $T_{h_a}(\xi_a^{(h_a)})$ in some shifted points $\xi_a^{(h_a)}$, where $h_a$ is the level of fusion. In the higher spin $gl_2$ case, they are simply obtained from the $Q$-operator evaluated in shifted inhomogeneities as $Q(\xi_a^{(h_a)})$. In all these cases, the coefficients $N_{\underline{\mathbf{h}}}^{\underline{\mathbf{l}}}(\lambda)$, $N_{\underline{\mathbf{h}},\underline{\mathbf{k}}}^{\underline{\mathbf{l}}}$, $C_{\underline{\mathbf{h}},\underline{\mathbf{k}}}^{\underline{\mathbf{l}}}$ and $\tilde{C}_{\underline{\mathbf{h}},\underline{\mathbf{k}}}^{\underline{\mathbf{l}}}$ are completely determined by the fusion relations or the $T$-$Q$ relations satisfied by the transfer matrices and the Baxter $Q$-operator.

The conditions on the above sets of conserved charges to indeed generate SoV bases were identified and proven[5] in [1], together with the factorization of the wave functions in terms of conserved charge eigenvalues and the proof of the completeness of the description of the transfer matrix spectrum. The discrete separate relations were proven to be equivalent to the quantum spectral curve equations, involving the transfer matrices and the $Q$-operator holding both at the eigenvalue and operator level, due to the proven simplicity of the transfer matrix spectrum [1]. In our approach, the separate variables relations are themselves proven to be originated by the structure constants of the abelian algebra of conserved charges, in particular by the transfer matrix fusion equations for the charges considered in [1]. From this perspective our SoV approach has the potential to be universal in the realm of quantum integrable model. Indeed, we have proven its applicability for a large class of quantum integrable models from the fundamental representations of $gl_n$, $gl_{n,m}$ and the $U_q(gl_n)$ Yang-Baxter algebras with simple spectrum twist matrices up to the higher rank reflection algebra cases with general boundary conditions, deriving new and complete descriptions of the transfer matrix spectrum [1,80–84] [6]. Moreover, in [85, 86] our construction of SoV bases using conserved charges has been extended to arbitrary finite dimensional rectangular representations of the $gl_n$ Yang-Baxter algebra.

The relation of our SoV approach with the Sklyanin's one has been first analyzed in [1]. There we have observed the coincidence of our SoV co-vector basis with the Sklyanin's $B$-operator co-vector eigenbasis for chains of arbitrary length in the $gl_2$ case. This correspondence has been obtained for special choices of the reference co-vector and of the set of conserved charges used to generate the SoV basis. The same result has been derived in [1] for the $gl_3$ case for chains of small sizes. In [85] this observation has been proven for arbitrary finite dimensional rectangular representations of the $gl_n$ Yang-Baxter algebra and for chains of any size. Moreover the simple spectrum of the Sklyanin's $B$-operator, and its $gl_n$ extensions proposed in [77], has been obtained in [85]. This result together with the completeness of the description of the spectrum by factorized wave functions in terms of polynomial $Q$-functions [1] implies the ABA type formula of [77] for all the transfer matrix eigenvectors[7].

---

[5]They mainly reduce to properties satisfied by the twist matrix and the inhomogeneities parameters.

[6]Note that our reference [83] describe our approach for higher spin representations for the rank one case. While [84] also contains the SoV basis construction for the quasi-periodic Hubbard model.

[7]Note that it was first remarked in [26] for non-compact rank one models that the factorization of the wave-functions in terms of polynomial $Q$-functions imply the ABA form of transfer matrix eigenvectors in the SoV basis

An important feature of our new approach to the SoV bases is that it relies only on finding a suitable set of commuting conserved charges and a corresponding reference co-vector/vector $\langle L| \in \mathcal{H}^*$ and $|R\rangle \in \mathcal{H}$, (the number of choices for those being in fact very large as shown in our first paper [1]). However, any other sets build from sums of products of given commuting conserved charges being again sets of commuting conserved charges, it results in a huge freedom in constructing SoV bases which was not available if one would have stick to SoV bases identified as eigenbasis of the Sklyanin's $B$-operator or its higher rank extensions.

Clearly, this is a very interesting built in aspect of our new approach to SoV that enables us to ask a new key question in this context: *what would be optimal choices of the sets of conserved charges determining the SoV bases for the quantum integrable model at hand?*

A first answer to this question, from the point of view of the determination of the spectrum, is that an optimal SoV basis is such that the action of the transfer matrix (and hence of the Hamiltonian of the model) on the chosen basis is as simple as possible. This could mean for example that the action of the family of $T(\lambda)$ on any element of the set $T_{\underline{h}}$ decomposes back on that set with only a very few non-zero coefficients, and moreover that it is given only by local shifts of finite and lowest possible order on the coordinates $h_a$. This amounts to have chosen the basis $T_{\underline{h}}$ of the space $\mathcal{C}_{T(\lambda)}$ in such a way that the structure constants $N_{\underline{h}}^{\underline{l}}(\lambda)$ have such a simple property; namely that the only non zero coefficients are those where $\underline{h}$ and $\underline{l}$ differ only by localized shifts in the coordinates. This is exactly what happens for SoV bases in the $gl_2$ case that are generated directly from the Baxter $Q$-operator. Indeed, the Baxter $T$-$Q$ relation determines an action of the transfer matrix $T(\lambda)$ on the basis generated by $Q(\lambda)$ which involves only two terms with a local shift $\pm 1$ for each coordinate $h_a$, to be compared to the dimension of the Hilbert space $\mathcal{H}$ and of the Bethe algebra $\mathcal{C}_{T(\lambda)}$ which is $2^N$ for a spin-1/2 chain of length $N$. This is in some sense the hallmark of integrability that generate a characteristic equation of degree two, hence much smaller than the dimension of the Hilbert space.

Another meaning of simplicity in the choice of our SoV bases could also be related to the coupling between the two chosen left (1.1) and right (1.2) SoV bases. Namely, a criterion of simplicity could be to take such two SoV covector/vector bases such that their scalar products are calculable in terms of manageable expressions. This is certainly an important question and criterion as it determines to what extend the chosen left (1.1) and right (1.2) SoV bases are easy to use when computing scalar products of separate states, form factors and correlation functions, that are our main goals.

The main purpose of the present paper is to study the important question of scalar products from this perspective.

In the class of rank one quantum integrable models, the SoV analysis so far developed [2–6,23–54] leads to the expectation that the transfer matrix construction of the co-vector/vector SoV bases can be defined in such a way that these are orthogonal bases. Similarly, in the Sklyanin's approach, this leads to the expectation that the co-vector/vector Sklyanin's $B$-operator eigenbases (orthogonal as soon as $B$ is diagonalizable with simple spectrum) both implement the separation of variables for the transfer matrix spectrum. This feature has been proven to be very useful in computing scalar products of the so-called separate states and also in obtaining

---

once the Sklyanin's $B$-operator is proven to be diagonalizable. As we have explained in [1], this proof extends also to the higher rank case under the same assumption as it only uses the SoV representation of the transfer matrix eigenvectors.

determinant formulae for the form factors of local operators. As we will see in the next, in the higher rank quantum integrable models, this is not directly the case if the charges used to construct the co-vector/vector SoV basis are simply the transfer matrices or their fused higher versions, for a generic twist $K$.

On the one hand, the SoV vector basis is univocally fixed in terms of the co-vector one defined in [1] if one requires that it is of SoV type, i.e. that it is generated by a factorized action of conserved charges, and that it satisfies the orthogonality conditions with the co-vector basis on one quantum site (this is obviously a necessary requirement for general orthogonality!). It turns out that in general such SoV vector basis stays only *pseudo-orthogonal* to the co-vector one for quantum chains of arbitrary length $N$. More precisely, the matrix of scalar products $\mathcal{N}_{\underline{\mathbf{h}},\underline{\mathbf{k}}} = \langle \underline{\mathbf{h}} | \underline{\mathbf{k}} \rangle$ for the natural SoV bases introduced in [1] is in general not a diagonal matrix.

The aim of the present paper is twofold:

- Characterize the matrix of scalar products $\mathcal{N}_{\underline{\mathbf{h}},\underline{\mathbf{k}}} = \langle \underline{\mathbf{h}} | \underline{\mathbf{k}} \rangle$ and the associated SoV measure (related to the inverse of $\mathcal{N}_{\underline{\mathbf{h}},\underline{\mathbf{k}}}$) for the natural SoV bases introduced in [1] in the example of the rank two $gl_3$ case in the fundamental representations.

- Determine, in the same $gl_3$ representations, two sets of commuting conserved charges, $T_{\underline{\mathbf{h}}}$ and $\tilde{T}_{\underline{\mathbf{k}}}$ generating a left and right SoV bases that are orthogonal to each other and compute the corresponding SoV measure.

Given our left SoV co-vector basis, we first prove that the defined set of SoV vectors indeed define a basis and we exactly characterize the *pseudo-orthogonality* conditions writing all the non-zero non-diagonal couplings in terms of the diagonal ones, which we explicitly compute. This set of SoV vectors has been introduced recently in [76] as the set of eigenvectors of a $C$-operator which plays a similar role to the Sklyanin's $B$-operator and some integral form has been given for the coupling of the SoV co-vectors/vectors in [76]. Due to the quite different representations, a direct comparison of the results of [76] with those that we obtained stays a complicate task which however deserves further analysis.

Let us comment that this pseudo-orthogonality is intrinsically related to the form of fusion relations of the transfer matrices for higher rank case when computed in the special inhomogeneous points. In fact the matrix of scalar products can be directly related to the structure constants of the algebra of commuting conserved charges (1.6) that are in fact determined completely by the fusion relations as shown in [1]. To be more precise let us illustrate this in the following situation. Suppose we have chosen a left SoV basis of the type (1.1). Then let us consider a right SoV basis (1.2) where we have chosen the right reference vector $|R\rangle$ in such a way that it satisfies $\langle \underline{\mathbf{h}} | R \rangle = \delta_{\underline{\mathbf{h}},\underline{\mathbf{h}}_0}$ for some $\underline{\mathbf{h}}_0$. Then the corresponding matrix $\mathcal{N}_{\underline{\mathbf{h}},\underline{\mathbf{k}}}$ of scalar products can be computed in terms of the structure constant $N_{\underline{\mathbf{h}},\underline{\mathbf{k}}}^{\underline{\mathbf{h}}_0}$ to be:

$$\mathcal{N}_{\underline{\mathbf{h}},\underline{\mathbf{k}}} = N_{\underline{\mathbf{h}},\underline{\mathbf{k}}}^{\underline{\mathbf{h}}_0} . \tag{1.9}$$

A very interesting question is thus if there exists an optimal choice of the left (1.1) and right (1.2) SoV bases such that for some $\underline{\mathbf{h}}_0$ we have $N_{\underline{\mathbf{h}},\underline{\mathbf{k}}}^{\underline{\mathbf{h}}_0} = \delta_{\underline{\mathbf{h}},\underline{\mathbf{k}}} \, n(\underline{\mathbf{h}})$ with a calculable non-zero coefficient $n(\underline{\mathbf{h}})$ whose inverse determines the SoV measure.

This naturally leads to the observation that if we want to obtain co-vector/vector SoV bases mutually orthogonal we have to chose in general a different family of commuting conserved charges than the simple choice taken in [1] to generate both of them (or at least look for different points where the transfer matrices are computed). These observations in the Sklyanin's SoV framework for rank two mean that while the Sklyanin's $B$-operator define the co-vector

SoV basis, its vector eigenbasis is actually only a *pseudo-SoV* basis, i.e. not all the wave functions of transfer matrix eigenco-vectors have factorized form in terms of the transfer matrix eigenvalues.

Despite the absence of direct orthogonality the *SoV-measure* that we derive in section 3 stays reasonably simple and can be used as the starting point to compute matrix elements of local operators in this SoV framework. While, this seems a sensible line of research and we will further analyze it in the future, we would like to further investigate the potentiality of our new SoV approach.

In the present paper, for the rank two $gl_3$ case in the fundamental representation, we define some new family of commuting conserved charges whose spectral problem is separated for both a co-vector and a vector bases, which are moreover orthogonal to each other. Further, we show that the corresponding *SoV measure* takes a form very similar to the rank one case. The consequence is that w.r.t. this family of commuting conserved charges scalar products simplify considerably and take a form very similar to the rank one case for the separate states. Of course, in order to be able to compute matrix elements of local operators we will need to address the problem of the representation of the local operators in these new SoV bases.

The paper is organized as follows:

Section 2 is dedicated to recall some fundamental properties satisfied by the transfer matrices in the fundamental representations of the $gl_3$ Yang-Baxter algebra. In subsection 2.2, we moreover recall the results of [1] for the construction of the SoV bases for the considered representations, that is equations (2.23) and (2.24).

In section 3, we introduce a standard construction of co-vector/vector SoV bases (3.6)-(3.7) using the choice of the generating charges made in [1], i.e. given by the transfer matrices evaluated in the inhomogeneity parameters. The Theorem 3.1 characterizes completely the co-vector/vector coupling of these two systems of SoV states. The main results of this section are *i)* that the given system of SoV vectors form a basis, *ii)* the computation in (3.10) of the known tensor product form (3.9) of the reference vector associated to a fixed reference co-vector in the SoV basis, *iii)* the exact characterization in Theorem 3.1 of the pseudo-orthogonality relations (3.14), with the description of the non-diagonal couplings in terms of the diagonal ones, and *iv)* the explicit computation of the diagonal couplings in (3.20). Finally, the subsection 3.3 characterizes with Corollary 3.1 the SoV measure in terms of the non-zero SoV co-vector/vector couplings.

In section 4, we use the freedom in the choice of the generating family of conserved charges to construct orthogonal co-vector/vector SoV bases. The subsection 4.1 is dedicated to this construction in the class of quasi-periodic boundary conditions associated to simple spectrum but non-invertible twist matrices. The main theorem there, Theorem 4.1, states the orthogonality properties and the form of the diagonal SoV co-vector/vector couplings. These are similar to the SoV co-vector/vector couplings of the rank one integrable quantum models. In subsection 4.2, these results are used to compute scalar product formulae of separate states (4.77) and (4.81), showing that they take a form similar to the rank one case. Finally, in section 4.3, we introduce a new set of charges (4.93) that extends the results of subsections 4.1 and 4.2 to the general quasi-periodic boundary conditions, associated to simple spectrum and invertible twist matrices.

We give several technical and important proofs in the three appendices. The appendix A details the proof of the tensor product form of SoV starting co-vector/vector in our SoV con-

struction. The appendix B details how our SoV construction holds in the $gl_2$ representations, the aim being to establish one simple example to which compare our higher rank construction. Finally, the appendix C is dedicated to the detailed proof of our Theorem 3.1. Subsection C.1 handles the orthogonality proof, while subsection C.2 details the description of the non-zero SoV co-vector/vector couplings.

## 2 SoV bases for the fundamental representation of the $gl_3$ Yang-Baxter algebra

### 2.1 Fundamental representation of the $gl_3$ Yang-Baxter algebra

We consider here the Yang-Baxter algebra associated to the rational $gl_3$ $R$-matrix:

$$R_{a,b}(\lambda) = \lambda I_{a,b} + \eta \mathbb{P}_{a,b} = \begin{pmatrix} a_1(\lambda) & b_1 & b_2 \\ c_1 & a_2(\lambda) & b_3 \\ c_2 & c_3 & a_3(\lambda) \end{pmatrix} \in \text{End}(V_a \otimes V_b), \qquad (2.1)$$

where $V_a \cong V_b \cong \mathbb{C}^3$ and we have defined:

$$a_j(\lambda) = \begin{pmatrix} \lambda + \eta \delta_{j,1} & 0 & 0 \\ 0 & \lambda + \eta \delta_{j,2} & 0 \\ 0 & 0 & \lambda + \eta \delta_{j,3} \end{pmatrix}, \quad \forall j \in \{1,2,3\},$$

$$b_1 = \begin{pmatrix} 0 & 0 & 0 \\ \eta & 0 & 0 \\ 0 & 0 & 0 \end{pmatrix}, \quad b_2 = \begin{pmatrix} 0 & 0 & 0 \\ 0 & 0 & 0 \\ \eta & 0 & 0 \end{pmatrix}, \quad b_3 = \begin{pmatrix} 0 & 0 & 0 \\ 0 & 0 & 0 \\ 0 & \eta & 0 \end{pmatrix},$$

$$c_1 = \begin{pmatrix} 0 & \eta & 0 \\ 0 & 0 & 0 \\ 0 & 0 & 0 \end{pmatrix}, \quad c_2 = \begin{pmatrix} 0 & 0 & \eta \\ 0 & 0 & 0 \\ 0 & 0 & 0 \end{pmatrix}, \quad c_3 = \begin{pmatrix} 0 & 0 & 0 \\ 0 & 0 & \eta \\ 0 & 0 & 0 \end{pmatrix}, \qquad (2.2)$$

which satisfies the Yang-Baxter equation

$$R_{12}(\lambda - \mu)R_{13}(\lambda)R_{23}(\mu) = R_{23}(\mu)R_{13}(\lambda)R_{12}(\lambda - \mu) \in \text{End}(V_1 \otimes V_2 \otimes V_3) \qquad (2.3)$$

and the scalar Yang-Baxter equation:

$$R_{12}(\lambda)K_1 K_2 = K_2 K_1 R_{12}(\lambda) \in \text{End}(V_1 \otimes V_2), \qquad (2.4)$$

where $K \in \text{End}(V)$ is any $3 \times 3$ matrix. We can define the following monodromy matrix:

$$M_a^{(K)}(\lambda) \equiv K_a R_{a,\mathsf{N}}(\lambda - \xi_\mathsf{N}) \cdots R_{a,1}(\lambda - \xi_1) \in \text{End}(V_a \otimes \mathcal{H}), \qquad (2.5)$$

where $\mathcal{H} = \bigotimes_{n=1}^{\mathsf{N}} V_n$. $M_a^{(K)}(\lambda)$ itself satisfies the Yang-Baxter equation and hence it defines an irreducible $3^\mathsf{N}$-dimensional representation of the $gl_3$ Yang-Baxter algebra for the *inhomogeneity* parameters $\{\xi_1, ..., \xi_\mathsf{N}\}$ in generic complex positions:

$$\xi_i - \xi_j \neq 0, \pm \eta, \quad \forall i, j \in \{1, ..., \mathsf{N}\}. \qquad (2.6)$$

Then, in the framework of the quantum inverse scattering [87–89], the following families of commuting charges exist according to the following:

**Proposition 2.1** ([87–89]). *Defined the antisymmetric projectors:*

$$P_{1,\dots,m}^- = \frac{1}{m!} \sum_{\pi \in S_m} (-1)^{\sigma_\pi} P_\pi, \tag{2.7}$$

*where $S_m$ is the symmetric group of rank $m$, $\sigma_\pi$ the signature of the permutation $\pi$ and*

$$P_\pi(v_1 \otimes \cdots \otimes v_m) = v_{\pi(1)} \otimes \cdots \otimes v_{\pi(m)}, \tag{2.8}$$

*then the following quantum spectral invariants (the fused transfer matrices):*

$$T_m^{(K)}(\lambda) \equiv \operatorname{tr}_{1,\dots,m} \left[ P_{1,\dots,m}^- M_1^{(K)}(\lambda) M_2^{(K)}(\lambda - \eta) \cdots M_m^{(K)}(\lambda - (m-1)\eta) \right], \quad \forall m \in \{1,2,3\}, \tag{2.9}$$

*are one parameter families of mutual commuting operators. Furthermore, the quantum determinant* q-det $M^{(K)}(\lambda) \equiv T_3^{(K)}(\lambda)$ *is central, i.e.*

$$\left[ \text{q-det}\, M^{(K)}(\lambda), M_a^{(K)}(\mu) \right] = 0. \tag{2.10}$$

Moreover, the general fusion identities [87–89] imply the following

**Proposition 2.2** ([87–89]). *The quantum determinant has the following explicit form:*

$$\text{q-det}\, M^{(K)}(\lambda) = \det K \prod_{b=1}^{\mathsf{N}} \left[ (\lambda - \xi_b + \eta) \prod_{m=1}^{2} (\lambda - \xi_b - m\eta) \right], \tag{2.11}$$

*and $T_1^{(K)}(\lambda)$ and $T_2^{(K)}(\lambda)$ are degree $\mathsf{N}$ and $2\mathsf{N}$ in $\lambda$. Their asymptotics are central and coincides with the corresponding two spectral invariants of the matrix $K$:*

$$T_1^{(K,\infty)} \equiv \lim_{\lambda \to \infty} \lambda^{-\mathsf{N}} T_1^{(K)}(\lambda) = \operatorname{tr} K, \quad T_2^{(K,\infty)} \equiv \lim_{\lambda \to \infty} \lambda^{-2\mathsf{N}} T_2^{(K)}(\lambda) = \frac{(\operatorname{tr} K)^2 - \operatorname{tr} K^2}{2}. \tag{2.12}$$

*The fusion identities hold:*

$$T_1^{(K)}(\xi_a) T_m^{(K)}(\xi_a - \eta) = T_{m+1}^{(K)}(\xi_a), \quad \forall m \in \{1,2\}, \tag{2.13}$$

*and $T_2^{(K)}(\lambda)$ has the following $\mathsf{N}$ central zeroes*

$$T_2^{(K)}(\xi_a + \eta) = 0. \tag{2.14}$$

Let us introduce the functions

$$g_{a,\underline{\mathbf{h}}}^{(m)}(\lambda) = \prod_{b \neq a, b=1}^{\mathsf{N}} \frac{\lambda - \xi_b^{(h_b)}}{\xi_a^{(h_a)} - \xi_b^{(h_b)}} \prod_{b=1}^{(m-1)\mathsf{N}} \frac{1}{\xi_a^{(h_a)} - \xi_b^{(-1)}}, \tag{2.15}$$

$$a(\lambda - \eta) = d(\lambda) = \prod_{a=1}^{\mathsf{N}} (\lambda - \xi_a), \quad \xi_b^{(h)} = \xi_b - h\eta, \quad \underline{\mathbf{h}} = \{h_1, \dots, h_{\mathsf{N}}\}, \tag{2.16}$$

and

$$T_{m,\underline{\mathbf{h}}}^{(K,\infty)}(\lambda) = T_m^{(K,\infty)} \prod_{b=1}^{\mathsf{N}} (\lambda - \xi_b^{(h_n)}). \tag{2.17}$$

The known central zeroes and asymptotic behavior imply that the transfer matrix $T_2^{(K)}(\lambda)$ is completely characterized in terms of $T_1^{(K)}(\lambda)$, e.g. by the following interpolation formula

$$T_2^{(K)}(\lambda) = d(\lambda - \eta) \left( T_{2,\underline{\mathbf{h}}=\underline{\mathbf{0}}}^{(K,\infty)}(\lambda) + \sum_{a=1}^{\mathsf{N}} g_{a,\underline{\mathbf{h}}=\underline{\mathbf{0}}}^{(2)}(\lambda) T_1^{(K)}(\xi_a - \eta) T_1^{(K)}(\xi_a) \right), \tag{2.18}$$

where $\underline{\mathbf{h}} = \underline{\mathbf{0}}$ means that for all $k \in \{1, \dots, \mathsf{N}\}$ we have $h_k = 0$.

From now on when we have an $\underline{\mathbf{h}}$ with all the elements equal to the integer 0, 1 or 2 we use directly the bold underlined notation $\underline{\mathbf{0}}$, $\underline{\mathbf{1}}$ and $\underline{\mathbf{2}}$.

## 2.2 On SoV bases construction in our approach

The general Proposition 2.4 of [1] for the construction of the SoV co-vector basis applies in particular to the fundamental representation of the $gl_3$ rational Yang-Baxter algebra. Note that we have presented the construction for the co-vector SoV basis just to get a factorized form of the wave-functions of the transfer matrix eigenvectors in terms of the transfer matrix eigenvalues. Evidently, the same construction applies as well to define a vector SoV basis in which the wave-functions of the transfer matrix eigenco-vectors have the same factorized form. In order to clarify this, we present in the following a proposition for this $gl_3$ case. Let $K$ be a $3 \times 3$ simple spectrum matrix and let us denote with $K_J$ the Jordan form of the matrix $K$ and $W_K$ the invertible matrix defining the change of basis:

$$K = W_K K_J W_K^{-1} \text{ with } K_J = \begin{pmatrix} k_0 & y_1 & 0 \\ 0 & k_1 & y_2 \\ 0 & 0 & k_2 \end{pmatrix}. \tag{2.19}$$

The requirement $K$ simple spectrum implies that we can reduce ourselves to the following three possible cases:

$$i) \ k_i \neq k_j, \ \forall i, j \in \{0, 1, 2\} \text{ and } y_1 = y_2 = 0, \tag{2.20}$$

$$ii) \ k_0 = k_1 \neq k_2, \ y_1 = 1, \ y_2 = 0, \tag{2.21}$$

$$ii) \ k_0 = k_1 = k_2, \ y_1 = 1, \ y_2 = 1. \tag{2.22}$$

Then,

**Proposition 2.3.** *Let $K$ be a $3 \times 3$ simple spectrum matrix, then for almost any choice of $\langle L|, |R\rangle$ and of the inhomogeneities under the condition* (2.6)*, the following set of co-vectors and vectors:*

$$\langle L| \prod_{n=1}^{N} \left( T_1^{(K)}(\xi_n) \right)^{h_n} \text{ for any } \{h_1, ..., h_N\} \in \{0, 1, 2\}^N, \tag{2.23}$$

$$\prod_{n=1}^{N} \left( T_1^{(K)}(\xi_n) \right)^{h_n} |R\rangle \text{ for any } \{h_1, ..., h_N\} \in \{0, 1, 2\}^N, \tag{2.24}$$

*forms a co-vector and vector basis of $\mathcal{H}$, respectively. In particular, we can take the following tensor product forms:*

$$\langle L| = \bigotimes_{a=1}^{N} (x, y, z)_a \Gamma_W^{-1}, \ |R\rangle = \bigotimes_{a=1}^{N} \Gamma_W(r, s, t)_a^{t_a}, \ \Gamma_W = \bigotimes_{a=1}^{N} W_{K,a} \tag{2.25}$$

*simply asking in the case i) $x y z \neq 0$ for the co-vector and $r s t \neq 0$ for the vector, in the case ii) $x z \neq 0$ for the co-vector and $s t \neq 0$ for the vector, in the case iii) $x \neq 0$ for the co-vector and $t \neq 0$ for the vector.*

*Proof.* As shown in the general Proposition 2.4 of [1], the fact that the transfer matrix in the inhomogeneity $\xi_n$ reduces to the twist matrix in the local space $n$ dressed by invertible products of $R$-matrices implies that the set of co-vectors and vectors above defined form bases of $\mathcal{H}^*$ and $\mathcal{H}$, once the following co-vectors and vectors (obtained by taking the asymptotic limit over the $\xi_a$)

$$(x, y, z)W_K^{-1}, (x, y, z)W_K^{-1}K, (x, y, z)W_K^{-1}K^2, \tag{2.26}$$

$$W_K(r, s, t)^t, KW_K(r, s, t)^t, K^2 W_K(r, s, t)^t, \tag{2.27}$$

or equivalently:

$$(x, y, z), (x, y, z)K_J, (x, y, z)K_J^2, \tag{2.28}$$

$$(r, s, t)^t, K_J(r, s, t)^t, K_J^2(r, s, t)^t, \tag{2.29}$$

form bases in $\mathbb{C}^3$, that is the next determinants are non-zero[8]:

$$\det\left((x, y, z)K_J^{i-1}e_j\right)_{i,j\in\{1,2,3\}} = \begin{cases} -xyzV(\mathsf{k}_0, \mathsf{k}_1, \mathsf{k}_2) & \text{in the case i)} \\ x^2zV^2(\mathsf{k}_0, \mathsf{k}_2) & \text{in the case ii)} \\ x^3 & \text{in the case iii)} \end{cases}, \tag{2.30}$$

$$\det\left(e_j^t K_J^{i-1}(r, s, t)\right)_{i,j\in\{1,2,3\}} = \begin{cases} rstV(\mathsf{k}_0, \mathsf{k}_1, \mathsf{k}_2) & \text{in the case i)} \\ s^2tV^2(\mathsf{k}_0, \mathsf{k}_2) & \text{in the case ii)} \\ t^3 & \text{in the case iii)} \end{cases}, \tag{2.31}$$

which leads to the given requirements on the components $x, y, z, r, s, t \in \mathbb{C}$ of the three dimensional co-vector and vectors. $\qquad\square$

Note that both these choices of co-vector and vector SoV bases are perfectly fine to fix the transfer matrix spectrum, by factorized wave functions in terms of transfer matrix eigenvalues for both eigenvectors and eigenco-vectors. However, if we wish to go beyond the spectrum, and compute matrix elements of local operators starting with scalar products of the so-called separate states, we need an appropriate choice of the co-vector and vector SoV bases. In the rank one quantum integrable models, the SoV analysis so far developed [2–6, 23–54] leads to the expectation that the transfer matrix construction of the co-vector and vector SoV bases can be defined in such a way that these are orthogonal bases or similarly that the co-vector and vector Sklyanin's $B$-operator eigenbases both implement the separation of variables for the transfer matrix spectrum. As we will see in the next, in the higher rank quantum integrable models, this is not directly the case if the charges used to construct the co-vector and vector SoV basis are simple powers, or even fusion, of the transfer matrices for general twist $K$.

## 3 Scalar products for co-vector/vector SoV bases

### 3.1 Another construction of co-vector/vector SoV bases

Let us first introduce a slight modification of the co-vector SoV basis w.r.t. the standard one introduced in the previous section by changing the set of conserved charges used to construct them. It reads[9]:

$$\langle\underline{\mathbf{h}}| \equiv \langle h_1, ..., h_\mathsf{N}| = \langle\underline{\mathbf{1}}|\prod_{n=1}^{\mathsf{N}} T_2^{(K)\delta_{h_n,0}}(\xi_n^{(1)})T_1^{(K)\delta_{h_n,2}}(\xi_n), \quad \forall\, h_n \in \{0, 1, 2\}, \tag{3.1}$$

where $\langle\underline{\mathbf{1}}|$ is some generic co-vector of $\mathcal{H}$. Let us remark that for an invertible twist matrix $K$ using the identification:

$$\langle\underline{\mathbf{1}}| = \langle L|\prod_{n=1}^{\mathsf{N}} T_1^{(K)}(\xi_n), \tag{3.2}$$

the two sets of co-vectors defined in (2.23) and (3.1) are identical up to a non-zero normalization of each co-vector; hence the two sets are related by the action of a diagonal matrix. To

---

[8]Here and in the following, we denote by $V(x_1, \dots, x_n)$ the standard Vandermonde determinant $\prod_{i<j}(x_j - x_i)$.

[9]Throughout this section we use compact notations for the left and right SoV bases defined as in (3.1) and (3.7).

be more precise, with such an identification and using the fact that for an invertible $K$-matrix the operator $T_2^{(K)}(\xi_n^{(1)})$ is proportional to the inverse of $T_1^{(K)}(\xi_n)$ due to the fusion relations, we get:

$$\langle \underline{\mathbf{h}}| = \alpha_{\underline{\mathbf{h}}} \langle L| \prod_{n=1}^{N} T_1^{(K)\delta_{h_n,2}-\delta_{h_n,0}+1}(\xi_n), \quad \forall h_n \in \{0,1,2\}, \tag{3.3}$$

where $\alpha_{\underline{\mathbf{h}}} = \prod_{n=1}^{N} (\text{q-det}\, M^{(K)}(\xi_n))^{\delta_{h_n,0}}$ is a non-zero coefficient. Then, being $\delta_{h_n,2}-\delta_{h_n,0}+1 = h_n$ for any $h_n \in \{0,1,2\}$, we get:

$$\langle \underline{\mathbf{h}}| = \alpha_{\underline{\mathbf{h}}} \langle L| \prod_{n=1}^{N} T_1^{(K)h_n}(\xi_n), \forall\, h_n \in \{0,1,2\}, \tag{3.4}$$

thus proving that the two sets defined in (2.23) and in (3.1) are equivalent bases up to an invertible diagonal matrix made of the non-zero coefficients $\alpha_{\underline{\mathbf{h}}}$. Moreover, even if $K$ has zero determinant, it can be proven that the two sets (3.1) and (2.23) are both SoV bases (see next section), the linear transformation relating them being in that case more involved.

## 3.2 Pseudo-orthogonality conditions of these co-vector/vector SoV bases

Here, we show that for the SoV co-vector basis chosen as in (3.1), we can define a *pseudo-orthogonal* vector SoV basis which is orthogonal to the left one for a large set of co-vector/vector couples. We exactly characterize these *pseudo-orthogonality* conditions and the non-zero couplings of these co-vector and vector SoV basis. The corresponding *SoV-measure*, related to the inverse of the scalar products matrix, is completely characterized in the next subsection. It is the starting ingredient to compute matrix elements of local operators in this SoV framework. This will be further employed in forthcoming analysis in this $gl_3$ case as, despite the absence of direct orthogonality, the *SoV-measure* stays reasonably simple to be used in practical computations.

Let us now introduced the vector $|\underline{\mathbf{0}}\rangle$ uniquely characterized by

$$\langle \underline{\mathbf{k}}|\underline{\mathbf{0}}\rangle = \prod_{a=1}^{N} \delta_{0,k_a}. \tag{3.5}$$

Then we have the following

**Proposition 3.1.** *Let $K$ be a $3 \times 3$ simple spectrum matrix, then for almost any choice of the co-vector $\langle \underline{\mathbf{1}}|$, of the vector $|\underline{\mathbf{0}}\rangle$ and of the inhomogeneities under the condition (2.6), the set of co-vectors (3.1)*

$$\langle \underline{\mathbf{h}}| = \langle \underline{\mathbf{1}}| \prod_{n=1}^{N} T_2^{(K)\delta_{h_n,0}}(\xi_n^{(1)}) T_1^{(K)\delta_{h_n,2}}(\xi_n), \tag{3.6}$$

*and the set of vectors:*

$$|\underline{\mathbf{h}}\rangle \equiv \prod_{n=1}^{N} T_2^{(K)\delta_{h_n,1}}(\xi_n) T_1^{(K)\delta_{h_n,2}}(\xi_n)|\underline{\mathbf{0}}\rangle, \tag{3.7}$$

*form co-vector and vector basis of $\mathcal{H}^*$ and $\mathcal{H}$, respectively. In particular, we can take $\langle \underline{\mathbf{1}}|$ of the following tensor product form:*

$$\langle \underline{\mathbf{1}}| = \bigotimes_{a=1}^{N} (x,y,z)_a \Gamma_W^{-1}, \quad \Gamma_W = \bigotimes_{a=1}^{N} W_{K,a}, \tag{3.8}$$

*simply asking $x\,y\,z \neq 0$ in the case i), $x\,z \neq 0$ in the case ii), $x \neq 0$ in the case iii). Then the associated vector $|\underline{0}\rangle$ having the property (3.5) also has tensor product form:*

$$|\underline{0}\rangle = \Gamma_W \bigotimes_{a=1}^{N} |0,a\rangle, \qquad (3.9)$$

*where we have defined*

$$|0,a\rangle = \frac{1}{\Delta} \begin{pmatrix} k_2(yk_0 - xy_1)(zk_2 + yy_2) - (yk_1 + xy_1)(xy_1y_2 + k_0(zk_1 - yy_2)) \\ x(xk_0y_1y_2 + k_0^2(zk_1 - yy_2) - k_1k_2(zk_2 + yy_2)) \\ x(k_0 + k_1)k_2(yk_1 + xy_1 - yk_0) \end{pmatrix}_a, \quad (3.10)$$

*with*

$$\Delta = x\Big(yk_0 - yk_1 - xy_1\Big)\Big(z(k_0 - k_2)(k_1 - k_2) + y_2\big(y(k_2 - k_1) + xy_1\big)\Big)\text{q-det}\,M^{(I)}(\xi_a - 2\eta). \quad (3.11)$$

*Proof.* The proof that these two sets are indeed bases of the Hilbert space and its dual can be performed along the same lines as the one presented already in [1] and in the previous section. Namely, using the polynomial character of all the expressions involved in the inhomogeneity parameters $\xi_n$ it is enough to prove the proposition in some point in the parameter space. This is achieved by scaling the inhomogeneity parameters from a single scalar, as $\xi_n = n\xi$, and sending the parameter $\xi$ to infinity. In turn, this amounts to obtain the asymptotic behavior of the transfer matrices in that limit. The leading term for the operator $T_1^{(K)}(\xi_n)$ is given by $\xi^{N-1}K_n$ times some constant, while for the operator $T_2^{(K)}(\xi_n^{(1)})$ it is given by $\xi^{2(N-1)}(2K_n^2 - 2K_n\,\text{tr}(K) + \text{tr}(K)^2 - \text{tr}(K^2))$ times some other constant. Hence, it is enough to exhibit a co-vector $\langle u|$ such that the set $\langle u|$, $\langle u|K$, $\langle u|K^2$ is a basis of $\mathbb{C}^3$, which is the case as soon as $K$ has simple spectrum. Similarly, the asymptotic of the operator $T_2^{(K)}(\xi_n^{(0)})$ is found proportional to the matrix $\xi^{2(N-1)}(K_n^2 - K_n\,\text{tr}(K))$, leading to the same conclusion. By these arguments, all we need to prove is that the co-vectors

$$(x,y,z)\tilde{K}_J, (x,y,z), (x,y,z)K_J, \qquad (3.12)$$

where $\tilde{K}_J$ is the adjoint matrix of $K_J$, form a tridimensional basis. If we denote by $M_{x,y,z,K_J}$ the $3 \times 3$ matrix which lines are given by these three co-vectors, it holds:

$$\det M_{x,y,z,K_J} = \begin{cases} -xyzV(k_0, k_1, k_2) & \text{in the case i)} \\ x^2zV^2(k_0, k_2) & \text{in the case ii)} \\ x^3 & \text{in the case iii),} \end{cases} \qquad (3.13)$$

so that in the case i) we take $x\,y\,z \neq 0$, in the case ii) we take $x\,z \neq 0$ and finally in the case iii) the condition is $x \neq 0$. The construction of the orthogonal vector is a standard computation in $\mathbb{C}^3$ and the fact that it defines a vector basis by action of $K$ and $K^2$ follows from a direct computation. Another proof uses the characteristic equation of $K$. Finally, the fact that the reference vector for the right SoV basis can be then chosen of tensor product form is proven in the appendix A. $\qquad \square$

Let us now compute the scalar products of these two SoV bases as follows:

**Theorem 3.1.** *Let all the notations be the same as in Proposition 3.1, then the following pseudo-orthogonality relations hold:*

$$\mathcal{N}_{\underline{h},\underline{k}} = \langle \underline{h}|\underline{k}\rangle = \langle \underline{k}|\underline{k}\rangle \left( \delta_{\underline{h},\underline{k}} + C_{\underline{h}}^{\underline{k}} \sum_{r=1}^{n_{\underline{k}}} (\det K)^r \sum_{\substack{\alpha \cup \beta \cup \gamma = \mathbf{1}_{\underline{k}}, \\ \alpha,\beta,\gamma \text{ disjoint}, \#\alpha = \#\beta = r}} \delta_{\underline{h},\underline{k}_{\alpha,\beta}^{(0,2)}} \right), \qquad (3.14)$$

where the $C_{\underline{\mathbf{h}}}^{\underline{\mathbf{k}}}$ are non-zero and independent w.r.t. $\det K$, $n_{\underline{\mathbf{k}}}$ is the integer part of $(\sum_{a=1}^{\mathsf{N}} \delta_{k_a,1})/2$. We have used the further notations

$$\underline{\mathbf{k}}_{\alpha,\beta}^{(0,2)} \equiv (k_1(\alpha,\beta),...,k_{\mathsf{N}}(\alpha,\beta)) \in \{0,1,2\}^{\mathsf{N}}, \tag{3.15}$$

$$\mathbf{1}_{\underline{\mathbf{k}}} \equiv \{a \in \{1,...,\mathsf{N}\} : k_a = 1\}, \tag{3.16}$$

with

$$k_a(\alpha,\beta) = 0, \quad k_b(\alpha,\beta) = 2, \quad \forall a \in \alpha, b \in \beta \tag{3.17}$$

$$k_c(\alpha,\beta) = k_c, \quad \forall c \in \{1,...,\mathsf{N}\}\setminus\{\alpha \cup \beta\}. \tag{3.18}$$

Moreover, we prove that it holds:

$$\mathsf{N}_{\underline{\mathbf{h}}} = \langle \underline{\mathbf{h}}|\underline{\mathbf{h}} \rangle \tag{3.19}$$

$$= \left(\prod_{a=1}^{\mathsf{N}} \frac{d\big(\xi_a^{(1)}\big)}{d\big(\xi_a^{(1+\delta_{h_a,1}+\delta_{h_a,2})}\big)}\right) \frac{V^2(\xi_1,...,\xi_{\mathsf{N}})}{V\big(\xi_1^{(\delta_{h_1,2}+\delta_{h_1,1})},...,\xi_{\mathsf{N}}^{(\delta_{h_{\mathsf{N}},1}+\delta_{h_{\mathsf{N}},2})}\big) V\big(\xi_1^{(\delta_{h_1,2})},...,\xi_{\mathsf{N}}^{(\delta_{h_{\mathsf{N}},2})}\big)}. \tag{3.20}$$

*Proof.* The heavy proofs of the pseudo-orthogonality and of the expressions of non-zero SoV co-vector/vector couplings are given in Appendix C. There, the coefficients $C_{\underline{\mathbf{h}}}^{\underline{\mathbf{k}}}$ are characterized completely, but implicitly, by an unwieldy recursion that we do not solve for the generic case. We compute them in the simplest case, see (C.73). $\square$

It is worth to make some remarks on the above theorem. Let us first comment that the sum in (3.14), for any fixed $\underline{\mathbf{k}}$ and $\underline{\mathbf{h}}$, always reduces to at most one single non-zero term. Indeed, fixing $\underline{\mathbf{k}} \neq \underline{\mathbf{h}}$, we can have a non-zero coupling between the vector and co-vector associated if and only if there exists a couple of sets $(\alpha,\beta) \subset \mathbf{1}_{\underline{\mathbf{k}}}$ with the same cardinality $r \leq n_{\underline{\mathbf{k}}}$ such that $\underline{\mathbf{h}} = \underline{\mathbf{k}}_{\alpha,\beta}^{(0,2)}$, and of course if the couple $(\alpha,\beta)$ exists it is unique. The above condition means that if $\sum_{a=1}^{\mathsf{N}} \delta_{k_a,1}$ is smaller or equal to one, then the standard orthogonality works, i.e. only $\underline{\mathbf{h}} = \underline{\mathbf{k}}$ produces a non-zero co-vector/vector coupling. While if $\sum_{a=1}^{\mathsf{N}} \delta_{k_a,1}$ is bigger or equal to two, we have non-zero couplings also for all the co-vectors of (3.1) with[10] $\underline{\mathbf{h}} = \underline{\mathbf{k}}_{\alpha,\beta}^{(0,2)}$. Let us remark that if one looks to this pseudo-orthogonality condition in one quantum site, then the basis (3.7) naturally emerges as the candidate to get the orthogonal basis to (3.1). Indeed, for one site, orthogonality is satisfied by them while the fact that the orthogonality is not satisfied for higher number of quantum sites is intrinsically related to the form of fusion relations of the transfer matrices for higher rank. From these considerations follows our statement that if we want to obtain mutually orthogonal co-vector/vector SoV bases, we have to use different[11] families of commuting conserved charges to generate the co-vector and the vector SoV bases.

It is also useful to make some link with the preexisting work [76] in the SoV framework. In fact, the set of vectors (3.7) has been introduced recently in [76] as the set of eigenvectors of a $C$-operator, which plays a similar role to the Sklyanin's $B$-operator. There, the starting vector, analogous to our $|\underline{\mathbf{0}}\rangle$, is taken as some not better defined eigenvector of this $C$-operator, and the proofs that $C$ is diagonalizable and that so (3.7) form a basis are not addressed, while the co-vector/vector coupling of these SoV bases is represented with some integral form.

In our paper, we prove that (3.7) is a basis, we fix the tensor product form of the starting vector $|\underline{\mathbf{0}}\rangle$ in terms of the starting co-vector $\langle \underline{\mathbf{1}}|$ and the general twist matrix $K$, we characterize

---

[10]That is for the $\underline{\mathbf{h}}$ obtained from $\underline{\mathbf{k}}$ removing one or more couples of $(k_a = 1, k_b = 1)$ and substituting them with $(k_a(\alpha,\beta) = 0, k_b(\alpha,\beta) = 2)$.

[11]w.r.t. those used above.

completely the form of the co-vector/vector couplings of the two SoV bases and from them the *SoV-measure*.

Let us also remark that in [76] is given a selection rule which selects sectors of the quantum space which are orthogonal, which translates in our setting as

$$\langle \underline{\mathbf{h}}|\underline{\mathbf{k}}\rangle = 0 \ \ \text{if} \ \sum_{a=1}^{\mathsf{N}} \delta_{h_a,1} \neq \sum_{a=1}^{\mathsf{N}} \delta_{k_a,1}. \tag{3.21}$$

This is compatible with our result (3.14), but much less restrictive as one can easily understand by looking, for example, to our formula for $r = 1$. In this case, the $\underline{\mathbf{h}}$ fixing the co-vector in (3.7) and $\underline{\mathbf{k}}$ fixing the vector in (3.1) differ only on one couple of index $(h_a, h_b) \neq (k_a, k_b)$. The above selection rule only imposes that $\langle \underline{\mathbf{h}}|\underline{\mathbf{k}}\rangle = 0$ if $h_a + h_b \neq k_a + k_b$, while our formula instead specifies that $\langle \underline{\mathbf{h}}|\underline{\mathbf{k}}\rangle = 0$ unless $k_a = k_b = 1$ and $h_a + h_b = 2$.

## 3.3 On higher rank SoV measure

In the Theorem 3.1, we have shown that the original higher rank SoV co-vector and vector bases as defined in (3.7) and (3.1) are not mutual orthogonal basis if the twist matrix is invertible. Here, we want to show that from the Theorem 3.1, we can also characterize the SoV measure associated to these bases, i.e. the measure to be used in the computation of scalar products of separate states in these co-vector and vector bases.

Let us start introducing the following sets of co-vectors and vectors that are bases of the Hilbert space orthogonal to our left and right SoV bases:

$$_p\langle \underline{\mathbf{h}}| \ \text{and} \ |\underline{\mathbf{h}}\rangle_p, \quad \forall \underline{\mathbf{h}} \in \{0,1,2\}^{\mathsf{N}}, \tag{3.22}$$

uniquely characterized by the following orthogonality conditions[12]:

$$_p\langle \underline{\mathbf{k}}|\underline{\mathbf{h}}\rangle = \delta_{\underline{\mathbf{k}},\underline{\mathbf{h}}} \langle \underline{\mathbf{h}}|\underline{\mathbf{h}}\rangle, \ \ \langle \underline{\mathbf{k}}|\underline{\mathbf{h}}\rangle_p = \delta_{\underline{\mathbf{k}},\underline{\mathbf{h}}} \langle \underline{\mathbf{h}}|\underline{\mathbf{h}}\rangle, \quad \forall \underline{\mathbf{h}}, \underline{\mathbf{k}} \in \{0,1,2\}^{\mathsf{N}}, \tag{3.23}$$

where $|\underline{\mathbf{h}}\rangle$ and $\langle \underline{\mathbf{h}}|$ are the vectors and co-vectors of the SoV basis (3.7) and (3.1), respectively. Clearly, the set generated by the $_p\langle \underline{\mathbf{h}}|$ and $|\underline{\mathbf{h}}\rangle_p$ are bases of the Hilbert space, and moreover we have the following decompositions of the identity:

$$\mathbb{I} = \sum_{\underline{\mathbf{h}}} \frac{|\underline{\mathbf{h}}\rangle_p \langle \underline{\mathbf{h}}|}{\mathsf{N}_{\underline{\mathbf{h}}}} = \sum_{\underline{\mathbf{h}}} \frac{|\underline{\mathbf{h}}\rangle \ _p\langle \underline{\mathbf{h}}|}{\mathsf{N}_{\underline{\mathbf{h}}}}, \tag{3.24}$$

where the sums run over all the possible values of the multiple index $(\underline{\mathbf{h}})$. As a consequence, the transfer matrix eigenco-vectors and eigenvectors admit the following SoV representations in terms of their eigenvalues:

$$|t_a\rangle = \sum_{\underline{\mathbf{h}}} \prod_{n=1}^{\mathsf{N}} t_{2,a}^{\delta_{h_n,0}}(\xi_n^{(1)}) t_{1,a}^{\delta_{h_n,2}}(\xi_n) \frac{|\underline{\mathbf{h}}\rangle_p}{\mathsf{N}_{\underline{\mathbf{h}}}}, \tag{3.25}$$

$$\langle t_a| = \sum_{\underline{\mathbf{h}}} \prod_{n=1}^{\mathsf{N}} t_{2,a}^{\delta_{h_n,1}}(\xi_n) t_{1,a}^{\delta_{h_n,2}}(\xi_n) \frac{_p\langle \underline{\mathbf{h}}|}{\mathsf{N}_{\underline{\mathbf{h}}}}. \tag{3.26}$$

Thus, we can naturally give the following definitions of separate vectors :

$$|\alpha\rangle = \sum_{\underline{\mathbf{k}}} \alpha_{\underline{\mathbf{k}}} \frac{|\underline{\mathbf{k}}\rangle_p}{\mathsf{N}_{\underline{\mathbf{k}}}}, \quad \alpha_{\underline{\mathbf{h}}} \equiv \prod_{a=1}^{\mathsf{N}} \alpha_a^{(h_a)} \tag{3.27}$$

---

[12]Note here that we have included, for convenience, in the orthogonality relations the normalisation factor $\langle \underline{\mathbf{h}}|\underline{\mathbf{h}}\rangle$ as it leads to more natural identifications in the particular case where the right and left SoV bases are directly orthogonal to each other.

with factorized coordinates $\alpha_{\underline{\mathbf{h}}}$ and separate co-vectors,

$$\langle \beta | = \sum_{\underline{\mathbf{h}}} \beta_{\underline{\mathbf{h}}} \frac{{}_p \langle \mathbf{h} |}{N_{\underline{\mathbf{h}}}}, \quad \beta_{\underline{\mathbf{h}}} \equiv \prod_{a=1}^{N} \beta_a^{(h_a)} \tag{3.28}$$

with factorized coordinates $\beta_{\underline{\mathbf{h}}}$ on the respective bases. The scalar product of such two separate vector and co-vector reads:

$$\langle \beta | \alpha \rangle = \sum_{\underline{\mathbf{h}}, \underline{\mathbf{k}}} \beta_{\underline{\mathbf{h}}} \mathcal{M}_{\underline{\mathbf{h}}, \underline{\mathbf{k}}} \alpha_{\underline{\mathbf{k}}} \tag{3.29}$$

with the *SoV measure* $\mathcal{M}_{\underline{\mathbf{h}}, \underline{\mathbf{k}}}$ defined as:

$$\mathcal{M}_{\underline{\mathbf{h}}, \underline{\mathbf{k}}} = \frac{{}_p \langle \underline{\mathbf{h}} | \underline{\mathbf{k}} \rangle_p}{N_{\underline{\mathbf{h}}} . N_{\underline{\mathbf{k}}}}. \tag{3.30}$$

It can be obtained from the knowledge of the scalar products between the vectors and co-vectors of the two bases orthogonal to our chosen SoV bases:

$$_p \langle \underline{\mathbf{h}} | \underline{\mathbf{k}} \rangle_p, \quad \forall \underline{\mathbf{h}}, \underline{\mathbf{k}} \in \{0, 1, 2\}^{N}. \tag{3.31}$$

Let us note here that these two matrices $\mathcal{N}_{\underline{\mathbf{h}}, \underline{\mathbf{k}}}$ and $\mathcal{M}_{\underline{\mathbf{h}}, \underline{\mathbf{k}}}$, modulo some normalisation, have direct interpretation as change of bases matrices between the bases $\langle \underline{\mathbf{h}} |$ and $_p \langle \underline{\mathbf{k}} |$, namely we have:

$$\langle \underline{\mathbf{h}} | = \sum_{\underline{\mathbf{k}}} \mathcal{N}_{\underline{\mathbf{h}}, \underline{\mathbf{k}}} \frac{{}_p \langle \mathbf{k} |}{N_{\underline{\mathbf{k}}}}, \tag{3.32}$$

and conversely,

$$\frac{{}_p \langle \mathbf{k} |}{N_{\underline{\mathbf{k}}}} = \sum_{\underline{\mathbf{h}}} \mathcal{M}_{\underline{\mathbf{k}}, \underline{\mathbf{h}}} \langle \underline{\mathbf{h}} |. \tag{3.33}$$

We also have similar relations (with transposition) for the bases $| \underline{\mathbf{k}} \rangle$ and $| \underline{\mathbf{h}} \rangle_p$. Moreover, it easy to verify that these two matrices are inverse to each other:

$$\sum_{\underline{\mathbf{h}}} \mathcal{M}_{\underline{\mathbf{k}}, \underline{\mathbf{h}}} \cdot \mathcal{N}_{\underline{\mathbf{h}}, \underline{\mathbf{l}}} = \delta_{\underline{\mathbf{k}}, \underline{\mathbf{l}}}. \tag{3.34}$$

Hence to compute the SoV measure $\mathcal{M}_{\underline{\mathbf{k}}, \underline{\mathbf{h}}}$, we just need to get the inverse of the matrix of scalar products $\mathcal{N}_{\underline{\mathbf{k}}, \underline{\mathbf{h}}}$; in the following we show how to characterize it in terms of $\mathcal{N}_{\underline{\mathbf{k}}, \underline{\mathbf{h}}}$, proving in particular that it has in fact the same form as $\mathcal{N}_{\underline{\mathbf{k}}, \underline{\mathbf{h}}}$[13].

Let us start proving the following:

**Lemma 3.1.** *The vectors* $| \underline{\mathbf{h}} \rangle_p$ *of the basis orthogonal to the SoV co-vector basis (3.1) admit the following decompositions in the SoV vector basis (3.7):*

$$| \underline{\mathbf{h}} \rangle_p = | \underline{\mathbf{h}} \rangle + \sum_{r=1}^{n_{\underline{\mathbf{h}}}} c^r \sum_{\substack{\alpha \cup \beta \cup \gamma = \mathbf{1}_{\underline{\mathbf{h}}}, \\ \alpha, \beta, \gamma \text{ disjoint}, \\ \#\alpha = \#\beta = r}} B_{\alpha, \beta, \underline{\mathbf{h}}} | \underline{\mathbf{h}}_{\alpha, \beta}^{(0, 2)} \rangle, \tag{3.35}$$

---

[13]Somehow this is not surprising as in the appropriate labelling of the SoV bases, the matrix $\mathcal{N}_{\underline{\mathbf{k}}, \underline{\mathbf{h}}}$ is lower-triangular with finite depth out of the diagonal and hence its inverse should have a similar form.

where the coefficients $B_{\alpha,\beta,\underline{\mathbf{h}}}$ are completely characterized by the following recursion formula:

$$B_{\alpha,\beta,\underline{\mathbf{h}}} = -\left( \bar{C}_{\underline{\mathbf{h}}_{\alpha,\beta}^{(0,2)}}^{\underline{\mathbf{h}}} + \sum_{\substack{\alpha'\subset\alpha,\ \beta'\subset\beta,\\ 1\leq\#\alpha'=\#\beta'\leq\#\alpha-1}} B_{\alpha',\beta',\underline{\mathbf{h}}} \bar{C}_{\underline{\mathbf{h}}_{\alpha,\beta}^{(0,2)}}^{\underline{\mathbf{h}}_{\alpha',\beta'}^{(0,2)}} \right), \tag{3.36}$$

and

$$\bar{C}_{\underline{\mathbf{s}}}^{\underline{\mathbf{r}}} = \frac{\langle\underline{\mathbf{r}}|\underline{\mathbf{r}}\rangle}{\langle\underline{\mathbf{s}}|\underline{\mathbf{s}}\rangle} C_{\underline{\mathbf{s}}}^{\underline{\mathbf{r}}}, \tag{3.37}$$

where $C_{\underline{\mathbf{s}}}^{\underline{\mathbf{r}}}$ are the coefficients of the measure (3.14).

*Proof.* The fact that we can write each vector $|\underline{\mathbf{h}}\rangle_p$, satisfying (3.23), in terms of the SoV vectors $|\underline{\mathbf{k}}\rangle$ follows from the fact that these last ones form a basis. Here we have to prove that the above expression for $|\underline{\mathbf{h}}\rangle_p$ and for its coefficients indeed imply the orthogonality condition (3.23).

Let us start observing that this is the case for the diagonal term. Indeed, the following identity follows:

$$\langle\underline{\mathbf{h}}|\underline{\mathbf{h}}\rangle_p = \langle\underline{\mathbf{h}}|\underline{\mathbf{h}}\rangle, \tag{3.38}$$

by the measure (3.14) being

$$\underline{\mathbf{h}} \underset{(C.45)}{\neq} \underline{\mathbf{h}}_{\alpha,\beta}^{(0,,2)}, \quad \forall\alpha,\beta\subset\mathbf{1}_{\underline{\mathbf{h}}}, \text{ disjoint with } 1\leq\#\alpha=\#\beta\leq n_{\underline{\mathbf{h}}}. \tag{3.39}$$

So we are left with the proof of the orthogonality of

$$\langle\underline{\mathbf{k}}|\underline{\mathbf{h}}\rangle_p = 0, \quad \forall\underline{\mathbf{k}}\neq\underline{\mathbf{h}}, \underline{\mathbf{k}}\in\{0,1,2\}^N. \tag{3.40}$$

Let us start observing that for any $\underline{\mathbf{k}}$ such that

$$\underline{\mathbf{k}} \underset{(C.45)}{\neq} \underline{\mathbf{h}}, \tag{3.41}$$

then it also follows

$$\underline{\mathbf{k}} \underset{(C.45)}{\neq} \underline{\mathbf{h}}_{\alpha,\beta}^{(0,2)}, \quad \forall\alpha,\beta\subset\mathbf{1}_{\underline{\mathbf{h}}}, \text{ disjoint with } 1\leq\#\alpha=\#\beta\leq n_{\underline{\mathbf{h}}}, \tag{3.42}$$

and so by the measure (3.14) the orthogonality holds.

So, we are left with the proof of the orthogonality for the case $\underline{\mathbf{k}} = \underline{\mathbf{h}}_{\mu,\delta}^{(0,2)}$ for any fixed disjoint sets $\mu\subset\mathbf{1}_{\underline{\mathbf{h}}}$ and $\delta\subset\mathbf{1}_{\underline{\mathbf{h}}}$ such that $1\leq\#\mu=\#\delta\leq n_{\underline{\mathbf{h}}}$. Let us observe that the following inequalities holds:

$$\underline{\mathbf{h}}_{\alpha,\beta}^{(0,2)} \underset{(C.45)}{\neq} \underline{\mathbf{h}}_{\mu,\delta}^{(0,2)}, \tag{3.43}$$

for any disjoint sets $\alpha$ and $\beta$ contained in $\mathbf{1}_{\underline{\mathbf{h}}}$ with $\#\alpha=\#\beta$ such that

$$\alpha\nsubseteq\mu \text{ and } \beta\nsubseteq\delta. \tag{3.44}$$

Then, by the measure (3.14), we get the following co-vector/vector coupling:

$$\langle\underline{\mathbf{h}}_{\mu,\delta}^{(0,2)}|\underline{\mathbf{h}}\rangle_p = \langle\underline{\mathbf{h}}_{\mu,\delta}^{(0,2)}|\underline{\mathbf{h}}\rangle + \sum_{r=1}^{n_{\underline{\mathbf{h}}}} c^r \sum_{\substack{\alpha\cup\beta\cup\gamma=\mathbf{1}_{\underline{\mathbf{h}}},\\ \alpha,\beta,\gamma \text{ disjoint}, \#\alpha=\#\beta=r}} B_{\alpha,\beta,\underline{\mathbf{h}}}\langle\underline{\mathbf{h}}_{\mu,\delta}^{(0,2)}|\underline{\mathbf{h}}_{\alpha,\beta}^{(0,2)}\rangle \tag{3.45}$$

$$= \langle\underline{\mathbf{h}}_{\mu,\delta}^{(0,2)}|\underline{\mathbf{h}}\rangle + c^{\#\delta}B_{\mu,\delta,\underline{\mathbf{h}}}\langle\underline{\mathbf{h}}_{\mu,\delta}^{(0,2)}|\underline{\mathbf{h}}_{\mu,\delta}^{(0,2)}\rangle + \sum_{r=1}^{\#\delta-1} c^r \sum_{\substack{\alpha\subset\mu,\beta\subset\delta,\\ \#\alpha=\#\beta=r}} B_{\alpha,\beta,\underline{\mathbf{h}}}\langle\underline{\mathbf{h}}_{\mu,\delta}^{(0,2)}|\underline{\mathbf{h}}_{\alpha,\beta}^{(0,2)}\rangle,$$

$$\tag{3.46}$$

that we impose to be zero to satisfy the orthogonality condition:

$$\langle \underline{\mathbf{h}}_{\mu,\delta}^{(0,2)} | \underline{\mathbf{h}} \rangle_p = 0, \quad \forall \mu \subset \mathbf{1}_{\underline{\mathbf{h}}}, \delta \subset \mathbf{1}_{\underline{\mathbf{h}}} \text{ disjoint with } 1 \leq \#\mu = \#\delta \leq n_{\underline{\mathbf{h}}}. \tag{3.47}$$

Here, the main observation is that this can be seen as one equation in one unknown $B_{\mu,\delta,\underline{\mathbf{h}}}$, and solved as it follows:

$$B_{\mu,\delta,\underline{\mathbf{h}}} = -\frac{\langle \underline{\mathbf{h}}_{\mu,\delta}^{(0,2)} | \underline{\mathbf{h}} \rangle c^{-\#\delta}}{\langle \underline{\mathbf{h}}_{\mu,\delta}^{(0,2)} | \underline{\mathbf{h}}_{\mu,\delta}^{(0,2)} \rangle} - \sum_{r=1}^{\#\delta-1} c^{r-\#\delta} \sum_{\substack{\alpha \subset \mu, \beta \subset \delta, \\ \#\alpha = \#\beta = r}} B_{\alpha,\beta,\underline{\mathbf{h}}} \frac{\langle \underline{\mathbf{h}}_{\mu,\delta}^{(0,2)} | \underline{\mathbf{h}}_{\alpha,\beta}^{(0,2)} \rangle}{\langle \underline{\mathbf{h}}_{\mu,\delta}^{(0,2)} | \underline{\mathbf{h}}_{\mu,\delta}^{(0,2)} \rangle}, \tag{3.48}$$

in terms of the known SoV co-vector/vector couplings and of the coefficients $B_{\alpha,\beta,\underline{\mathbf{h}}}$ for any $\alpha \subset \mu, \beta \subset \delta$, with $1 \leq \#\alpha = \#\beta \leq \#\delta - 1$. Then, by using the formulae (3.14) we get:

$$\frac{\langle \underline{\mathbf{h}}_{\mu,\delta}^{(0,2)} | \underline{\mathbf{h}} \rangle c^{-\#\delta}}{\langle \underline{\mathbf{h}}_{\mu,\delta}^{(0,2)} | \underline{\mathbf{h}}_{\mu,\delta}^{(0,2)} \rangle} = \bar{C}_{\underline{\mathbf{h}}_{\mu,\delta}^{(0,2)}}^{\underline{\mathbf{h}}}, \quad \left. \frac{\langle \underline{\mathbf{h}}_{\mu,\delta}^{(0,2)} | \underline{\mathbf{h}}_{\alpha,\beta}^{(0,2)} \rangle c^{r-\#\delta}}{\langle \underline{\mathbf{h}}_{\mu,\delta}^{(0,2)} | \underline{\mathbf{h}}_{\mu,\delta}^{(0,2)} \rangle} \right|_{\substack{\alpha \subset \mu, \beta \subset \delta, \\ \#\alpha = \#\beta = r}} = \bar{C}_{\underline{\mathbf{h}}_{\mu,\delta}^{(0,2)}}^{\underline{\mathbf{h}}_{\alpha,\beta}^{(0,2)}}, \tag{3.49}$$

from which our formula (3.36) easily follows.

Now, it is simple to argue that (3.36) gives us recursively all the coefficient $B_{\mu,\delta,\underline{\mathbf{h}}}$ for any $\mu \subset \mathbf{1}_{\underline{\mathbf{h}}}, \delta \subset \mathbf{1}_{\underline{\mathbf{h}}}$ disjoint, with $1 \leq \#\mu = \#\delta \leq n_{\underline{\mathbf{h}}}$.

In the case $\#\mu = \#\delta = 1$, the formula (3.36) reads:

$$B_{a,b,\underline{\mathbf{h}}} = -\bar{C}_{\underline{\mathbf{h}}_{a,b}^{(0,2)}}^{\underline{\mathbf{h}}}, \quad \forall a \neq b \in \mathbf{1}_{\underline{\mathbf{h}}}, \tag{3.50}$$

which fixes completely these coefficients. Then, we can consider the case of the generic couple of disjoint sets $\mu \subset \mathbf{1}_{\underline{\mathbf{h}}}, \delta \subset \mathbf{1}_{\underline{\mathbf{h}}}$, with $\#\mu = \#\delta = 2$. In these cases, we have that the formula (3.36) reads:

$$B_{\mu,\delta,\underline{\mathbf{h}}} = -\bar{C}_{\underline{\mathbf{h}}_{\mu,\delta}^{(0,2)}}^{\underline{\mathbf{h}}} - \sum_{a \in \mu, b \in \delta} B_{a,b,\underline{\mathbf{h}}} \bar{C}_{\underline{\mathbf{h}}_{\mu,\delta}^{(0,2)}}^{\underline{\mathbf{h}}_{a,b}^{(0,2)}}, \tag{3.51}$$

which fixes completely these coefficients in terms of those computed in the first step of the recursion.

In this way the formula (3.36) fixes the coefficients $B_{\mu',\delta',\underline{\mathbf{h}}}$ for any fixed couple of disjoint sets $\mu' \subset \mathbf{1}_{\underline{\mathbf{h}}}, \delta' \subset \mathbf{1}_{\underline{\mathbf{h}}}$, with $\#\mu' = \#\delta' = m + 1 \leq n_{\underline{\mathbf{h}}}$, in terms of those already computed, i.e. the $B_{\mu,\delta,\underline{\mathbf{h}}}$ for any fixed couple of disjoint sets $\mu \subset \mu' \subset \mathbf{1}_{\underline{\mathbf{h}}}, \delta \subset \delta' \subset \mathbf{1}_{\underline{\mathbf{h}}}$, with $\#\mu = \#\delta \leq m$. $\square$

Let us note that the coefficients $B_{\alpha,\beta,\underline{\mathbf{h}}}$ are, as previously with the coefficients $C_{\underline{\mathbf{h}}}^{\underline{\mathbf{k}}}$, also characterized in a recursive manner, and their generic expression is missing from this Lemma.

The previous lemma implies the following corollary, which completely characterizes the SoV measure.

**Corollary 3.1.** *Under the same condition of Theorem 3.1, the SoV measure is defined by the following pseudo-orthogonality relations:*

$$_p\langle \underline{\mathbf{h}} | \underline{\mathbf{k}} \rangle_p = \langle \underline{\mathbf{h}} | \underline{\mathbf{h}} \rangle \left( \delta_{\underline{\mathbf{h}},\underline{\mathbf{k}}} + \sum_{r=1}^{n_{\underline{\mathbf{k}}}} c^r \sum_{\substack{\alpha \cup \beta \cup \gamma = \mathbf{1}_{\underline{\mathbf{k}}}, \\ \alpha,\beta,\gamma \text{ disjoint, } \#\alpha = \#\beta = r}} B_{\alpha,\beta,\underline{\mathbf{k}}} \delta_{\underline{\mathbf{h}},\underline{\mathbf{k}}_{\alpha,\beta}^{(0,2)}} \right). \tag{3.52}$$

*Proof.* We have to use just the expression derived in the previous lemma for the generic vector

$$|\underline{\mathbf{k}}\rangle_p = |\underline{\mathbf{k}}\rangle + \sum_{r=1}^{n_{\underline{\mathbf{k}}}} \mathsf{c}^r \sum_{\substack{\alpha \cup \beta \cup \gamma = \mathbf{1}_{\underline{\mathbf{k}}}, \\ \alpha,\beta,\gamma \text{ disjoint}, \, \#\alpha=\#\beta=r}} B_{\alpha,\beta,\underline{\mathbf{k}}} |\underline{\mathbf{k}}^{(0,2)}_{\alpha,\beta}\rangle, \tag{3.53}$$

and the definition of the co-vectors $_p\langle\underline{\mathbf{h}}|$ for which it holds:

$$_p\langle\underline{\mathbf{h}}|\underline{\mathbf{k}}\rangle_p = {}_p\langle\underline{\mathbf{h}}|\underline{\mathbf{k}}\rangle + \sum_{r=1}^{n_{\underline{\mathbf{k}}}} \mathsf{c}^r \sum_{\substack{\alpha \cup \beta \cup \gamma = \mathbf{1}_{\underline{\mathbf{k}}}, \\ \alpha,\beta,\gamma \text{ disjoint}, \, \#\alpha=\#\beta=r}} B_{\alpha,\beta,\underline{\mathbf{k}}} \, {}_p\langle\underline{\mathbf{h}}|\underline{\mathbf{k}}^{(0,2)}_{\alpha,\beta}\rangle \tag{3.54}$$

$$= {}_p\langle\underline{\mathbf{h}}|\underline{\mathbf{h}}\rangle \left( \delta_{\underline{\mathbf{h}},\underline{\mathbf{k}}} + \sum_{r=1}^{n_{\underline{\mathbf{k}}}} \mathsf{c}^r \sum_{\substack{\alpha \cup \beta \cup \gamma = \mathbf{1}_{\underline{\mathbf{k}}}, \\ \alpha,\beta,\gamma \text{ disjoint}, \, \#\alpha=\#\beta=r}} B_{\alpha,\beta,\underline{\mathbf{k}}} \, \delta_{\underline{\mathbf{h}},\underline{\mathbf{k}}^{(0,2)}_{\alpha,\beta}} \right), \tag{3.55}$$

and being $_p\langle\underline{\mathbf{h}}|\underline{\mathbf{h}}\rangle = \langle\underline{\mathbf{h}}|\underline{\mathbf{h}}\rangle$, our result follows. $\qquad\square$

# 4 On the construction of orthogonal co-vector/vector SoV bases

We would like now to introduce a new family of commuting conserved charges in order to construct from them orthogonal co-vector/vector SoV bases. We first describe our construction for the class of simple spectrum and non-invertible $K$-matrices. Then, from this class, we will define a new family of commuting conserved charges $\mathbb{T}(\lambda)$ which allows for the construction of the co-vector/vector orthogonal SoV bases for a generic simple spectrum $K$-matrix. The scalar product of separate states w.r.t. the charges $\mathbb{T}(\lambda)$, a class of co-vector/vector which contains the transfer matrix eigenstates, are computed and shown to have a form similar to those of the $gl_2$ case once one of the two states is a $\mathbb{T}(\lambda)$ eigenvector.

## 4.1 The case of non-invertible $\hat{K}$-matrices with simple spectrum

In the $gl_3$ case, the construction of a vector SoV basis orthogonal to the left one is not automatic, as it was in the $gl_2$ case. Here, it seems that the choice of the appropriate family of commuting conserved charges to construct the basis plays a fundamental role. In this subsection, we consider the special case of a simple spectrum $\hat{K}$-matrix with one zero eigenvalue. The orthogonal co-vector and vector SoV bases will be constructed using the transfer matrices as in the previous section.

**Theorem 4.1.** *Let $\hat{K}$ be a $3 \times 3$ simple spectrum matrix with one zero eigenvalue. For almost any choice of the co-vector $\langle\mathbf{1}|$ and of the inhomogeneities under the condition* (2.6)*, the set of co-vectors* (3.1) *and vectors* (3.7) *form SoV co-vector and SoV vector bases of $\mathcal{H}^*$ and $\mathcal{H}$, respectively. In particular, we can take $\langle\mathbf{1}|$ of the tensor product form* (3.8)*, then the associated vector $|\mathbf{0}\rangle$ has the tensor product form defined in* (3.9)-(3.10) *and* (3.1) *and* (3.7) *are basis of $\mathcal{H}$ simply asking $x\,y\,z \neq 0$ in the case i), $x\,z \neq 0$ in the case ii), $x \neq 0$ in the case iii).*

*Furthermore,* (3.1) *and* (3.7) *are mutually orthogonal SoV bases, i.e. they define the following decomposition of the identity:*

$$\mathbb{I} \equiv \sum_{\underline{\mathbf{h}}} \frac{|\underline{\mathbf{h}}\rangle\langle\underline{\mathbf{h}}|}{N_{\underline{\mathbf{h}}}}, \tag{4.1}$$

*with*

$$N_{\underline{\mathbf{h}}} = \prod_{a=1}^{N} \frac{d(\xi_a^{(1)})}{d\left(\xi_a^{(1+\delta_{h_a,1}+\delta_{h_a,2})}\right)} \frac{V^2(\xi_1,...,\xi_N)}{V\left(\xi_1^{(\delta_{h_1,2}+\delta_{h_1,1})},...,\xi_N^{(\delta_{h_N,1}+\delta_{h_N,2})}\right) V\left(\xi_1^{(\delta_{h_1,2})},...,\xi_N^{(\delta_{h_N,2})}\right)}. \tag{4.2}$$

*Proof.* This theorem follows immediately from the results of Theorem 3.1 putting to zero the determinant of the matrix $K$. $\qquad\square$

However, the proof of our Theorem 3.1 is rather involved and takes quite numerous steps that we give in the appendices. It is therefore of interest to have a more elementary proof in the case at hand, namely whenever the simple spectrum twist matrix $K$ has zero determinant or better to say as soon as the fusion relations for the transfer matrices simplify due to the vanishing of its associated quantum determinant. In fact, this case will provide the generic idea to get orthogonal left and right SoV bases in the general situation. So let us explain from now on a direct proof of this theorem.

*Idea of the direct proof.* The statement that (3.1) is a co-vector basis of $\mathcal{H}$ is proven as in the previous proposition. Indeed the main condition:

$$\det M_{x,y,z,\hat{K}_J} \neq 0 \tag{4.3}$$

can be satisfied as well in the case $\det \hat{K} = 0$. In fact, if the matrix $\hat{K}$ satisfies the case i), we take $\mathsf{k}_2 = 0$ and the condition is still $x\,y\,z \neq 0$; if the matrix $\hat{K}$ satisfies the case ii), we take $\mathsf{k}_0 = 0$ or $\mathsf{k}_2 = 0$ and the condition is still $x\,z \neq 0$. Finally in the case iii) with $\mathsf{k}_0 = \mathsf{k}_1 = \mathsf{k}_2 = 0$ the condition is still $x \neq 0$. So that we are left with the proof of the orthogonality conditions. which can be proven by using the next results. $\qquad\square$

The first step in the direct proof of the above theorem is to obtain the SoV representations for the action of the transfer matrices in the case where the fusion relations simplify due to the vanishing of its associated quantum determinant. It is given in the following Proposition.

**Proposition 4.1.** *Under the same conditions of the above theorem, the following interpolation formulae hold for the transfer matrices:*
*i) On the SoV co-vector basis:*

$$\langle\underline{\mathbf{h}}|T_2^{(\hat{K})}(\lambda) = d(\lambda-\eta)\left(\sum_{a=1}^{N}\delta_{h_a,1}g_{a,\underline{\mathbf{z}}(\underline{\mathbf{h}})}^{(2)}(\lambda)\langle\underline{\mathbf{h}}|T_a^- + T_{2,\underline{\mathbf{z}}(\underline{\mathbf{h}})}^{(\hat{K},\infty)}(\lambda)\langle\underline{\mathbf{h}}|\right), \tag{4.4}$$

*and*

$$\langle\underline{\mathbf{h}}|T_1^{(\hat{K})}(\lambda) = T_{1,\underline{\mathbf{y}}(\underline{\mathbf{h}})}^{(\hat{K},\infty)}(\lambda)\langle\underline{\mathbf{h}}| + \sum_{a=1}^{N}\delta_{h_a,1}g_{a,\underline{\mathbf{y}}(\underline{\mathbf{h}})}^{(1)}(\lambda)\langle\underline{\mathbf{h}}|T_a^+ + \sum_{a=1}^{N}\delta_{h_a,2}g_{a,\underline{\mathbf{y}}(\underline{\mathbf{h}})}^{(1)}(\lambda)d(\xi_a^{(1)})$$

$$\times\left(\sum_{b=1}^{N}\delta_{h_b(a),1}g_{b,\underline{\mathbf{z}}(\underline{\mathbf{h}}_a^{(1)})}^{(2)}(\xi_a)\langle\underline{\mathbf{h}}_a^{(1)}|T_b^- + T_{2,\underline{\mathbf{z}}(\underline{\mathbf{h}}_a^{(1)})}^{(\hat{K},\infty)}(\xi_a)\langle\underline{\mathbf{h}}_a^{(1)}|\right), \tag{4.5}$$

*where*

$$\underline{\mathbf{z}}(\underline{\mathbf{h}}) = \{\delta_{h_1,1}+\delta_{h_1,2},...,\delta_{h_N,1}+\delta_{h_N,2}\}, \quad \underline{\mathbf{y}}(\underline{\mathbf{h}}) = \{\delta_{h_1,2},...,\delta_{h_N,2}\}, \tag{4.6}$$

$$\underline{\mathbf{h}}_a^{(1)} = \underline{\mathbf{h}} - (h_a-1)\underline{\mathbf{e}}_a \quad \text{with} \quad \underline{\mathbf{e}}_a = \{\delta_{1,a},...,\delta_{N,a}\}, \tag{4.7}$$

*and*

$$\langle h_1,...,h_a,...,h_N|T_a^\pm = \langle h_1,...,h_a\pm 1,...,h_N|. \tag{4.8}$$

*ii) On the SoV vector basis:*

$$T_2^{(\hat{K})}(\lambda)|\underline{\mathbf{h}}\rangle = d(\lambda-\eta)\left(\sum_{a=1}^{\mathsf{N}}\delta_{h_a,0}g_{a,\underline{\mathbf{z}}(\underline{\mathbf{h}})}^{(2)}(\lambda)\boldsymbol{T}_a^+|\underline{\mathbf{h}}\rangle + |\underline{\mathbf{h}}\rangle T_{2,\underline{\mathbf{z}}(\underline{\mathbf{h}})}^{(\hat{K},\infty)}(\lambda)\right),\tag{4.9}$$

*and*

$$T_1^{(\hat{K})}(\lambda)|\underline{\mathbf{h}}\rangle = |\underline{\mathbf{h}}\rangle T_{1,\underline{\mathbf{y}}(\underline{\mathbf{h}})}^{(\hat{K},\infty)}(\lambda) + \sum_{a=1}^{\mathsf{N}}\delta_{h_a,0}g_{a,\underline{\mathbf{y}}(\underline{\mathbf{h}})}^{(1)}(\lambda)\left(\boldsymbol{T}_a^+\right)^2|\underline{\mathbf{h}}\rangle + \sum_{a=1}^{\mathsf{N}}\delta_{h_a,2}g_{a,\underline{\mathbf{y}}(\underline{\mathbf{h}})}^{(1)}(\lambda)\boldsymbol{T}_a^-|\underline{\mathbf{h}}\rangle$$
$$+ \sum_{a=1}^{\mathsf{N}}\delta_{h_a,1}g_{a,\underline{\mathbf{y}}(\underline{\mathbf{h}})}^{(1)}(\lambda)d(\xi_a^{(1)})\left(\sum_{b=1}^{\mathsf{N}}\delta_{\bar{h}_b(a),0}g_{b,\underline{\mathbf{z}}(\underline{\mathbf{h}}_a^{(2)})}^{(2)}(\xi_a)\boldsymbol{T}_b^+|\underline{\mathbf{h}}_a^{(2)}\rangle + |\underline{\mathbf{h}}_a^{(2)}\rangle T_{2,\underline{\mathbf{z}}(\underline{\mathbf{h}}_a^{(2)})}^{(\hat{K},\infty)}(\xi_a)\right),\tag{4.10}$$

*where*

$$\underline{\mathbf{h}}_a^{(2)} = \underline{\mathbf{h}} - (h_a-2)\underline{\mathbf{e}}_a\,,\tag{4.11}$$

*and*

$$\boldsymbol{T}_a^{\pm}|h_1,...,h_a,...,h_{\mathsf{N}}\rangle = |h_1,...,h_a\pm 1,...,h_{\mathsf{N}}\rangle.\tag{4.12}$$

*Proof.* The fusion identities take the following form in the case $\det\hat{K}=0$:

$$T_2^{(\hat{K})}(\xi_a^{(1)})T_1^{(\hat{K})}(\xi_a) = \text{q-det}\,M^{(\hat{K})}(\xi_a) = 0,\tag{4.13}$$

$$T_1^{(\hat{K})}(\xi_a^{(1)})T_1^{(\hat{K})}(\xi_a) = T_2^{(\hat{K})}(\xi_a),\tag{4.14}$$

$$T_2^{(\hat{K})}(\xi_a^{(1)})T_2^{(\hat{K})}(\xi_a) = T_1^{(\hat{K})}(\xi_a^{(1)})\,\text{q-det}\,M_a^{(\hat{K})}(\xi_a) = 0.\tag{4.15}$$

Let us take the generic co-vector[14] $\langle h_1,...,h_{\mathsf{N}}|$ and then use the interpolation formula:

$$T_2^{(\hat{K})}(\lambda) = d(\lambda-\eta)\left(T_{2,\underline{\mathbf{z}}(\underline{\mathbf{h}})}^{(\hat{K},\infty)}(\lambda) + \sum_{a=1}^{\mathsf{N}}g_{a,\underline{\mathbf{z}}(\underline{\mathbf{h}})}^{(2)}(\lambda)T_2^{(\hat{K})}\left(\xi_a^{(\delta_{h_a,1}+\delta_{h_a,2})}\right)\right),\tag{4.16}$$

to compute the action of $T_2^{(\hat{K})}(\lambda)$ on $\langle h_1,...,h_{\mathsf{N}}|$:

$$\langle\underline{\mathbf{h}}|T_2(\lambda) = d(\lambda-\eta)\left(T_{2,\underline{\mathbf{z}}(\underline{\mathbf{h}})}^{(\hat{K},\infty)}(\lambda)\langle\underline{\mathbf{h}}| + \sum_{a=1}^{\mathsf{N}}g_{a,\underline{\mathbf{z}}(\underline{\mathbf{h}})}^{(2)}(\lambda)\langle\underline{\mathbf{h}}|T_2^{(\hat{K})}\left(\xi_a^{(\delta_{h_a,1}+\delta_{h_a,2})}\right)\right),\tag{4.17}$$

where it holds:

$$\langle h_1,...,h_a,...,h_{\mathsf{N}}|T_2^{(\hat{K})}\left(\xi_a^{(\delta_{h_a,2}+\delta_{h_a,1})}\right) = \delta_{h_a,1}\langle h_1,...,h_a'=0,...,h_{\mathsf{N}}|\,,\tag{4.18}$$

being by the fusion identities:

$$\langle h_1,...,h_a=2,...,h_{\mathsf{N}}|T_2^{(\hat{K})}(\xi_a^{(1)}) = 0,\tag{4.19}$$

$$\langle h_1,...,h_a=0,...,h_{\mathsf{N}}|T_2^{(\hat{K})}(\xi_a) = 0.\tag{4.20}$$

This proves our interpolation formula for the action of $T_2^{(\hat{K})}(\lambda)$ on the generic element of the co-vector basis $\langle h_1,...,h_{\mathsf{N}}|$. Let us now use the following interpolation formula:

$$T_1^{(\hat{K})}(\lambda) = T_{1,\underline{\mathbf{y}}(\underline{\mathbf{h}})}^{(\hat{K},\infty)}(\lambda) + \sum_{a=1}^{\mathsf{N}}g_{a,\underline{\mathbf{y}}(\underline{\mathbf{h}})}^{(1)}(\lambda)T_1^{(\hat{K})}(\xi_a^{(\delta_{h_a,2})}),\tag{4.21}$$

---

[14]For convenience, in this section, we do not use uniformly compact notations for the SoV basis co-vectors as their explicit form is sometimes more convenient to write the action of the transfer matrices on them.

to compute the action of $T_1^{(\hat{K})}(\lambda)$ on $\langle h_1, ..., h_N|$:

$$\langle \underline{\mathbf{h}}|T_1^{(\hat{K})}(\lambda) = T_{1,\mathbf{y}(\underline{\mathbf{h}})}^{(\hat{K},\infty)}(\lambda)\langle \underline{\mathbf{h}}| + \sum_{a=1}^{N} g_{a,\mathbf{y}(\underline{\mathbf{h}})}^{(1)}(\lambda)\left(\delta_{h_a,1}\langle \underline{\mathbf{h}}|T_1^{(\hat{K})}(\xi_a) + \delta_{h_a,2}\langle \underline{\mathbf{h}}|T_1^{(\hat{K})}(\xi_a^{(1)})\right), \quad (4.22)$$

where we have used that by the fusion identity it holds:

$$\langle h_1, ..., h_a = 0, ..., h_N|T_1^{(\hat{K})}(\xi_a) = 0, \quad (4.23)$$

so that the above formula reduces to:

$$\langle \underline{\mathbf{h}}|T_1^{(\hat{K})}(\lambda) = T_{1,\mathbf{y}(\underline{\mathbf{h}})}^{(\hat{K},\infty)}(\lambda)\langle \underline{\mathbf{h}}| + \sum_{a=1}^{N} g_{a,\mathbf{y}(\underline{\mathbf{h}})}^{(1)}(\lambda)\delta_{h_a,1}\langle \underline{\mathbf{h}}|T_1^{(\hat{K})}(\xi_a) + \sum_{a=1}^{N} g_{a,\mathbf{y}(\underline{\mathbf{h}})}^{(1)}(\lambda)\delta_{h_a,2}\langle \underline{\mathbf{h}}_a^{(1)}|T_2^{(\hat{K})}(\xi_a).$$
$$(4.24)$$

This leads to our result for the action of $T_1^{(\hat{K})}(\lambda)$ on $\langle \underline{\mathbf{h}}|$ once we use the proven formula for the action of $T_2^{(\hat{K})}(\lambda)$ on $\langle \underline{\mathbf{h}}|$.

Let us now prove the interpolation formulae for the action on SoV vectors. The fusion identities for the case $\det \hat{K} = 0$ imply[15]:

$$T_2^{(\hat{K})}(\xi_a^{(1)})|h_1, ..., h_a \neq 0, ..., h_N\rangle = 0, \quad (4.25)$$

and so the only contributions to the action of $T_2^{(\hat{K})}(\lambda)$ on a vector $|\underline{\mathbf{h}}\rangle$ come from the central asymptotic term and the terms for $h_a = 0$, from which the action of $T_2^{(\hat{K})}(\lambda)$ easily follows. Let us now remark that the fusion identities together with the commutativity of the transfer matrices also imply the following actions:

$$T_1^{(\hat{K})}(\xi_a)|h_1, ..., h_a = 1, ..., h_N\rangle = T_2^{(\hat{K})}(\xi_a)|h_1, ..., h_a = 2, ..., h_N\rangle, \quad (4.26)$$

$$T_1^{(\hat{K})}(\xi_a^{(1)})|h_1, ..., h_a = 2, ..., h_N\rangle = |h_1, ..., h_a = 1, ..., h_N\rangle, \quad (4.27)$$

$$T_1^{(\hat{K})}(\xi_a)|h_1, ..., h_a = 0, ..., h_N\rangle = |h_1, ..., h_a = 2, ..., h_N\rangle, \quad (4.28)$$

from which we get the following action by interpolation formula

$$T_1^{(\hat{K})}(\lambda)|\underline{\mathbf{h}}\rangle = |\underline{\mathbf{h}}\rangle T_{1,\mathbf{y}(\underline{\mathbf{h}})}^{(\hat{K},\infty)}(\lambda) + \sum_{a=1}^{N} \delta_{h_a,0} g_{a,\mathbf{y}(\underline{\mathbf{h}})}^{(1)}(\lambda)\left(T_a^+\right)^2|\underline{\mathbf{h}}\rangle$$

$$+ \sum_{a=1}^{N} \delta_{h_a,2} g_{a,\mathbf{y}(\underline{\mathbf{h}})}^{(1)}(\lambda)T_a^-|\underline{\mathbf{h}}\rangle + \sum_{a=1}^{N} \delta_{h_a,1} g_{a,\mathbf{y}(\underline{\mathbf{h}})}^{(1)}(\lambda)T_2^{(\hat{K})}(\xi_a)T_a^+|\underline{\mathbf{h}}\rangle, \quad (4.29)$$

from which our formula for $T_1^{(\hat{K})}(\lambda)$ on $|\underline{\mathbf{h}}\rangle$ follows by using the one proven for $T_2^{(\hat{K})}(\lambda)$ on $|\underline{\mathbf{h}}\rangle$. $\qquad \square$

We can complete now the proof of the Theorem 4.1:

*Proof of Theorem 4.1.* Let us start proving the orthogonality condition:

$$\langle h_1, ..., h_N|k_1, ..., k_N\rangle = 0, \quad \forall\{k_1, ..., k_N\} \neq \{h_1, ..., h_N\} \in \{0, 1, 2\}^{\otimes N}. \quad (4.30)$$

The proof is done by induction, assuming that it is true for any vector $|k_1, ..., k_N\rangle$ such that $\sum_{n=1}^{N}(\delta_{k_n,1} + \delta_{k_n,2}) = l$, $l \leq N - 1$, and proving it for vectors $|k_1', ..., k_N'\rangle$ with $\sum_{n=1}^{N}(\delta_{k_n',1} + \delta_{k_n',2}) = l + 1$. To this aim we fix a vector $|k_1, ..., k_N\rangle$ with $\sum_{n=1}^{N}(\delta_{k_n,1} + \delta_{k_n,2}) = l$ and we denote by $\pi$ a permutation on the set $\{1, ..., N\}$ such that:

$$\delta_{k_{\pi(a)},1} + \delta_{k_{\pi(a)},2} = 1 \text{ for } a \leq l \text{ and } k_{\pi(a)} = 0 \text{ for } l < a. \quad (4.31)$$

---

[15]It is important to remark that the only ingredient of the proof for this theorem involve only the simplified fusion relations.

**a)** Let us first compute:

$$\langle h_1,...,h_N|T_2^{(\hat{K})}(\xi_{\pi(l+1)})|k_1,...,k_N\rangle = \langle h_1,...,h_N|k'_1,...,k'_N\rangle, \tag{4.32}$$

where we have defined:

$$k'_{\pi(a)} = k_{\pi(a)}, \ \forall a \in \{1,...,N\}\backslash\{l+1\} \quad \text{and} \quad k'_{\pi(l+1)} = 1, \tag{4.33}$$

for any $\{h_1,...,h_N\} \neq \{k'_1,...,k'_N\} \in \{0,1\}^{\otimes N}$. There are three cases. The first case is $h_{\pi(l+1)} = 0$, then the fusion identity implies:

$$\langle h_1,...,h_N|T_2^{(\hat{K})}(\xi_{\pi(l+1)})|k_1,...,k_N\rangle = 0. \tag{4.34}$$

In the remaining two cases $h_{\pi(l+1)} = 1$ or $h_{\pi(l+1)} = 2$, we can use the interpolation formula to compute the action of $T_2^{(\hat{K})}(\xi_{\pi(l+1)})$ on the co-vector $\langle h_1,...,h_N|$:

$$\langle h_1,...,h_N|T_2^{(\hat{K})}(\xi_{\pi(l+1)})|k_1,...,k_N\rangle = d(\xi_{\pi(l+1)}^{(1)})T_{2,\underline{\mathbf{z}}(\underline{\mathbf{h}})}^{(\hat{K},\infty)}(\xi_{\pi(l+1)})\langle h_1,...,h_N|k_1,...,k_N\rangle \tag{4.35}$$

$$+d(\xi_{\pi(l+1)}^{(1)})\sum_{a=1}^{N}\delta_{h_a,1}g_{a,\underline{\mathbf{z}}(\underline{\mathbf{h}})}^{(2)}(\xi_{\pi(l+1)})\langle h_1,...,h'_a=0,...,h_N|k_1,...,k_N\rangle. \tag{4.36}$$

Let us remark now that from $\{h_1,...,h_N\} \neq \{k'_1,...,k'_N\}$ and $h_{\pi(l+1)} = 1$ or $h_{\pi(l+1)} = 2$, it follows also that $\{h_1,...,h_N\} \neq \{k_1,...,k_N\}$, being $k_{\pi(l+1)} = 0$, so that:

$$\langle h_1,...,h_N|k_1,...,k_N\rangle = 0. \tag{4.37}$$

Moreover, it holds:

$$\delta_{h_a,1}\langle h_1,...,h'_a=0,...,h_N|k_1,...,k_N\rangle = 0, \quad \forall a \in \{1,...,N\}. \tag{4.38}$$

Indeed, if $a \in \{1,...,N\}\backslash\{\pi(l+1)\}$ and $h_a = 1$, we have $\{h_1,...,h'_a=0,...,h_N\} \neq \{k_1,...,k_N\}$, being $k_{\pi(l+1)} = 0 \neq h_{\pi(l+1)} \in \{1,2\}$. While in the case $a = \pi(l+1)$ either $h_{\pi(l+1)} = 2$, so that $\delta_{h_{\pi(l+1)},1} = 0$, or $h_{\pi(l+1)} = 1$ and the condition $\{h_1,...,h_N\} \neq \{k'_1,...,k'_N\}$ implies that there exists at least a $j \neq \pi(l+1)$ such that $h_j \neq k_j$, so that we have still $\{h_1,...,h'_{\pi(l+1)}=0,...,h_N\} \neq \{k_1,...,k_N\}$.

**b)** Let us compute now:

$$\langle h_1,...,h_N|T_1^{(\hat{K})}(\xi_{\pi(l+1)})|k_1,...,k_N\rangle = \langle h_1,...,h_N|k'_1,...,k'_N\rangle, \tag{4.39}$$

where we have defined:

$$k'_{\pi(a)} = k_{\pi(a)}, \ \forall a \in \{1,...,N\}\backslash\{l+1\} \quad \text{and} \quad k'_{\pi(l+1)} = 2, \tag{4.40}$$

for any $\{h_1,...,h_N\} \neq \{k'_1,...,k'_N\} \in \{0,1\}^{\otimes N}$. There are three cases as well. The first case is $h_{\pi(l+1)} = 0$, then the fusion identity implies:

$$\langle h_1,...,h_N|T_1^{(\hat{K})}(\xi_{\pi(l+1)})|k_1,...,k_N\rangle = 0. \tag{4.41}$$

For the second case for $h_{\pi(l+1)} = 1$, it holds:

$$\langle h_1,...,h_N|T_1^{(\hat{K})}(\xi_{\pi(l+1)})|k_1,...,k_N\rangle$$
$$= \langle h_1,...,h_{\pi(l+1)}=2,...,h_N|k_1,...,k_{\pi(l+1)}=0,...,k_N\rangle = 0. \tag{4.42}$$

So we are left with the case $h_{\pi(l+1)} = 2$. Note that in this case the condition $\{h_1, ..., h_N\} \neq \{k'_1, ..., k'_N\}$ implies that there exists a $j \neq \pi(l+1)$ such that $h_j \neq k_j$, being by definition $h_{\pi(l+1)} = k'_{\pi(l+1)} = 2$. We can use the following interpolation formula to compute the action of $T_1^{(\hat{K})}(\xi_{\pi(l+1)})$ on the co-vector $\langle h_1, ..., h_N|$:

$$
\begin{aligned}
\langle h_1, ..., h_N | T_1^{(\hat{K})}(\xi_{\pi(l+1)}) | \underline{\mathbf{k}} \rangle = {} & T_{1, \mathbf{y}(\underline{\mathbf{h}})}^{(\hat{K}, \infty)}(\xi_{\pi(l+1)}) \langle h_1, ..., h_N | \underline{\mathbf{k}} \rangle \\
& + \sum_{a=1}^{N} g_{a, \mathbf{y}(\underline{\mathbf{h}})}^{(1)}(\xi_{\pi(l+1)}) \delta_{h_a, 1} \langle h_1, ..., h_N | T_1^{(\hat{K})}(\xi_a) | \underline{\mathbf{k}} \rangle \\
& + \sum_{a=1}^{N} g_{a, \mathbf{y}(\underline{\mathbf{h}})}^{(1)}(\xi_{\pi(l+1)}) \delta_{h_a, 2} \langle h_1, ..., h_N | T_1^{(\hat{K})}(\xi_a^{(1)}) | \underline{\mathbf{k}} \rangle.
\end{aligned} \tag{4.43}
$$

From $\{h_1, ..., h_N\} \neq \{k_1, ..., k_N\}$ it follows:

$$
\langle h_1, ..., h_N | \underline{\mathbf{k}} \rangle = 0, \tag{4.44}
$$

and, moreover, it holds:

$$
\delta_{h_a, 1} \langle h_1, ..., h_N | T_1^{(\hat{K})}(\xi_a) | \underline{\mathbf{k}} \rangle = \delta_{h_a, 1} \langle h_1, ..., h'_a = 2, ..., h_N | \underline{\mathbf{k}} \rangle = 0, \tag{4.45}
$$

as for $a = \pi(l+1)$ it holds $\delta_{h_a, 1} = 0$, because $h_{\pi(l+1)} = 2$, while for $a \neq \pi(l+1)$ we have still $h_{\pi(l+1)} = 2 \neq k_{\pi(l+1)} = 0$ so that:

$$
\langle h_1, ..., h'_a = 2, ..., h_N | \underline{\mathbf{k}} \rangle = 0. \tag{4.46}
$$

So, we are left with the last sum in (4.43), for which it holds:

$$
\delta_{h_a, 2} \langle h_1, ..., h_N | T_1^{(\hat{K})}(\xi_a^{(1)}) | \underline{\mathbf{k}} \rangle = \delta_{h_a, 2} \langle h_1, ..., h'_a = 1, ..., h_N | T_2^{(\hat{K})}(\xi_a) | \underline{\mathbf{k}} \rangle. \tag{4.47}
$$

i) For $a = \pi(r)$ for $r \geq l+1$ it holds:

$$
\begin{aligned}
\delta_{h_a, 2} \langle h_{\pi(1)}, ..., h'_{\pi(r)} = 1, ..., h_{\pi(N)} | T_2^{(\hat{K})}(\xi_{\pi(r)}) | \underline{\mathbf{k}} \rangle \\
= \delta_{h_a, 2} \langle h_{\pi(1)}, ..., h'_{\pi(r)} = 1, ..., h_{\pi(N)} | k_{\pi(1)}, ..., k''_{\pi(r)} = 1, ..., k_{\pi(N)} \rangle, \quad (4.48)
\end{aligned}
$$

with $\{h_{\pi(1)}, ..., h'_{\pi(r)} = 1, ..., h_{\pi(N)}\} \neq \{k_{\pi(1)}, ..., k''_{\pi(r)} = 1, ..., k_{\pi(N)}\}$. Indeed, if $r = l+1$ we have shown that there is a $j \neq \pi(l+1)$ such that $h_j \neq k_j$, while if $r \geq l+2$ we have still $h_{\pi(l+1)} = 2 \neq k_{\pi(l+1)} = 0$.

So that for $a = \pi(r)$ for $r \geq l+1$, we can use the step a) of the proof to get:

$$
\langle h_{\pi(1)}, ..., h'_{\pi(r)} = 1, ..., h_{\pi(N)} | k_{\pi(1)}, ..., k''_{\pi(r)} = 1, ..., k_{\pi(N)} \rangle = 0. \tag{4.49}
$$

ii) For $a = \pi(r)$ for $r \leq l$. If $k_{\pi(r)} = 2$, then we can write the l.h.s. of (4.47) as it follows:

$$
\delta_{h_{\pi(r)}, 2} \langle h_1, ..., h_N | T_1^{(\hat{K})}(\xi_{\pi(r)}^{(1)}) | \underline{\mathbf{k}} \rangle = \delta_{h_{\pi(r)}, 2} \langle h_1, ..., h_N | k_{\pi(1)}, ..., k''_{\pi(r)} = 1, ..., k_{\pi(N)} \rangle, \tag{4.50}
$$

which is zero being $h_{\pi(r)} = 2 \neq k''_{\pi(r)} = 1$.

If $k_{\pi(r)} = 1$, then we use the interpolation formula:

$$
T_2^{(\hat{K})}(\xi_{\pi(r)}) | \underline{\mathbf{k}} \rangle = d(\xi_{\pi(r)}^{(1)}) \left( | \underline{\mathbf{k}} \rangle T_{2, \underline{\mathbf{z}}(\underline{\mathbf{h}})}^{(\hat{K}, \infty)}(\xi_{\pi(r)}) + \sum_{n=l+1}^{N} g_{n, \underline{\mathbf{z}}(\underline{\mathbf{h}})}^{(2)}(\xi_{\pi(r)}) T_2^{(\hat{K})}(\xi_{\pi(n)}) | \underline{\mathbf{k}} \rangle \right), \tag{4.51}
$$

where we have used that, by the fusion identity:

$$T_2^{(\hat{K})}(\xi_{\pi(n)}^{(1)})|\underline{\mathbf{k}}\rangle = 0 \text{ for } n \le l \text{ as } T_2^{(\hat{K})}(\xi_{\pi(n)}^{(1)})T_{2-\delta_{k_{\pi(n)},2}}^{(\hat{K})}(\xi_{\pi(n)}) = 0. \tag{4.52}$$

Then, for $a = \pi(r)$ for $r \le l$ and if $k_{\pi(r)} = 1$, (4.47) reads:

$$\delta_{h_{\pi(r)},2}\langle h_{\pi(1)}, ..., h'_{\pi(r)} = 1, ..., h_{\pi(\mathsf{N})}|T_2^{(\hat{K})}(\xi_a)|\underline{\mathbf{k}}\rangle$$
$$= \delta_{h_{\pi(r)},2}d(\xi_{\pi(r)}^{(1)})(T_{2,\underline{\mathbf{z}}(\underline{\mathbf{h}})}^{(\hat{K},\infty)}(\xi_{\pi(r)})\langle h_{\pi(1)}, ..., h'_{\pi(r)} = 1, ..., h_{\pi(\mathsf{N})}|\underline{\mathbf{k}}\rangle$$
$$+ \sum_{n=l+1}^{\mathsf{N}} g_{n,\underline{\mathbf{z}}(\underline{\mathbf{h}})}^{(2)}(\xi_{\pi(r)})\langle h_{\pi(1)}, ..., h'_{\pi(r)} = 1, ..., h_{\pi(\mathsf{N})}|k_{\pi(1)}, ..., k''_{\pi(n)} = 1, ..., k_{\pi(\mathsf{N})}\rangle). \tag{4.53}$$

Here we have:

$$\langle h_{\pi(1)}, ..., h'_{\pi(r)} = 1, ..., h_{\pi(\mathsf{N})}|\underline{\mathbf{k}}\rangle = 0, \tag{4.54}$$

being $h_{\pi(l+1)} = 2 \neq k_{\pi(l+1)} = 0$. Moreover, the remaining matrix elements

$$\langle h_{\pi(1)}, ..., h'_{\pi(r)} = 1, ..., h_{\pi(\mathsf{N})}|k_{\pi(1)}, ..., k''_{\pi(n)} = 1, ..., k_{\pi(\mathsf{N})}\rangle, \tag{4.55}$$

for $r \le l$ and $l + 1 \le n$ are such that $\{h_{\pi(1)}, ..., h'_{\pi(r)} = 1, ..., h_{\pi(\mathsf{N})}\} \neq \{k_{\pi(1)}, ..., k''_{\pi(n)} = 1, ..., k_{\pi(\mathsf{N})}\}$. Indeed, for $n = l+1$ it holds $h_{\pi(l+1)} = 2 \neq k''_{\pi(l+1)} = 1$ while for $l + 2 \le n$ it still holds $h_{\pi(l+1)} = 2 \neq k_{\pi(l+1)} = 0$.

Finally, we can apply the step a) of our proof to show that (4.55) is zero for any fixed $l + 1 \le n$, just exchanging the permutation $\pi$ with the following one

$$\pi_n(a) = \pi(a)(1 - \delta_{a,l+1})(1 - \delta_{a,n}) + \pi(n)\delta_{a,l+1} + \pi(l+1)\delta_{a,n}. \tag{4.56}$$

The computation of the SoV measure is standard [37, 39] once one uses the interpolation formulae of the transfer matrices given above. Let us write the elements of the proof. Let us first define:

$$\underline{\mathbf{h}}_a^{(j)} = \underline{\mathbf{h}} - (h_a - j)\underline{\mathbf{e}}_a, \quad \forall a, j \in \{1, ..., \mathsf{N}\} \times \{0, 1, 2\},$$

and compute the matrix elements:

$$\langle \underline{\mathbf{h}}_a^{(1)}|T_2^{(\hat{K})}(\xi_a^{(1)})|\underline{\mathbf{h}}_a^{(0)}\rangle = \langle \underline{\mathbf{h}}_a^{(0)}|\underline{\mathbf{h}}_a^{(0)}\rangle. \tag{4.57}$$

We compute the action of $T_2^{(\hat{K})}(\xi_a^{(1)})$ on the right by using the corresponding interpolation formula, and from the orthogonality conditions we get that there is only one term with non-zero contribution, which reads:

$$\langle \underline{\mathbf{h}}_a^{(1)}|T_2^{(\hat{K})}(\xi_a^{(1)})|\underline{\mathbf{h}}_a^{(0)}\rangle = \langle \underline{\mathbf{h}}_a^{(1)}|\underline{\mathbf{h}}_a^{(1)}\rangle \frac{d(\xi_a^{(2)})}{d(\xi_a^{(1)})} \prod_{n\neq a,n=1}^{\mathsf{N}} \frac{\xi_a^{(1)} - \xi_n^{(\delta_{h_n,1}+\delta_{h_n,2})}}{\xi_a - \xi_n^{(\delta_{h_n,1}+\delta_{h_n,2})}}. \tag{4.58}$$

Similarly, we want to compute:

$$\langle \underline{\mathbf{h}}_a^{(1)}|T_1^{(\hat{K})}(\xi_a)|\underline{\mathbf{h}}_a^{(2)}\rangle = \langle \underline{\mathbf{h}}_a^{(2)}|\underline{\mathbf{h}}_a^{(2)}\rangle, \tag{4.59}$$

by using the interpolation formula for the right action of $T_1^{(\hat{K})}(\xi_a)$, we obtain that once again there is just one term that give a non-zero contribution due to the orthogonality and it reads:

$$\langle \underline{\mathbf{h}}_a^{(1)}|T_1^{(\hat{K})}(\xi_a)|\underline{\mathbf{h}}_a^{(2)}\rangle = \langle \underline{\mathbf{h}}_a^{(1)}|\underline{\mathbf{h}}_a^{(1)}\rangle \prod_{n\neq a,n=1}^{\mathsf{N}} \frac{\xi_a - \xi_n^{(\delta_{h_n,2})}}{\xi_a^{(1)} - \xi_n^{(\delta_{h_n,2})}}, \tag{4.60}$$

from which our formula for the normalization holds. $\qquad\square$

The following corollary holds:

**Corollary 4.1.** *Let $\hat{K}$ be a $3\times3$ simple spectrum matrix with one zero eigenvalue. Then for almost any choice of the co-vector $\langle\mathbf{1}|$ and of the inhomogeneities under the condition* (2.6) *the states*

$$\langle\underline{\mathbf{0}}| = \langle h_1 = 0, ..., h_N = 0|, \quad \langle\underline{\mathbf{2}}| = \langle h_1 = 2, ..., h_N = 2| \tag{4.61}$$

*are $T_2^{(\hat{K})}(\lambda)$ eigenstates:*

$$\langle\underline{\mathbf{0}}|T_2^{(\hat{K})}(\lambda) = t_{2,0}d(\lambda-\eta)d(\lambda)\langle\underline{\mathbf{0}}|, \tag{4.62}$$

$$\langle\underline{\mathbf{2}}|T_2^{(\hat{K})}(\lambda) = t_{2,0}d(\lambda-\eta)d(\lambda+\eta)\langle\underline{\mathbf{2}}|, \tag{4.63}$$

$$T_2^{(\hat{K})}(\lambda)|\underline{\mathbf{h}}\rangle = |\underline{\mathbf{h}}\rangle t_{2,0}d(\lambda-\eta)d(\lambda+\eta), \quad \forall\underline{\mathbf{h}}\in\{1,2\}^N \tag{4.64}$$

*while $\langle\underline{\mathbf{0}}|$ is also $T_1^{(\hat{K})}(\lambda)$ eigenstate:*

$$\langle\underline{\mathbf{0}}|T_1^{(\hat{K})}(\lambda) = t_{1,0}d(\lambda)\langle\underline{\mathbf{0}}|. \tag{4.65}$$

*Proof.* It is enough to take the interpolation formulae for the transfer matrices and apply them over these states. $\square$

**Theorem 4.2.** *Let $\hat{K}$ be a $3 \times 3$ simple spectrum matrix with one zero eigenvalue and with the inhomogeneities under the condition* (2.6). *Then the transfer matrix spectrum is simple and, for almost any choice of the co-vector $\langle\mathbf{1}|$, the vector $|t_a\rangle$ and the co-vector $\langle t_a|$ are transfer matrix eigenstates if and only if they admit (up to an overall normalization) the following separate form in the co-vector and vector SoV eigenbasis:*

$$|t_a\rangle = \sum_{\underline{\mathbf{h}}}\prod_{n=1}^{N} t_{2,a}^{\delta_{h_n,0}}(\xi_n^{(1)})t_{1,a}^{\delta_{h_n,2}}(\xi_n) \frac{|\underline{\mathbf{h}}\rangle}{N_{\underline{\mathbf{h}}}}, \tag{4.66}$$

$$\langle t_a| = \sum_{\underline{\mathbf{h}}}\prod_{n=1}^{N} t_{2,a}^{\delta_{h_n,1}}(\xi_n)t_{1,a}^{\delta_{h_n,2}}(\xi_n) \frac{\langle\underline{\mathbf{h}}|}{N_{\underline{\mathbf{h}}}}, \tag{4.67}$$

*where the index a run in the set of the transfer matrix eigenvalues of $T_1^{(\hat{K})}(\lambda)$ and the coefficients of the states are written in terms of the corresponding eigenvalues:*

$$T_1^{(\hat{K})}(\lambda)|t_a\rangle = |t_a\rangle t_{1,a}(\lambda), \quad T_2^{(\hat{K})}(\lambda)|t_a\rangle = |t_a\rangle t_{2,a}(\lambda), \tag{4.68}$$

$$\langle t_a|T_1^{(\hat{K})}(\lambda) = t_{1,a}(\lambda)\langle t_a|, \quad \langle t_a|T_2^{(\hat{K})}(\lambda) = t_{2,a}(\lambda)\langle t_a|. \tag{4.69}$$

*Finally, if the matrix $\hat{K}$ has simple spectrum and is diagonalizable, the same is true for the transfer matrix $T_1^{(\hat{K})}(\lambda)$, which therefore admits $3^N$ distinct eigenvalues $t_{1,a}(\lambda)$ with $a \in \{1, ..., 3^N\}$.*

*Proof.* Let us compute the matrix element:

$$\langle\underline{\mathbf{h}}|t\rangle = \langle\underline{\mathbf{1}}|\prod_{a=1}^{N} T_2^{(\hat{K})\delta_{h_a,0}}(\xi_a^{(1)})T_1^{(\hat{K})\delta_{h_a,2}}(\xi_a)|t\rangle = \langle\underline{\mathbf{1}}|t\rangle\prod_{a=1}^{N} t_2^{\delta_{h_a,0}}(\xi_a^{(1)})t_1^{\delta_{h_a,2}}(\xi_a). \tag{4.70}$$

From our SoV decomposition of the identity, it holds:

$$|t\rangle = \sum_{\underline{\mathbf{h}}}\langle\underline{\mathbf{h}}|t\rangle \frac{|\underline{\mathbf{h}}\rangle}{N_{\underline{\mathbf{h}}}}, \tag{4.71}$$

and then fixing the normalization of the state $|t\rangle$ by imposing $\langle\underline{\mathbf{1}}|t\rangle = 1$, our statement is proven. $\square$

The functional equation characterization of the transfer matrix eigenvalues and ABA like representations of the states hold also in the case where the $3 \times 3$ simple spectrum matrix $\hat{K}$ has one zero eigenvalue.

## 4.2 Scalar products of separate states in orthogonal SoV basis

Let us introduce the following class of "separate" co-vectors and vectors:

$$|\alpha\rangle = \sum_{\underline{\mathbf{h}}} \prod_{a=1}^{\mathsf{N}} \alpha_a^{(h_a)} \frac{|\underline{\mathbf{h}}\rangle}{\mathsf{N}_{\underline{\mathbf{h}}}}, \quad \langle\alpha| = \sum_{\underline{\mathbf{h}}} \prod_{a=1}^{\mathsf{N}} \alpha_a^{(h_a)} \frac{\langle\underline{\mathbf{h}}|}{\mathsf{N}_{\underline{\mathbf{h}}}}. \tag{4.72}$$

The eigenvectors and co-vectors of the transfer matrix are of this form, with coefficients $\alpha_a^{h_a}$ constrained by their eigenvalue. We have the following scalar product formulae:

**Theorem 4.3.** *Let $\hat{K}$ be a $3 \times 3$ simple spectrum matrix with one zero eigenvalue and let the inhomogeneity condition (2.6) be satisfied. Then, taken the generic transfer matrix eigenvector:*

$$|t_n\rangle = \sum_{\underline{\mathbf{h}}} \prod_{a=1}^{\mathsf{N}} t_{2,n}^{\delta_{h_a,0}}(\xi_a^{(1)}) t_{1,n}^{\delta_{h_a,2}}(\xi_a) \frac{|\underline{\mathbf{h}}\rangle}{\mathsf{N}_{\underline{\mathbf{h}}}}, \tag{4.73}$$

*there exists a permutation $\pi_n$ of the set $\{1, ..., \mathsf{N}\}$ such that:*

$$t_{1,n}(\xi_{\pi_n(b)}) = t_{2,n}(\xi_{\pi_n(a)} - \eta) = 0, \quad \forall (a,b) \in A \times B, \tag{4.74}$$

$$t_{1,n}(\xi_{\pi_n(a)}) \neq 0, \quad t_{2,n}(\xi_{\pi_n(b)} - \eta) \neq 0, \quad \forall (a,b) \in A \times B, \tag{4.75}$$

*where we have defined:*

$$A \equiv \{1, ..., \mathsf{M}_n\}, \quad B \equiv \{\mathsf{M}_n + 1, ..., \mathsf{N}\}. \tag{4.76}$$

*Moreover, the action of the generic separate co-vector $\langle\alpha|$ on it reads:*

$$\langle\alpha|t_n\rangle = \prod_{a=1}^{\mathsf{N}} \frac{d(\xi_a^{(2)})}{d(\xi_a^{(1)})} \frac{V\big(\xi_{\pi_n(1)}^{(1)}, ..., \xi_{\pi_n(\mathsf{M}_n)}^{(1)}\big)}{V\big(\xi_{\pi_n(1)}, ..., \xi_{\pi_n(\mathsf{M}_n)}\big)}$$
$$\times \frac{\det_{\mathsf{N}-\mathsf{M}_n} \mathcal{M}_{+,\mathsf{N}-\mathsf{M}_n}^{(\alpha|x_A t_{2,n})}}{V\big(\xi_{\pi_n(\mathsf{M}_n+1)}, ..., \xi_{\pi_n(\mathsf{N})}\big)} \frac{\det_{\mathsf{M}_n} \mathcal{M}_{-,\mathsf{M}_n}^{(\alpha|x_B t_{1,n})}}{V\big(\xi_{\pi_n(1)}, ..., \xi_{\pi_n(\mathsf{M}_n)}\big)}, \tag{4.77}$$

*where we have defined:*

$$\left(\mathcal{M}_{+,\mathsf{N}-\mathsf{M}_n}^{(\alpha|x_A t_{2,n})}\right)_{(i,j)\in\{1,...,\mathsf{N}-\mathsf{M}_n\}^2} = \sum_{h=0}^{1} \alpha_{\pi_n(\mathsf{M}_n+i)}^{(h)} x_A^{1-h}(\xi_{\pi_n(\mathsf{M}_n+i)}) \underline{t}_{2,n}^h(\xi_{\pi_n(\mathsf{M}_n+i)}^{(1)})(\xi_{\pi_n(\mathsf{M}_n+i)}^{(h)})^{j-1}, \tag{4.78}$$

$$\left(\mathcal{M}_{-,\mathsf{M}_n}^{(\alpha|x_B t_{1,n})}\right)_{(i,j)\in\{1,...,\mathsf{M}_n\}^2} = \sum_{h=0}^{1} \alpha_{\pi_n(i)}^{(h+1)} x_B^h(\xi_{\pi_n(i)}) t_{1,n}^h(\xi_{\pi_n(i)})(\xi_{\pi_n(i)}^{(h)})^{j-1}, \tag{4.79}$$

*with*

$$x_A(\lambda) = \prod_{a=1}^{\mathsf{M}_n} \frac{\lambda - \xi_{\pi_n(a)} + \eta}{\lambda - \xi_{\pi_n(a)}}, \quad x_B(\lambda) = \prod_{a=1+\mathsf{M}_n}^{\mathsf{N}} \frac{\lambda - \xi_{\pi_n(b)} - \eta}{\lambda - \xi_{\pi_n(b)}}, \tag{4.80}$$

$$\underline{t}_{2,n}(\lambda) = d(\lambda) t_{2,n}(\lambda)/d(\lambda - \eta).$$

*We have the following identity for the action of the eigenco-vector $\langle t_n|$ on the eigenvector $|t_n\rangle$:*

$$\langle t_n|t_n\rangle = \prod_{a=1}^{\mathsf{N}} \frac{V\big(\xi_{\pi_n(1)}^{(1)}, ..., \xi_{\pi_n(\mathsf{M}_n)}^{(1)}\big)}{V\big(\xi_{\pi_n(1)}, ..., \xi_{\pi_n(\mathsf{M}_n)}\big)} \prod_{b=1+\mathsf{M}_n}^{\mathsf{N}} t_{2,n}(\xi_{\pi_n(b)}^{(1)}) x_A(\xi_{\pi_n(b)}) \prod_{a=1}^{\mathsf{M}_n} t_{1,n}(\xi_{\pi_n(a)}) \det_{\mathsf{M}_n} \mathcal{T}_{\mathsf{M}_n}, \tag{4.81}$$

*where we have defined:*

$$\left(\mathcal{T}_{\mathsf{M}_n}\right)_{(i,j)\in A^2} = \sum_{h=0}^{1} t_{1,n}(\xi_{\pi_n(i)}^{(1-h)}) x_B^h(\xi_{\pi_n(i)})(\xi_{\pi_n(i)}^{(h)})^{j-1}. \tag{4.82}$$

*Proof.* It is worth recalling that the zero and non-zero pattern of (4.74) and (4.75) has been derived in [81]. There, we have moreover observed that the eigenvalue of the transfer matrix $T_2^{(\hat{K})}(\lambda)$ is completely fixed by them, i.e. it holds

$$t_{2,n}(\lambda) = T_2^{(K,\infty)} d(\lambda - \eta) \prod_{a=1}^{\mathsf{M}_n} (\lambda - \xi_{\pi_n(a)}^{(1)}) \prod_{b=1+\mathsf{M}_n}^{\mathsf{N}} (\lambda - \xi_{\pi_n(b)}). \tag{4.83}$$

The proof of this theorem is a direct consequence of the new found SoV measure (4.2) and of the form of the separate states, from which we get

$$\langle \alpha | t_n \rangle = \sum_{h_1,\dots,h_{\mathsf{N}}=0}^{2} \prod_{a=1}^{\mathsf{N}} \frac{d(\xi_a^{(1+\delta_{h_a,1}+\delta_{h_a,2})})}{d(\xi_a^{(1)})} t_{2,n}^{\delta_{h_a,0}}(\xi_a^{(1)}) t_{1,n}^{\delta_{h_a,2}}(\xi_a) \, \alpha_a^{(h_a)}$$
$$\times \frac{V\left(\xi_1^{(\delta_{h_1,2}+\delta_{h_1,1})},\dots,\xi_{\mathsf{N}}^{(\delta_{h_{\mathsf{N}},1}+\delta_{h_{\mathsf{N}},2})}\right) V\left(\xi_1^{(\delta_{h_1,2})},\dots,\xi_{\mathsf{N}}^{(\delta_{h_{\mathsf{N}},2})}\right)}{V^2(\xi_1,\dots,\xi_{\mathsf{N}})}. \tag{4.84}$$

We now use the existence of the permutation $\pi_n$ and the characterization of the zero and non-zero pattern for the transfer matrix eigenvalues (4.74) and (4.75) to factorize the above sum into two sum and get our result. Indeed, by using them the r.h.s. of (4.84) reads ($\mathsf{M}_n^+ = \mathsf{M}_n + 1$):

$$\sum_{h_{\pi_n(1)},\dots,h_{\pi_n(\mathsf{M}_n)}=1}^{2} \sum_{h_{\pi_n(\mathsf{M}_n+1)},\dots,h_{\pi_n(\mathsf{N})}=0}^{1} \prod_{a=1}^{\mathsf{M}_n} \frac{d(\xi_{\pi_n(a)}^{(2)})}{d(\xi_{\pi_n(a)}^{(1)})} t_{1,n}^{\delta_{h_{\pi_n(a)},2}}(\xi_{\pi_n(a)}) \, \alpha_{\pi_n(a)}^{(h_{\pi_n(a)})}$$
$$\prod_{b=\mathsf{M}_n^+}^{\mathsf{N}} \frac{d\left(\xi_{\pi_n(b)}^{(1+\delta_{h_{\pi_n(b)},1})}\right)}{d(\xi_{\pi_n(b)}^{(1)})} t_{2,n}^{\delta_{h_{\pi_n(b)},0}}(\xi_{\pi_n(b)}^{(1)}) \alpha_{\pi_n(b)}^{(h_{\pi_n(b)})} \frac{V\left(\xi_{\pi_n(1)}^{(1)},\dots,\xi_{\pi_n(\mathsf{M}_n)}^{(1)}\right) V\left(\xi_{\pi_n(1)}^{(\delta_{h_{\pi_n(1)},2})},\dots,\xi_{\pi_n(\mathsf{M}_n)}^{(\delta_{h_{\pi_n(\mathsf{M}_n)},2})}\right)}{V(\xi_{\pi_n(1)},\dots,\xi_{\pi_n(\mathsf{M}_n)}) V(\xi_{\pi_n(1)},\dots,\xi_{\pi_n(\mathsf{M}_n)})}$$
$$\prod_{a=1}^{\mathsf{M}_n} \prod_{b=\mathsf{M}_n^+}^{\mathsf{N}} \frac{\xi_{\pi_n(b)}^{(\delta_{h_{\pi_n(b)},1})} - \xi_{\pi_n(a)}^{(1)}}{\xi_{\pi_n(b)} - \xi_{\pi_n(a)}} \prod_{a=1}^{\mathsf{M}_n} \prod_{b=\mathsf{M}_n^+}^{\mathsf{N}} \frac{\xi_{\pi_n(a)}^{(\delta_{h_{\pi_n(a)},2})} - \xi_{\pi_n(b)}}{\xi_{\pi_n(a)} - \xi_{\pi_n(b)}} \frac{V\left(\xi_{\pi_n(\mathsf{M}_n+1)}^{(\delta_{h_{\pi_n(\mathsf{M}_n+1)},1})},\dots,\xi_{\pi_n(\mathsf{N})}^{(\delta_{h_{\pi_n(\mathsf{N})},1})}\right)}{V(\xi_{\pi_n(\mathsf{M}_n+1)},\dots,\xi_{\pi_n(\mathsf{N})})}. \tag{4.85}$$

We can then factorize out of the above sum the factors:

$$\prod_{a=1}^{\mathsf{N}} \frac{d(\xi_a^{(2)})}{d(\xi_a^{(1)})} \frac{V(\xi_{\pi_n(1)}^{(1)},\dots,\xi_{\pi_n(\mathsf{M}_n)}^{(1)})}{V(\xi_{\pi_n(1)},\dots,\xi_{\pi_n(\mathsf{M}_n)})}, \tag{4.86}$$

being left with the product of the following two independent sum, i.e.

$$\sum_{h_{\pi_n(1)},\dots,h_{\pi_n(\mathsf{M}_n)}=1}^{2} \prod_{a=1}^{\mathsf{M}_n} t_{1,n}^{\delta_{h_{\pi_n(a)},2}}(\xi_{\pi_n(a)}) \, \alpha_{\pi_n(a)}^{(h_{\pi_n(a)})} x_B^{(h_{\pi_n(a)}-1)}(\xi_{\pi_n(a)}) \frac{V\left(\xi_{\pi_n(1)}^{(\delta_{h_{\pi_n(1)},2})},\dots,\xi_{\pi_n(\mathsf{M}_n)}^{(\delta_{h_{\pi_n(\mathsf{M}_n)},2})}\right)}{V(\xi_{\pi_n(1)},\dots,\xi_{\pi_n(\mathsf{M}_n)})} \tag{4.87}$$

times

$$\sum_{h_{\pi_n(\mathsf{M}_n^+)},\dots,h_{\pi_n(\mathsf{N})}=0}^{1} \prod_{b=\mathsf{M}_n^+}^{\mathsf{N}} t_{2,n}^{\delta_{h_{\pi_n(b)},0}}(\xi_{\pi_n(b)}^{(1)}) \alpha_{\pi_n(b)}^{(h_{\pi_n(b)})} x_A^{1-h_{\pi_n(b)}}(\xi_{\pi_n(b)}) \frac{V\left(\xi_{\pi_n(\mathsf{M}_n+1)}^{(\delta_{h_{\pi_n(\mathsf{M}_n+1)},1})},\dots,\xi_{\pi_n(\mathsf{N})}^{(\delta_{h_{\pi_n(\mathsf{N})},1})}\right)}{V(\xi_{\pi_n(\mathsf{M}_n^+)},\dots,\xi_{\pi_n(\mathsf{N})})}. \tag{4.88}$$

As previously remarked in [37, 39], these sums admit a representation in terms of one determinant formulae, thanks to the multi-linearity of the Vandermonde determinant. From this, our result (4.77) follows.

To derive the formula for the "norm" of the transfer matrix eigenvectors, we just have to observe that by the definition of the vector SoV basis, it holds:

$$\langle t_n|\underline{\mathbf{h}}\rangle = \prod_{a=1}^{\mathsf{N}} t_{2,n}^{\delta_{h_a,1}}(\xi_a) t_{1,n}^{\delta_{h_a,2}}(\xi_a) = \prod_{a=1}^{\mathsf{N}} t_{1,n}^{\delta_{h_a,1}}(\xi_a^{(1)}) t_{1,n}^{(\delta_{h_a,1}+\delta_{h_a,2})}(\xi_a), \tag{4.89}$$

and so we have that

$$\langle t_n|\underline{\mathbf{h}}\rangle = 0, \quad \forall (h_{\pi_n(\mathsf{M}_n+1)},...,h_{\pi_n(\mathsf{N})}) \neq (0,...,0). \tag{4.90}$$

Then the sum (4.88) reduces to

$$\prod_{b=1+\mathsf{M}_n}^{\mathsf{N}} \underline{t}_{2,n}(\xi_{\pi_n(b)}^{(1)}) x_A(\xi_{\pi_n(b)}), \tag{4.91}$$

while the first one reads:

$$\prod_{a=1}^{\mathsf{M}_n} t_{1,n}(\xi_{\pi_n(a)}) \sum_{h_{\pi_n(1)},...,h_{\pi_n(\mathsf{M}_n)}=1}^{2} \prod_{a=1}^{\mathsf{M}_n} t_{1,n}(\xi_{\pi_n(a)}^{(2-h_{\pi_n(a)})}) x_B^{(h_{\pi_n(a)}-1)}(\xi_{\pi_n(a)})$$

$$\times \frac{V\left(\xi_{\pi_n(1)}^{(\delta_{h_{\pi_n(1)},2})},...,\xi_{\pi_n(\mathsf{M}_n)}^{(\delta_{h_{\pi_n(\mathsf{M}_n)},2})}\right)}{V(\xi_{\pi_n(1)},...,\xi_{\pi_n(\mathsf{M}_n)})}. \tag{4.92}$$

It is now quite direct to verify the formula (4.81). □

## 4.3 On the extension to the case of simple spectrum and invertible $K$-matrices

The results of the previous subsections give us the possibility to define a new family of conserved charges, from which we can introduce the orthogonal left and right SoV bases also in the case of a general simple spectrum $K$-matrix with non-zero eigenvalues.

Let us assume that $K$ is $3 \times 3$ simple spectrum and diagonalizable matrix with non-zero eigenvalues. Then, by our previous results in the SoV approach [1], we know that the associated transfer matrix $T_1^{(K)}(\lambda)$ is diagonalizable with simple spectrum almost for any value of the inhomogeneities under the condition (2.6), and we have the SoV complete characterization of its spectrum.

Let $\{|t_a^{(K)}\rangle, a \in \{1,..,3^{\mathsf{N}}\}\}$ be the eigenvector basis and let $\{\langle t_a^{(K)}|, a \in \{1,..,3^{\mathsf{N}}\}\}$ be the eigenco-vector basis associated to the transfer matrix $T_1^{(K)}(\lambda)$. We define the two new families of conserved charges:

$$\mathbb{T}_j^{(K)}(\lambda) = \sum_{a=1}^{3^{\mathsf{N}}} t_{j,a}^{(\hat{K})}(\lambda) \frac{|t_a^{(K)}\rangle\langle t_a^{(K)}|}{\langle t_a^{(K)}|t_a^{(K)}\rangle}, \quad \text{with } j \in \{1,2\}. \tag{4.93}$$

Here, we have denoted with $t_{j,a}^{(\hat{K})}(\lambda)$ the spectrum of the transfer matrices $T_j^{(\hat{K})}(\lambda)$ associated to the matrix $\hat{K}$, obtained from $K$ by putting one of its eigenvalue to zero while keeping its spectrum simplicity and its diagonalizable character, i.e.:

$$T_1^{(\hat{K})}(\lambda)|t_a^{(\hat{K})}\rangle = |t_a^{(\hat{K})}\rangle t_{1,a}^{(\hat{K})}(\lambda), \quad T_2^{(\hat{K})}(\lambda)|t_a^{(\hat{K})}\rangle = |t_a^{(\hat{K})}\rangle t_{2,a}^{(\hat{K})}(\lambda), \tag{4.94}$$

$$\langle t_a^{(\hat{K})}|T_1^{(\hat{K})}(\lambda) = t_{1,a}^{(\hat{K})}(\lambda)\langle t_a^{(\hat{K})}|, \quad \langle t_a^{(\hat{K})}|T_2^{(\hat{K})}(\lambda) = t_{2,a}^{(\hat{K})}(\lambda)\langle t_a^{(\hat{K})}|. \tag{4.95}$$

Note that, by construction, the families $\mathbb{T}_j^{(K)}(\lambda)$ are mutually commuting and they commute with the original transfer matrices as they have diagonal form in the eigenbasis of the original transfer matrix $T_1^{(K)}(\lambda)$:

$$\left[\mathbb{T}_l^{(K)}(\lambda),\mathbb{T}_m^{(K)}(\lambda)\right]=\left[T_l^{(K)}(\lambda),\mathbb{T}_m^{(K)}(\lambda)\right]=0,\quad l,m\in\{1,2\}, \tag{4.96}$$

and they share the same spectrum as the transfer matrices $T_j^{(\hat{K})}(\lambda)$. Hence, they satisfy the following fusion equations:

$$\mathbb{T}_2^{(K)}(\xi_a^{(1)})\mathbb{T}_1^{(K)}(\xi_a)=\mathbb{T}_2^{(K)}(\xi_a^{(1)})\mathbb{T}_2^{(K)}(\xi_a)=0, \tag{4.97}$$

$$\mathbb{T}_1^{(K)}(\xi_a^{(1)})\mathbb{T}_1^{(K)}(\xi_a)=\mathbb{T}_2^{(K)}(\xi_a). \tag{4.98}$$

We can now use these new family of conserved charges to construct SoV basis according to (3.1) and (3.7) since the twist matrix $\hat{K}$ has simple spectrum:

$$\langle\widehat{\underline{\mathbf{h}}}|\equiv\langle\mathbf{1}|\prod_{n=1}^{\mathsf{N}}\mathbb{T}_2^{(K)\delta_{h_n,0}}(\xi_n^{(1)})\mathbb{T}_1^{(K)\delta_{h_n,2}}(\xi_n),\quad\forall\,h_n\in\{0,1,2\}, \tag{4.99}$$

$$|\widehat{\underline{\mathbf{h}}}\rangle\equiv\prod_{n=1}^{\mathsf{N}}\mathbb{T}_2^{(K)\delta_{h_n,1}}(\xi_n)\mathbb{T}_1^{(K)\delta_{h_n,2}}(\xi_n)|\mathbf{0}\rangle,\quad\forall\,h_n\in\{0,1,2\}. \tag{4.100}$$

They are mutually orthogonal as the direct proof of Theorem 4.1 uses only the fusion relations which are just identical to the above ones (4.97) and (4.98):

$$\langle\widehat{\underline{\mathbf{k}}}|\widehat{\underline{\mathbf{h}}}\rangle=\mathsf{N}_{\underline{\mathbf{h}}}\prod_{a=1}^{\mathsf{N}}\delta_{h_a,k_a} \tag{4.101}$$

$$=\prod_{a=1}^{\mathsf{N}}\delta_{h_a,k_a}\frac{d(\xi_a^{(1)})}{d\left(\xi_a^{(1+\delta_{h_a,1}+\delta_{h_a,2})}\right)}\frac{V\left(\xi_1^{(\delta_{h_1,2}+\delta_{h_1,1})},...,\xi_{\mathsf{N}}^{(\delta_{h_{\mathsf{N}},1}+\delta_{h_{\mathsf{N}},2})}\right)V\left(\xi_1^{(\delta_{h_1,2})},...,\xi_{\mathsf{N}}^{(\delta_{h_{\mathsf{N}},2})}\right)}{V^2(\xi_1,...,\xi_{\mathsf{N}})}. \tag{4.102}$$

They are also SoV bases as the spectrum of the $\mathbb{T}_j^{(K)}(\lambda)$ is separate in these bases. We have the following representation of the vector and co-vector of the original transfer matrix $T_1^{(K)}(\lambda)$:

$$|t_a^{(K)}\rangle=\sum_{\underline{\mathbf{h}}}\prod_{n=1}^{\mathsf{N}}t_{2,a}^{(\hat{K})\delta_{h_n,0}}(\xi_n^{(1)})t_{1,a}^{(\hat{K})\delta_{h_n,2}}(\xi_n)\frac{|\widehat{\underline{\mathbf{h}}}\rangle}{\mathsf{N}_{\underline{\mathbf{h}}}}, \tag{4.103}$$

$$\langle t_a^{(K)}|=\sum_{\underline{\mathbf{h}}}\prod_{n=1}^{\mathsf{N}}t_{2,a}^{(\hat{K})\delta_{h_n,1}}(\xi_n)t_{1,a}^{(\hat{K})\delta_{h_n,2}}(\xi_n)\frac{\langle\widehat{\underline{\mathbf{h}}}|}{\mathsf{N}_{\underline{\mathbf{h}}}}. \tag{4.104}$$

Moreover, let us comment that separate states of the form

$$|\alpha\rangle=\sum_{\underline{\mathbf{h}}}\prod_{a=1}^{\mathsf{N}}\alpha_a^{(h_a)}\frac{|\widehat{\underline{\mathbf{h}}}\rangle}{\mathsf{N}_{\underline{\mathbf{h}}}},\quad\langle\alpha|=\sum_{\underline{\mathbf{h}}}\prod_{a=1}^{\mathsf{N}}\alpha_a^{(h_a)}\frac{\langle\widehat{\underline{\mathbf{h}}}|}{\mathsf{N}_{\underline{\mathbf{h}}}}, \tag{4.105}$$

satisfy the same Theorem 4.3 with the transfer matrix $T_1^{(K)}(\lambda)$ eigenvectors. This is easily derived by using the representation of the transfer matrix eigenvector in the SoV bases constructed by the conserved charges $\mathbb{T}_1^{(K)}(\lambda)$, since from them one gets scalar product formulae similar to those of the $gl_2$ case, even for the simple spectrum invertible $K$ matrix.

# 5 Conclusions and perspectives

In the present paper we have addressed the problem of computing the scalar products between the left and right SoV bases introduced earlier in [1] (see also [75]) for the fundamental representations of the $\mathcal{Y}(gl_3)$ lattice model with $N$ sites. These SoV bases are determined from chosen sets of conserved charges generated by the transfer matrix. In the model at hand the left and right SoV bases following the construction given in [1] can be written in terms of the transfer matrix $T_1^{(K)}(\lambda)$ and its fused transfer matrix $T_2^{(K)}(\lambda)$. An important feature of these SoV bases is that they are not orthogonal to each other for generic twist matrix $K$ having simple spectrum. The first key result of the present paper is the computation of the matrix of scalar products between these right and left SoV bases as stated in Theorem 3.1.

Theorem 3.1 also shows that whenever the twist matrix $K$ has simple spectrum and zero determinant, the chosen left and right SoV bases are orthogonal to each others since the off-diagonal elements of the matrix of scalar products are all proportional to some strictly positive power of the determinant of $K$. Moreover, in that case, we have been able to give a direct proof of this result simply using the simplified fusion relations resulting from the vanishing of the corresponding quantum determinant. As a consequence, it leads to very simple formulae for the scalar products of the so-called separate states. In that case they are just given by products of determinants which are similar to the ones of $\mathcal{Y}(gl_2)$ type.

This observation leads us to consider the generalization of these features for the case of a generic twist matrix $K$ having simple spectrum and non zero determinant. This amounts to define new SoV bases constructed from different sets of conserved charges with respect to the one's so far considered. We have shown that such sets of conserved charges indeed exists and we have characterized them using their generating functional $\mathbb{T}_j^{(K)}(\lambda)$ for $j = 1, 2$ defined in (4.93). By using them, we have determined new left and right SoV bases that are indeed orthogonal to each other, leading to simple scalar products formula for their separate states, and in particular for the scalar products of separate states with transfer matrix eigenstates. They are given as products of $\mathcal{Y}(gl_2)$ type determinants. This paves the way for their use in computing form factors and possibly even correlation functions of local operators. For this we will need to be able to write the resolution of the quantum inverse scattering problem in a form suitable to act in a simple manner on separate states. This question is now under study.

Another important question, with regards to the key results obtained in the present paper is how to determine in general sets of charges having properties similar to the one determined in (4.93). One possible route to this could be to construct explicitly the similarity transformation between the operator families $\mathbb{T}_j^{(K)}(\lambda)$ and $T_j^{(\hat{K})}(\lambda)$. In a future publication, we plan to show for example how to compute

$$\langle t_a^{(K)} | t_b^{(\hat{K})} \rangle, \quad \forall a, b \in \{1, .., 3^N\}, \tag{5.1}$$

which just define the matrix elements of the similarity transformation from $\mathbb{T}_1^{(K)}(\lambda)$ to $T_1^{(\hat{K})}(\lambda)$. This seems accessible thanks to the scalar products analyzed in Theorem 3.1. Another important open problem, deserving further analysis, is the possibility to find a direct construction of the new family of conserved charges $\mathbb{T}_j^{(K)}(\lambda)$ in terms of the original transfer matrix $T_1^{(K)}(\lambda)$ or the associated known family of commuting operators like the $T_2^{(K)}(\lambda)$ and the Baxter $Q$-operators for the general invertible twist $K$. More generally, the purely algebraic construction of a family satisfying the same simplified form of the fusion equations, like those written in (4.97)-(4.98), is one of our future goals.

## Acknowledgements

J.M.M. and G.N. are supported by CNRS and ENS de Lyon. L.V. is supported by Ecole Polytechnique and ENS de Lyon.

## A Explicit tensor product form of SoV starting co-vector/vector

Here, we want to prove the statements of the Proposition 3.1 about the fact that given the co-vector $\langle \mathbf{1}|$ of tensor product type then we can write explicitly the vector $|\mathbf{0}\rangle$ and it has a tensor product form too according to (3.9) and (3.10).

Let us start proving the following general property, that we state for the $gl_3$ case but that indeed can be extended to the $gl_n$ cases as well for rational $R$-matrices:

**Proposition A.1.** *Let $K$ be a $3 \times 3$ matrix, then we have the following explicit formula for the product of transfer matrices:*

$$\prod_{j=1}^{\mathsf{M}} T_1^{(K)}(\xi_{a_j}) = n_{a_1,...,a_\mathsf{M}} R_{a_1;1,...,a_1-1} \, \hat{R}_{a_2;1,...,a_2-1}^{(a_1)} \cdots \hat{R}_{a_{j+1},1,...,a_{j+1}-1}^{(a_1,...,a_j)} \cdots \hat{R}_{a_\mathsf{M};1,...,a_\mathsf{M}-1}^{(a_1,...,a_{\mathsf{M}-1})}$$

$$\times \bigotimes_{j=1}^{\mathsf{M}} K_{a_j} \hat{R}_{a_1;a_1+1,...,\mathsf{N}}^{(a_2,...,a_\mathsf{M})} \cdots \hat{R}_{a_j;a_j+1,...,\mathsf{N}}^{(a_{j+1},...,a_\mathsf{M})} \cdots \hat{R}_{a_{\mathsf{M}-1};a_{\mathsf{M}-1}+1,...,\mathsf{N}}^{(a_\mathsf{M})} R_{a_\mathsf{M};a_\mathsf{M}+1,...,\mathsf{N}}, \quad \text{(A.1)}$$

*where we have taken $a_1 < a_2 < \cdots < a_{\mathsf{M}-1} < a_\mathsf{M}$ and $\mathsf{M} \leq \mathsf{N}$ and we have used the notation:*

$$R_{a;b_1,...,b_M} = R_{ab_M}(\xi_a - \xi_{b_M}) \cdots R_{ab_1}(\xi_a - \xi_{b_1}), \quad \text{(A.2)}$$

*while $\hat{R}_{a;b_1,...,b_M}^{(b_{a_1},...,b_{a_k})}$ denotes the same product of R-matrices however with the factors $R_{ab_{a_1}}$ up to $R_{ab_{a_k}}$ omitted and $n_{a_1,...,a_\mathsf{M}} = \prod_{i<j} n_{a_i,a_j}$, with $n_{a_i,a_j} = \eta^2 - (\xi_{a_i} - \xi_{a_j})^2$. Then, for any choice of $1 \leq h_{a_j} \leq 2$ we have:*

$$\langle \mathbf{0}| \prod_{j=1}^{\mathsf{M}} T_1^{(K)}(\xi_{a_j})^{h_{a_j}} = \sum_{r_1 \in A_1,...,r_\mathsf{M} \in A_\mathsf{M}} \sum_{s_1 \in B_1,...,s_\mathsf{M} \in B_\mathsf{M}} C_{r_1,...,r_\mathsf{M},s_1,...,s_\mathsf{M}}$$

$$\times V(r_1,...,r_\mathsf{M}) V(s_1,...,s_\mathsf{M}) \langle \mathbf{0}| \bigotimes_{j=1}^{\mathsf{M}} K_{r_j} \bigotimes_{j=1}^{\mathsf{M}} K_{s_j}^{h_j-1}, \quad \text{(A.3)}$$

*where we take the following tensor product form for the co-vector:*

$$\langle \mathbf{0}| = \bigotimes_{a=1}^{\mathsf{N}} \langle 0,a|. \quad \text{(A.4)}$$

*$V(x_1,...,x_\mathsf{M})$ is the Vandermonde determinant, and $C_{r_1,...,r_\mathsf{M},s_1,...,s_\mathsf{M}}$ are some finite non-zero coefficients. We have defined:*

$$A_j = \{a_j,...,\mathsf{N}\} \cup \begin{cases} \{1,...,a_j-1\} & \text{if } h_{a_j} = 2, \\ \emptyset & \text{if } h_{a_j} = 1, \end{cases} \quad \text{(A.5)}$$

$$B_j = a_j \cup \begin{cases} \{a_j+1,...,\mathsf{N}\} & \text{if } h_{a_j} = 2, \\ \emptyset & \text{if } h_{a_j} = 1. \end{cases} \quad \text{(A.6)}$$

*Proof.* Let us consider the following product:

$$R_{a;a+1,\dots,N}R_{a+l;1,\dots,a+l-1} = R_{a;a+2,\dots,N}R_{a+l;a+2,\dots,a+l-1}R_{a,a+1}R_{a+l,a+1}R_{a+l,a}R_{a+l;1,\dots,a-1} \tag{A.7}$$

$$= R_{a;a+2,\dots,N}R_{a+l;a+2,\dots,a+l-1}R_{a+l,a}R_{a+l,a+1}R_{a,a+1}R_{a+l;1,\dots,a-1} \tag{A.8}$$

$$= R_{a;a+3,\dots,N}R_{a+l;a+3,\dots,a+l-1}R_{a,a+2}R_{a+l,a+2}R_{a+l,a}$$
$$\times R_{a+l,a+1}R_{a+l;1,\dots,a-1}R_{a,a+1} \tag{A.9}$$

$$= R_{a;a+3,\dots,N}R_{a+l;a+3,\dots,a+l-1}R_{a+l,a}R_{a+l,a+2}R_{a+l,a+1}$$
$$\times R_{a+l;1,\dots,a-1}R_{a,a+2}R_{a,a+1}, \tag{A.10}$$

where we have used the commutativity of $R$-matrices on different spaces and the Yang-Baxter equation. So, by iterating it, we get:

$$R_{a;a+1,\dots,N}R_{a+l;1,\dots,a+l-1} = n_{a,a+l}\hat{R}^{(a)}_{a+l;1,\dots,a+l-1}\hat{R}^{(a+l)}_{a;a+1,\dots,N}, \tag{A.11}$$

once we use that:

$$R_{a,a+l}R_{a+l,a} = n_{a,a+l}. \tag{A.12}$$

From this identity we get:

$$T_1^{(K)}(\xi_a)T_1^{(K)}(\xi_{a+l}) = n_{a,a+l}R_{a;1,\dots,a-1}\hat{R}^{(a)}_{a+l;1,\dots,a+l-1}K_a\bigotimes K_{a+l}\hat{R}^{(a+l)}_{a;a+1,\dots,N}R_{a+l;a+l+1,\dots,N}, \tag{A.13}$$

from which we easily obtain our statement (A.1) in the case $M = 2$. The general case is proven by induction on $M$. To get the $M + 1$ case knowing that the formula (A.1) is satisfied for $M$, we need to prove the following equality for $a_{l-1} < a_l < \cdots < a_k < a_{k+1}$:

$$\hat{R}^{(a_l,\dots,a_k)}_{a_{l-1};a_{l-1}+1,\dots,N}\hat{R}^{(a_l,\dots,a_k)}_{a_{k+1};1,\dots,a_{k+1}-1} = n_{a_{k+1},a_{l-1}}\hat{R}^{(a_{l-1},\dots,a_k)}_{a_{k+1};1,\dots,a_{k+1}-1}\hat{R}^{(a_l,\dots,a_{k+1})}_{a_{l-1};a_{l-1}+1,\dots,N}. \tag{A.14}$$

We have the following chain of equalities using the Yang-Baxter commutation relations, then the unitarity relation for the $R$-matrix and in the last step the fact that two $R$-matrices acting on different spaces commute:

$$\hat{R}^{(a_l,\dots,a_k)}_{a_{l-1};a_{l-1}+1,\dots,N}\hat{R}^{(a_l,\dots,a_k)}_{a_{k+1};1,\dots,a_{k+1}-1} =$$
$$= \hat{R}^{(a_l,\dots,a_k)}_{a_{l-1};a_{k+1}+1,\dots,N}R_{a_{l-1}a_{k+1}}\hat{R}^{(a_l,\dots,a_k)}_{a_{l-1};a_{l-1}+1,\dots,a_{k+1}-1}\hat{R}^{(a_l,\dots,a_k)}_{a_{k+1};a_{l-1}+1,\dots,a_{k+1}-1}R_{a_{k+1}a_{l-1}}\hat{R}^{(a_l,\dots,a_k)}_{a_{k+1};1,\dots,a_{l-1}-1}$$
$$= \hat{R}^{(a_l,\dots,a_k)}_{a_{l-1};a_{k+1}+1,\dots,N}\hat{R}^{(a_l,\dots,a_k)}_{a_{k+1};a_{l-1}+1,\dots,a_{k+1}-1}\hat{R}^{(a_l,\dots,a_k)}_{a_{l-1};a_{l-1}+1,\dots,a_{k+1}-1}R_{a_{l-1}a_{k+1}}R_{a_{k+1}a_{l-1}}\hat{R}^{(a_l,\dots,a_k)}_{a_{k+1};1,\dots,a_{l-1}-1}$$
$$= n_{a_{k+1},a_{l-1}}\hat{R}^{(a_l,\dots,a_k)}_{a_{k+1};a_{l-1}+1,\dots,a_{k+1}-1}\hat{R}^{(a_l,\dots,a_k)}_{a_{k+1};1,\dots,a_{l-1}-1}\hat{R}^{(a_l,\dots,a_k)}_{a_{l-1};a_{k+1}+1,\dots,N}\hat{R}^{(a_l,\dots,a_k)}_{a_{l-1};a_{l-1}+1,\dots,a_{k+1}-1}$$
$$= n_{a_{k+1},a_{l-1}}\hat{R}^{(a_{l-1},\dots,a_k)}_{a_{k+1};1,\dots,a_{k+1}-1}\hat{R}^{(a_l,\dots,a_{k+1})}_{a_{l-1};a_{l-1}+1,\dots,N}. \tag{A.15}$$

et us prove the induction going from $M$ to $M + 1$. We have:

$$\prod_{j=1}^{M+1}T_1^{(K)}(\xi_{a_j}) = \prod_{j=1}^{M}T_1^{(K)}(\xi_{a_j})\,T_1^{(K)}(\xi_{a_{M+1}})$$
$$= n_{a_1,\dots,a_M}R_{a_1;1,\dots,a_1-1}\hat{R}^{(a_1)}_{a_2;1,\dots,a_2-1}\cdots\hat{R}^{(a_1,\dots,a_j)}_{a_{j+1};1,\dots,a_{j+1}-1}\cdots\hat{R}^{(a_1,\dots,a_{M-1})}_{a_M;1,\dots,a_M-1} \tag{A.16}$$
$$\times\bigotimes_{j=1}^{M}K_{a_j}\hat{R}^{(a_2,\dots,a_M)}_{a_1;a_1+1,\dots,N}\cdots\hat{R}^{(a_{j+1},\dots,a_M)}_{a_j;a_j+1,\dots,N}\cdots\hat{R}^{(a_M)}_{a_{M-1};a_{M-1}+1,\dots,N}R_{a_M;a_M+1,\dots,N}$$
$$\times R_{a_{M+1};1,\dots,a_{M+1}-1}K_{a_{M+1}}R_{a_{M+1};a_{M+1}+1,\dots,N}.$$

Then, keeping the last factor as it is and moving the term $R_{a_{M+1};1,\dots,a_{M+1}-1}$ to the left using the above proven exchange relation (A.14) successively, and then moving $K_{a_{M+1}}$ freely (there is no

object acting in the same space) to the left until it will join the products of other matrices $K$, we get the desired result.

We have to use now that $\langle\underline{0}|$ is an eigenco-vector for a generic product of rational $R$-matrices acting on the local quantum spaces, hence:

$$\langle\underline{0}|R_{a_1;1,...,a_1-1}\hat{R}^{(a_1)}_{a_2;1,...,a_2-1}\cdots\hat{R}^{(a_1,...,a_j)}_{a_{j+1};1,...,a_{j+1}-1}\cdots\hat{R}^{(a_1,...,a_{M-1})}_{a_M;1,...,a_M-1}=m_{a_1,...,a_M}\langle\underline{0}|, \tag{A.17}$$

with $m_{a_1,...,a_M}$ some calculable non-zero coefficient. Using the explicit formula for the $R$-matrix, this implies the following identity:

$$\langle\underline{0}|\prod_{j=1}^{M}T_1^{(K)}(\xi_{a_j})=n_{a_1,...,a_M}m_{a_1,...,a_M}\langle\underline{0}|\bigotimes_{j=1}^{M}K_{a_j}\hat{R}^{(a_2,...,a_M)}_{a_1;a_1+1,...,N}\cdots\hat{R}^{(a_{j+1},...,a_M)}_{a_j;a_j+1,...,N}\cdots$$
$$\cdots\hat{R}^{(a_M)}_{a_{M-1};a_{M-1}+1,...,N}R_{a_M;a_M+1,...,N}, \tag{A.18}$$

and so:

$$\langle\underline{0}|\prod_{j=1}^{M}T_1^{(K)}(\xi_{a_j})=\sum_{r_1\in\{a_1+1,...,N\},...,r_M\in\{a_M+1,...,N\}}C_{r_1,...,r_M}V(r_1,...,r_M)\langle\underline{0}|\bigotimes_{j=1}^{M}K_{r_j}. \tag{A.19}$$

Applying once again this formula, we get our second statement. $\qquad\square$

The following lemma holds for a general simple $K$ matrix.

**Lemma A.1.** *Let $K$ be a $3\times3$ w-simple matrix, then if we chose the tensor product form:*

$$\langle\underline{1}|=\left(\bigotimes_{a=1}^{N}\langle1,a|\right)\Gamma_W^{-1}, \quad \Gamma_W=\bigotimes_{a=1}^{N}W_{K,a}, \tag{A.20}$$

*we have that the vector $|\underline{0}\rangle$ defined in (3.5) has the tensor product form:*

$$|\underline{0}\rangle=\Gamma_W\bigotimes_{a=1}^{N}|0,a\rangle, \tag{A.21}$$

*where $|0,a\rangle$ has the form (3.10) and it satisfies the following local properties*

$$\langle1,a|\tilde{K}_J^{(a)}|0,a\rangle=1/\text{q-det}\,M^{(I)}(\xi_a), \tag{A.22}$$
$$\langle1,a|(K_J^{(a)})^h|0,a\rangle=0, \quad for\ h=0,1, \tag{A.23}$$

*where $\tilde{K}_J$ is the adjoint matrix of $K_J$:*

$$\tilde{K}_J K_J=K_J\tilde{K}_J=\det K. \tag{A.24}$$

*Proof.* Let us take the following normalization for the SoV co-vector basis:

$$\langle h_1,...,h_N|=\langle\underline{0}|\prod_{n=1}^{N}\frac{T_1^{(K)}(\xi_n)^{h_n}}{\text{q-det}M^{(K)}(\xi_n)^{(1-\delta_{h_n,0})}}, \tag{A.25}$$

where we have defined:

$$\langle\underline{0}|=\langle\underline{1}|\prod_{a=1}^{N}\text{q-det}M^{(I)}(\xi_a)\bigotimes_{a=1}^{N}\tilde{K}^{(a)}. \tag{A.26}$$

Now we can use the previous lemma to get the following statement:

$$\langle h_1,...,h_N| = \sum_{k_1,...,k_N=0,1,2} c^{k_1,...,k_N}_{h_1,...,h_N} \langle \underline{0}| \bigotimes_{a=1}^{N} K^{(a)k_a}, \tag{A.27}$$

with

$$c^{k_1=0,...,k_N=0}_{h_1,...,h_N} = 0 \ \text{ if } \exists j \in \{1,...,N\} : h_j \neq 0. \tag{A.28}$$

By definition:

$$\langle \underline{0}| \bigotimes_{a=1}^{N} K^{(a)k_a}|\underline{0}\rangle = \prod_{a=1}^{N} \langle 0,a|K_J^{(a)k_a}|0,a\rangle = 0 \ \text{ if } \exists a \in \{1,...,N\} : k_a \neq 0, \tag{A.29}$$

so we get:

$$\langle h_1,...,h_N|\underline{0}\rangle = 0 \ \text{ if } \exists j \in \{1,...,N\} : h_j \neq 0. \tag{A.30}$$

The fact that:

$$\langle \underline{0}|\underline{0}\rangle = \prod_{a=1}^{N} \langle 0,a|0,a\rangle = 1 \tag{A.31}$$

is proven by direct computations.

Finally, let us observe that the following identities:

$$\langle \underline{0}| \prod_{n=1}^{N} \frac{\left(T_1^{(K)}(\xi_n)\right)^{h_n}}{\left(q\text{-det}M^{(K)}(\xi_n)\right)^{1-\delta_{h_n,0}}} = \langle \underline{1}| \prod_{n=1}^{N} T_2^{(K)\delta_{h_n,0}}(\xi_n - \eta)T_1^{(K)\delta_{h_n,2}}(\xi_n), \tag{A.32}$$

holds for any $h_n \in \{0,1,2\}$. Now, in the limit $\det K \to 0$, keeping $K$ a $3 \times 3$ w-simple matrix[16], we have that the r.h.s. of the equation (A.32) is well defined and it defines the limit of the l.h.s., so that our co-vector SoV basis goes back to the one defined in the case $\det K = 0$. Moreover, the $|0,a\rangle$ are well defined and so the $|\underline{0}\rangle$ above defined in this limit still satisfies (3.5). $\qquad\square$

# B  Orthogonal co-vector/vector SoV basis for $gl_2$ representations

Here, we consider the fundamental representations of the $gl_2$ Yang-Baxter algebra associated to generic quasi-periodic boundary conditions, with transfer matrix:

$$T^{(K)}(\lambda) \equiv \text{tr}_{V_a} K_a R_{a,N}(\lambda - \xi_N)\cdots R_{a,1}(\lambda - \xi_1) \in \text{End}(\mathcal{H}), \tag{B.1}$$

where $\mathcal{H}$ is the quantum space of the representation, $R_{a,b}(\lambda) \in \text{End}(V_a \otimes V_b)$, $V_a \simeq \mathbb{C}^2$, $V_b \simeq \mathbb{C}^2$ is the rational 6-vertex R-matrix solution of the Yang-Baxter equation and the twist matrix reads

$$K = \begin{pmatrix} a & b \\ c & d \end{pmatrix} \in \text{End}(\mathbb{C}^2). \tag{B.2}$$

The construction of the orthogonal co-vector and vector SoV bases for these $gl_2$ representations is here implemented to define a reference to compare with for the more involved constructions that we have considered in this paper for $gl_3$ representations. One should mention that up similarity transformations[17] the SoV bases in these $gl_2$ cases are already available in the literature using the framework of the traditional Sklyanin's SoV construction, see for

---

[16]That is according to the three cases considered in the Theorem 3.1.

[17]As discussed in section 3.4 of [1]

example [40] for the antiperiodic case and [55] for more general twists. However, here we are interested in implementing these constructions entirely inside our new approach [1].

The following proposition allows to produce the orthogonal basis to the left SoV basis

$$\langle h_1, ..., h_{\mathsf{N}}| \equiv \langle \mathbf{0}| \prod_{a=1}^{\mathsf{N}} \left( \frac{T^{(K)}(\xi_a)}{a(\xi_a)} \right)^{h_a} \quad \text{for any } \{h_1, ..., h_{\mathsf{N}}\} \in \{0,1\}^{\mathsf{N}}, \tag{B.3}$$

and to show that itself is of SoV type just using the polynomial form of the transfer matrix and the fusion equations.

Let us denote with $|\mathbf{0}\rangle$ the non-zero vector orthogonal to all the SoV co-vectors with the exception of $\langle \mathbf{0}|$, i.e.

$$\langle h_1, ..., h_{\mathsf{N}}|\mathbf{0}\rangle = \frac{\prod_{n=1}^{\mathsf{N}} \delta_{h_n,0}}{V^2(\xi_1, ..., \xi_{\mathsf{N}})}, \quad \forall \{h_1, ..., h_{\mathsf{N}}\} \in \{0,1\}^{\mathsf{N}}, \tag{B.4}$$

with $\langle h_1, \ldots, h_{\mathsf{N}}|$ the set of SoV co-vectors a basis. $|\mathbf{0}\rangle$ is uniquely defined with the above normalization. Similarly, we can introduce the non-zero vector $|\mathbf{1}\rangle$ orthogonal to all the SoV co-vectors with the exception of $\langle 1, ..., 1|$, i.e.

$$\langle h_1, ..., h_{\mathsf{N}}|\mathbf{1}\rangle = \frac{\prod_{n=1}^{\mathsf{N}} \delta_{h_n,1}}{V(\xi_1, ..., \xi_{\mathsf{N}})V(\xi_1^{(1)}, ..., \xi_{\mathsf{N}}^{(1)})}, \quad \forall \{h_1, ..., h_{\mathsf{N}}\} \in \{0,1\}^{\mathsf{N}}, \tag{B.5}$$

which also fixes the normalization of $|\mathbf{1}\rangle$.

**Proposition B.1.** *Under the same conditions assuring that the set of SoV co-vectors is a basis (i.e. almost any choice of $\langle \mathbf{0}|$, $K \neq xI$, for any $x \in \mathbb{C}$, and the condition (2.6)), then the following set of vectors:*

$$|h_1, ..., h_{\mathsf{N}}\rangle = \prod_{a=1}^{\mathsf{N}} \left( \frac{T^{(K)}(\xi_a - \eta)}{a(\xi_a)} \right)^{1-h_a} |\mathbf{1}\rangle, \quad \forall \{h_1, ..., h_{\mathsf{N}}\} \in \{0,1\}^{\mathsf{N}} \tag{B.6}$$

*forms an orthogonal basis to the left SoV basis:*

$$\langle h_1, ..., h_{\mathsf{N}}|k_1, ..., k_{\mathsf{N}}\rangle = \frac{\prod_{n=1}^{\mathsf{N}} \delta_{h_n,k_n}}{V(\xi_1, ..., \xi_{\mathsf{N}})V(\xi_1^{(h_1)}, ..., \xi_{\mathsf{N}}^{(h_{\mathsf{N}})})}. \tag{B.7}$$

*Let $t(\lambda)$ be an element of the spectrum of $T^{(K)}(\lambda)$, then the uniquely defined eigenvector $|t\rangle$ and eigenco-vector $\langle t|$ admit the following SoV representations:*

$$|t\rangle = \sum_{h_1,...,h_{\mathsf{N}}=0}^{1} \prod_{a=1}^{\mathsf{N}} (\frac{t(\xi_a)}{a(\xi_a)})^{h_a} V(\xi_1^{(h_1)}, ..., \xi_{\mathsf{N}}^{(h_{\mathsf{N}})})|h_1, ..., h_{\mathsf{N}}\rangle, \tag{B.8}$$

$$\langle t| = \sum_{h_1,...,h_{\mathsf{N}}=0}^{1} \prod_{a=1}^{\mathsf{N}} (\frac{t(\xi_a - \eta)}{a(\xi_a)})^{1-h_a} V(\xi_1^{(h_1)}, ..., \xi_{\mathsf{N}}^{(h_{\mathsf{N}})})\langle h_1, ..., h_{\mathsf{N}}|, \tag{B.9}$$

*where we have fixed their normalization by imposing:*

$$\langle \mathbf{0}|t\rangle = \langle t|\mathbf{1}\rangle = 1/V(\xi_1, ..., \xi_{\mathsf{N}}). \tag{B.10}$$

*Proof.* Let us start proving the orthogonality condition:

$$\langle h_1, ..., h_{\mathsf{N}}|k_1, ..., k_{\mathsf{N}}\rangle = 0, \quad \forall \{k_1, ..., k_{\mathsf{N}}\} \neq \{h_1, ..., h_{\mathsf{N}}\} \in \{0,1\}^{\mathsf{N}}. \tag{B.11}$$

The proof is done by induction, assuming that it is true for any vector $|k_1, ..., k_N\rangle$ with $\sum_{n=1}^{N} k_n = N - l$ for $l \leq N - 1$ and proving it for vectors $|k'_1, ..., k'_N\rangle$ with $\sum_{n=1}^{N} k'_n = N - (l+1)$. To this aim we fix a vector $|k_1, ..., k_N\rangle$ with $\sum_{n=1}^{N} k_n = N - l$ and we denote with $\pi$ a permutation on the set $\{1, ..., N\}$ such that:

$$k_{\pi(a)} = 0 \text{ for } a \leq l \text{ and } k_{\pi(a)} = 1 \text{ for } l < a, \tag{B.12}$$

then we compute:

$$\langle h_1, ..., h_N | T^{(K)}(\xi^{(1)}_{\pi(l+1)}) | k_1, ..., k_N \rangle = a(\xi_{\pi(l+1)}) \langle h_1, ..., h_N | k'_1, ..., k'_N \rangle, \tag{B.13}$$

where we have defined:

$$k'_{\pi(a)} = k_{\pi(a)}, \ \forall a \in \{1, ..., N\} \backslash \{l+1\} \text{ and } k'_{\pi(l+1)} = 0, \tag{B.14}$$

for any $\{h_1, ..., h_N\} \neq \{k'_1, ..., k'_N\} \in \{0, 1\}^N$. There are two cases. The first case is $h_{\pi(l+1)} = 1$, then it holds:

$$\langle h_1, ..., h_N | T^{(K)}(\xi^{(1)}_{\pi(l+1)}) | k_1, ..., k_N \rangle = \frac{q\text{-det} M^{(K)}(\xi_{\pi(l+1)})}{a(\xi_{\pi(l+1)})} \langle h'_1, ..., h'_N | k_1, ..., k_N \rangle, \tag{B.15}$$

where we have defined:

$$h'_{\pi(a)} = h_{\pi(a)}, \ \forall a \in \{1, ..., N\} \backslash \{l+1\} \text{ and } h'_{\pi(l+1)} = 0. \tag{B.16}$$

Then from $\{h_1, ..., h_N\} \neq \{k'_1, ..., k'_N\} \in \{0, 1\}^N$ it follows also that $\{h'_1, ..., h'_N\} \neq \{k_1, ..., k_N\} \in \{0, 1\}^N$ and so the induction implies that the r.h.s. of (B.15) is zero and so we get:

$$\langle h_1, ..., h_N | k'_1, ..., k'_N \rangle = 0. \tag{B.17}$$

The second case is $h_{\pi(l+1)} = 0$. We can use the following interpolation formula:

$$T^{(K)}(\xi^{(1)}_{\pi(l+1)}) = (\text{tr} K) \prod_{a=1}^{N} (\xi^{(1)}_{\pi(l+1)} - \xi^{(h_{\pi(a)})}_{\pi(a)}) + \sum_{a=1}^{N} \prod_{b \neq a, b=1}^{N} \frac{\xi^{(1)}_{\pi(l+1)} - \xi^{(h_{\pi(b)})}_{\pi(b)}}{\xi^{(h_{\pi(a)})}_{\pi(a)} - \xi^{(h_{\pi(b)})}_{\pi(b)}} T^{(K)}\left(\xi^{(h_{\pi(a)})}_{\pi(a)}\right), \tag{B.18}$$

from which $\langle h_1, ..., h_N | T^{(K)}(\xi^{(1)}_{\pi(l+1)}, \{\xi\}) | k_1, ..., k_N \rangle$ reduces to the following sum:

$$(\text{tr} K) \prod_{a=1}^{N} (\xi^{(1)}_{\pi(l+1)} - \xi^{(h_{\pi(a)})}_{\pi(a)}) \langle h_1, ..., h_N | k_1, ..., k_N \rangle + \sum_{a=1}^{N} \prod_{b \neq a, b=1}^{N} \frac{\xi^{(1)}_{\pi(l+1)} - \xi^{(h_{\pi(b)})}_{\pi(b)}}{\xi^{(h_{\pi(a)})}_{\pi(a)} - \xi^{(h_{\pi(b)})}_{\pi(b)}}$$

$$\times \frac{(q\text{-det} M^{(K)}(\xi_{\pi(a)}))^{h_{\pi(a)}}}{\left(a(\xi_{\pi(a)})\right)^{2h_{\pi(a)} - 1}} \langle h_1^{(a)}, ..., h_N^{(a)} | k_1, ..., k_N \rangle, \tag{B.19}$$

where we have defined:

$$h^{(a)}_{\pi(j)} = h_{\pi(j)}, \ \forall j \in \{1, ..., N\} \backslash \{a\} \text{ and } h^{(a)}_{\pi(a)} = 1 - h_{\pi(a)}. \tag{B.20}$$

Let us now note that from $h_{\pi(l+1)} = 0$ it follows that $\{h_1, ..., h_N\} \neq \{k_1, ..., k_N\}$, as $k_{\pi(l+1)} = 1$ by definition and similarly $\{h_1^{(a)}, ..., h_N^{(a)}\} \neq \{k_1, ..., k_N\}$ being by definition $h^{(a)}_{\pi(l+1)} = h_{\pi(l+1)} = 0$ for any $a \in \{1, ..., N\} \backslash \{l+1\}$. Finally, from $\{h_1, ..., h_N\} \neq \{k'_1, ..., k'_N\}$ with $h_{\pi(l+1)} = k'_{\pi(l+1)} = 0$, clearly it follows that $\{h_1^{(l+1)}, ..., h_N^{(l+1)}\} \neq \{k_1, ..., k_N\}$. So, by using the induction argument,

we get that any term in the above sum is zero. So that also in the case $h_{\pi(l+1)} = 0$, we get that (B.17) is satisfied, and so it is satisfied for any $\{h_1, ..., h_N\} \neq \{k'_1, ..., k'_N\}$ which proves the induction of the orthogonality to $l + 1$. Indeed, by changing the permutation $\pi$ we can both take for $\{\pi(1), ..., \pi(l)\}$ any subset of cardinality $l$ in $\{1, ..., N\}$ and with $\pi(l + 1)$ any element in its complement $\{1, ..., N\} \backslash \{\pi(1), ..., \pi(l)\}$.

We can compute now the left/right normalization, and to do this we just need to compute the following type of ratio:

$$\frac{\langle h_1^{(a)}, ..., h_N^{(a)} | h_1^{(a)}, ..., h_N^{(a)} \rangle}{\langle \bar{h}_1^{(a)}, ..., \bar{h}_N^{(a)} | \bar{h}_1^{(a)}, ..., \bar{h}_N^{(a)} \rangle} = a(\xi_a) \frac{\langle h_1^{(a)}, ..., h_N^{(a)} | h_1^{(a)}, ..., h_N^{(a)} \rangle}{\langle \bar{h}_1^{(a)}, ..., \bar{h}_N^{(a)} | T^{(K)}(\xi_a^{(1)}) | h_1^{(a)}, ..., h_N^{(a)} \rangle}, \tag{B.21}$$

with $\bar{h}_j^{(a)} = h_j^{(a)}$ for any $j \in \{1, ..., N\} \backslash \{a\}$ while $\bar{h}_j^{(a)} = 0$ and $h_j^{(a)} = 1$. We can use now once again the interpolation formula (B.18) which by the orthogonality condition produces only one non-zero term, the one associate to $T^{(K)}(\xi_a, \{\xi\})$. It holds:

$$\frac{\langle h_1^{(a)}, ..., h_N^{(a)} | h_1^{(a)}, ..., h_N^{(a)} \rangle}{\langle \bar{h}_1^{(a)}, ..., \bar{h}_N^{(a)} | \bar{h}_1^{(a)}, ..., \bar{h}_N^{(a)} \rangle} = \prod_{b \neq a, b=1}^{N} \frac{\xi_a - \xi_b^{(h_b)}}{\xi_a^{(1)} - \xi_b^{(h_b)}}. \tag{B.22}$$

Using the above result, it is now standard to get the proof of the Vandermonde determinant form for the normalization.

Let us note that being the set of SoV co-vectors and vectors basis in $\mathcal{H}$, it follows that for any transfer matrix eigenstates $|t\rangle$ and $\langle t|$ there exists at least a $\{r_1, ..., r_N\} \in \{0, 1\}^N$ and a $\{s_1, ..., s_N\} \in \{0, 1\}^N$ such that:

$$\langle r_1, ..., r_N | t \rangle \neq 0, \quad \langle t | s_1, ..., s_N \rangle \neq 0, \tag{B.23}$$

which together with the identities:

$$\langle h_1, ..., h_N | t \rangle \propto \langle \underline{0} | t \rangle, \quad \langle t | h_1, ..., h_N \rangle \propto \langle t | \underline{1} \rangle, \quad \forall \{h_1, ..., h_N\} \in \{0, 1\}^N, \tag{B.24}$$

imply that:

$$\langle \underline{0} | t \rangle \neq 0, \quad \langle t | \underline{1} \rangle \neq 0. \tag{B.25}$$

So we are free to fix the normalization of $|t\rangle$ and $\langle t|$ by (B.10). Finally, the representations for these eigenco-vectors and eigenvectors follow from the use of the SoV decomposition of the identity:

$$\mathbb{I} = V(\{\xi\}) \sum_{h_1, ..., h_N = 0}^{1} V(\xi_1^{(h_1)}, ..., \xi_N^{(h_N)}) | h_1, ..., h_N \rangle \langle h_1, ..., h_N |. \tag{B.26}$$

$$\square$$

**Corollary B.1.** *Let us assume that the condition (2.6) is satisfied and that $K \neq xI$, for any $x \in \mathbb{C}$, and furthermore $\det K \neq 0$, then the vectors of the right SoV basis admit also the following representations:*

$$|h_1, ..., h_N\rangle = \prod_{a=1}^{N} \left( \frac{T^{(K)}(\xi_a)}{\det K \, d(\xi_a^{(1)})} \right)^{h_a} |\underline{0}\rangle, \quad \forall \{h_1, ..., h_N\} \in \{0, 1\}^N, \tag{B.27}$$

*as well as for any element of the spectrum of $T^{(K)}(\lambda)$ the unique associated eigenco-vector $\langle t|$ admit the following SoV representations:*

$$\langle t| = N_t \sum_{h_1, ..., h_N = 0}^{1} \prod_{a=1}^{N} \left( \frac{t(\xi_a)}{\det K \, d(\xi_a^{(1)})} \right)^{h_a} V(\xi_1^{(h_1)}, ..., \xi_N^{(h_N)}) \langle h_1, ..., h_N |, \tag{B.28}$$

*once we fix the normalization by (B.10), where we have defined:*

$$N_t = \langle t|\underline{0}\rangle = \prod_{a=1}^{N} \frac{t(\xi_a^{(1)})}{a(\xi_a)} \neq 0. \tag{B.29}$$

*Proof.* Taking into account the chosen normalizations clearly it holds:

$$|\underline{0}\rangle = |h_1 = 0, ..., h_N = 0\rangle = \prod_{a=1}^{N} \frac{T^{(K)}(\xi_a^{(1)})}{a(\xi_a)}|\underline{1}\rangle, \tag{B.30}$$

*so that:*

$$\begin{aligned}
\prod_{a=1}^{N} \left( \frac{T^{(K)}(\xi_a)}{\det K d(\xi_a^{(1)})} \right)^{h_a} |\underline{0}\rangle &= \prod_{a=1}^{N} \left( \frac{T^{(K)}(\xi_a)}{\det K d(\xi_a^{(1)})} \right)^{h_a} \frac{T^{(K)}(\xi_a^{(1)})}{a(\xi_a)}|\underline{1}\rangle \\
&= \prod_{a=1}^{N} \left( \frac{T^{(K)}(\xi_a) T^{(K)}(\xi_a^{(1)})}{\det K d(\xi_a^{(1)}) a(\xi_a)} \right)^{h_a} \left( \frac{T^{(K)}(\xi_a^{(1)})}{a(\xi_a)} \right)^{1-h_a} |\underline{1}\rangle \\
&= |h_1, ..., h_N\rangle,
\end{aligned} \tag{B.31}$$

*by the quantum determinant identity. From this representation of the right SoV vectors it follows also that for any fixed left transfer matrix eigenstate $\langle t|$ it holds:*

$$\langle t|h_1, ..., h_N\rangle \propto \langle t|\underline{0}\rangle, \quad \forall\{h_1, ..., h_N\} \in \{0, 1\}^N, \tag{B.32}$$

*so that it must holds $\langle t|\underline{0}\rangle \neq 0$.* $\qquad\square$

As we have already shown in the previous appendix for $gl_3$ representations, also in $gl_2$ representations the tensor product forms hold.

**Corollary B.2.** *Let the inhomogeneity condition (2.6) be satisfied and $K \neq rI$, for any $r \in \mathbb{C}$, and let $(x, y) \in \mathbb{C}^2$ be such that:*

$$n_K(x, y) = bx^2 + (d - a)xy - cy^2 \neq 0. \tag{B.33}$$

*Then, once we define:*

$$\langle \underline{0}| = \bigotimes_{a=1}^{N} (x, y)_a, \tag{B.34}$$

*it holds:*

$$|\underline{1}\rangle = \frac{1}{n_1} \bigotimes_{n=1}^{N} \begin{pmatrix} -y \\ x \end{pmatrix}_n, \quad |\underline{0}\rangle = \frac{1}{n_0} \bigotimes_{n=1}^{N} \begin{pmatrix} bx + dy \\ -(ax + cy) \end{pmatrix}_n, \tag{B.35}$$

*where:*

$$n_1 = n_{1,...,N} \, n_K^N(x, y) \, V(\xi_1, ..., \xi_N) V(\xi_1^{(1)}, ..., \xi_N^{(1)}) \left( \prod_{n=1}^{N} a(\xi_n) \right)^{-1}, \tag{B.36}$$

$$n_0 = n_K^N(x, y) \, V^2(\xi_1, ..., \xi_N), \quad n_{1,...,N} = \prod_{1 \leq i < j \leq N} (\eta^2 - (\xi_i - \xi_j)^2). \tag{B.37}$$

# C    Proof of Theorem 3.1

This appendix is dedicated to the completion of the proof of the Theorem 3.1: here we prove the orthogonality properties and the non-zero coupling of the SoV co-vectors/vectors. It is worth remarking that the proof of the "pseudo-orthogonality" is quite intricate and we have divided it in several steps to make it more intelligible. The orthogonality conditions are established in section C.1, while section C.2 is dedicated to the proof of the form of the non-zero couplings of co-vectors/vectors.

The form of the orthogonality condition naturally leads to consider in the first instance vectors with $\underline{\mathbf{k}} \in \{0,2\}^{\mathsf{N}}$, this is achieved in subsection C.1.1. In this case, the co-vector/vector coupling is diagonal, i.e. standard orthogonality holds with non-zero coupling only for co-vector/vector associated to the same N-tuple $\underline{\mathbf{h}} = \underline{\mathbf{k}} \in \{0,2\}^{\mathsf{N}}$. This proof requires already different steps. We prove it first for the case with only one $k_a = 2$ while all the others being zero, and then by induction for the generic N-tuple $\underline{\mathbf{k}} \in \{0,2\}^{\mathsf{N}}$. In subsection C.1.2, we then consider the case with just one $k_a = 1$ while all the others $k_{b \neq a}$ being in $\{0,2\}$. Here, we prove that the standard orthogonality still works. In subsection C.1.3, we finally consider the proof for the case with non-diagonal and diagonal couplings, which correspond to SoV vectors associated to $\underline{\mathbf{k}}$ with at least one couple $(k_a = 1,\ k_b = 1)$. First, the case with just one couple $(k_a = 1,\ k_b = 1)$ is developed, and then the case of vectors associated to a general $\underline{\mathbf{k}} \in \{0,1,2\}^{\mathsf{N}}$.

In subsection C.2.1, we write the non-diagonal couplings in terms of the diagonal ones. In particular, we prove the formula (3.14) and its power dependence w.r.t. $\det K$. The coefficients $C_{\underline{\mathbf{h}}}^{\underline{\mathbf{k}}}$ in (3.14) are shown to be independent w.r.t. $\det K$ and completely characterized by the Lemma C.3 and by the solutions to the recursion equations derived in Lemma C.4. We do not resolve these recursions in general. Rather we argue the dependence of the coefficients in terms of the involved transfer matrix interpolation formulae and explicitly present them in the case of co-vectors having one couple of $(h_a = 0,\ h_b = 2)$ associated to vectors with a couple $(k_a = 1,\ k_b = 1)$. Finally, in subsection C.2.2, we prove the explicit form of the co-vector/vector diagonal coupling. The proof derived there does not use the fact that for $\det K = 0$ we have an independent derivation of the same SoV measure.

## C.1    Orthogonality proof

We use the following *incomplete*[18] notation for the interpolation formulae in the shifted inhomogeneities $\{\xi_n^{(h_n)}\}$ of the transfer matrix:

$$T_a^{(K)}(\lambda) \underset{\mathrm{UpC}}{=} \mathsf{t}_a + \sum_{n=1}^{\mathsf{N}} \mathsf{T}_a(\xi_n^{(h_n)}), \tag{C.1}$$

with

$$\mathsf{t}_a = \mathsf{a}^{\delta_{a,1}} \big( \mathsf{b}\, d(\lambda - \eta) \big)^{\delta_{a,2}} \prod_{a=1}^{\mathsf{N}} (\lambda - \xi_a^{(h_a)}), \quad \mathsf{T}_a(\xi_n^{(h_n)}) = T_a^{(K)}(\xi_n^{(h_n)}) d^{\delta_{a,2}} (\lambda - \eta) g_{n,\underline{\mathbf{h}}}^{(a)}(\lambda) \tag{C.2}$$

for $a \in \{1,2\}$, where

$$\mathsf{a} = \mathrm{tr}\, K, \quad \mathsf{b} = \frac{(\mathrm{tr}\, K)^2 - \mathrm{tr}(K^2)}{2}, \quad \mathsf{c} = \det K,$$

are the spectral invariants of the matrix $K$ and

$$\mathsf{c}_n = \mathsf{q}\text{-det}\, M^{(K)}(\xi_n) = \mathsf{c}\, \mathsf{q}\text{-det}\, M^{(I)}(\xi_n). \tag{C.3}$$

---

[18] We need only to keep partial information on the interpolation formulae for our current aims.

Note that this shorted notation hides the original value in which the transfer matrix was computed before the interpolation, which is $\lambda$ in (C.1). It also loses the coefficients of the same interpolation formulae. In the following of this appendix, all the equalities written down with symbol $\underset{\text{UpC}}{=}$ have to be interpreted up to these implicit, missing coefficients. This does not represent a problem, as here we are only interested in proofs that some matrix elements are zero or proportional to each other, which is something that remains true independently of the exact coefficients (as long as they do not vanish).

### C.1.1  First step: the case $\left|\underline{\mathbf{k}}\right\rangle$ with $\underline{\mathbf{k}} \in \{0,2\}^{\mathsf{N}}$

In the following, we introduce needed notations to implement operations on N-tuple of indices. Let us denote with $\underline{\mathbf{x}} = \{x_1, ..., x_{\mathsf{N}}\} \in \{0,1,2\}^{\mathsf{N}}$ a generic N-tuple from $\{0,1,2\}$, and with $(j_1, ..., j_m) \in \{0,1,2\}^m$ a generic $m$-tuple from $\{0,1,2\}$. We introduce the following notations:

$$\underline{\mathbf{x}}^{(j_1,...,j_m)}_{a_1,...,a_m} = \underline{\mathbf{x}} - \sum_{\substack{i=1 \\ i \neq r_1,...,r_l, 0 \leq l \leq m-1}}^{m} (x_{a_i} - j_i)\underline{\mathbf{e}}_{a_i} \quad \forall a_i \in \{1,...,\mathsf{N}\},\ i \leq m \leq \mathsf{N}, \qquad (\text{C.4})$$

where $\underline{\mathbf{e}}_a = \{\delta_{1,a}, ..., \delta_{\mathsf{N},a}\}$ and the $r_1, ..., r_l$ are defined as follows for any fixed choice of $a_1, ..., a_m$:

$$\forall h \leq l,\ r_h \in \{1,...,m\} : \exists s \in \{1,...,m\} \backslash \{r_1,...,r_l\},\ r_h < s \text{ with } a_{r_h} = a_s, \qquad (\text{C.5})$$

while it holds:

$$a_p \neq a_q, \quad \forall p \neq q \in \{1,...,m\} \backslash \{r_1,...,r_l\}. \qquad (\text{C.6})$$

In simple words, for any fixed $a_1, ..., a_m$, the $r_1, ..., r_l$ are defined as the minimal set of the smallest integers in $\{1,...,m\}$ such that removing them from $\{1,...,m\}$ make the condition (C.6) satisfied. Clearly, we have $l = 0$ if the $a_1, ..., a_m$ are all distinct.

**i) Only one $k_n = 2$**  Let us first prove:

$$\langle \underline{\mathbf{h}} | \underline{\mathbf{0}}^{(2)}_n \rangle = \langle \underline{\mathbf{h}} | \mathsf{T}_1(\xi_n) | \underline{\mathbf{0}} \rangle = 0, \quad \text{for } \underline{\mathbf{h}} \neq \underline{\mathbf{0}}^{(2)}_n. \qquad (\text{C.7})$$

If $h_n = 0, 1$ this statement is evident, since

$$\langle \underline{\mathbf{h}} | \mathsf{T}_1(\xi_n) | \underline{\mathbf{0}} \rangle = c_n^{\delta_{h_n,0}} \langle \underline{\mathbf{h}}^{(h_n+1)}_n | \underline{\mathbf{0}} \rangle = 0. \qquad (\text{C.8})$$

Now, let us fix $h_n = 2$. Here we proceed by induction, first assuming that all the others $h_{j \neq n} = 0, 1$:

$$\begin{aligned}
\langle \underline{\mathbf{h}} | \mathsf{T}_1(\xi_n) | \underline{\mathbf{0}} \rangle &\underset{\text{UpC}}{=} t_1 \langle \underline{\mathbf{h}} | \underline{\mathbf{0}} \rangle + \langle \underline{\mathbf{h}}^{(1)}_n | \mathsf{T}_2(\xi_n) | \underline{\mathbf{0}} \rangle + \langle \underline{\mathbf{h}} | \sum_{l \neq n, l=1}^{\mathsf{N}} \mathsf{T}_1(\xi_l) | \underline{\mathbf{0}} \rangle \\
&\underset{\text{UpC}}{=} t_1 \langle \underline{\mathbf{h}} | \underline{\mathbf{0}} \rangle + \langle \underline{\mathbf{h}}^{(1)}_n | \mathsf{T}_2(\xi_n) | \underline{\mathbf{0}} \rangle + \sum_{l \neq n, l=1}^{\mathsf{N}} c_l^{\delta_{h_l,0}} \langle \underline{\mathbf{h}}^{(h_l+1)}_l | \underline{\mathbf{0}} \rangle \\
&\underset{\text{UpC}}{=} \langle \underline{\mathbf{h}}^{(1)}_n | \mathsf{T}_2(\xi_n) | \underline{\mathbf{0}} \rangle.
\end{aligned} \qquad (\text{C.9})$$

Now, we have to use the interpolation formula for $T_2(\xi_n)$

$$
\begin{aligned}
\langle \underline{\mathbf{h}}|T_1(\xi_n)|\underline{\mathbf{0}}\rangle &= \langle \underline{\mathbf{h}}_n^{(1)}|T_2(\xi_n)|\underline{\mathbf{0}}\rangle \\
&\underset{\text{UpC}}{=} t_2\langle \underline{\mathbf{h}}_n^{(1)}|\underline{\mathbf{0}}\rangle + \langle \underline{\mathbf{h}}_n^{(1)}|\sum_{l=1}^{N}(\delta_{h_l',0}T_2(\xi_l)+\delta_{h_l',1}T_2(\xi_l^{(1)}))|\underline{\mathbf{0}}\rangle \\
&\underset{\text{UpC}}{=} t_2\langle \underline{\mathbf{h}}_n^{(1)}|\underline{\mathbf{0}}\rangle + \sum_{l=1}^{N}\delta_{h_l',1}\langle \underline{\mathbf{h}}_{n,l}^{(1,0)}|\underline{\mathbf{0}}\rangle + \sum_{l=1}^{N}\delta_{h_l',0}c_l\langle \underline{\mathbf{h}}_{n,l}^{(1,1)}|T_1(\xi_l^{(1)})|\underline{\mathbf{0}}\rangle \\
&= 0,
\end{aligned}
\tag{C.10}
$$

where we have defined $\underline{\mathbf{h}}' = \{h_1',...,h_N'\} = \underline{\mathbf{h}}_n^{(1)}$ and used that $\underline{\mathbf{h}}_{n,l}^{(1,0)} \neq \underline{\mathbf{0}}$ holds even for $l = n$, as the condition $\underline{\mathbf{h}}_n^{(2)} \neq \underline{\mathbf{0}}_n^{(2)}$ implies $\underline{\mathbf{h}}_{n,n}^{(1,0)} = \underline{\mathbf{h}}_n^{(0)} \neq \underline{\mathbf{0}}$. Moreover, it holds

$$
\langle \underline{\mathbf{h}}_{n,l}^{(1,1)}|T_1(\xi_l^{(1)})|\underline{\mathbf{0}}\rangle \underset{\text{UpC}}{=} t_1\langle \underline{\mathbf{h}}_{n,l}^{(1,1)}|\underline{\mathbf{0}}\rangle + \sum_{m=1}^{N}\langle \underline{\mathbf{h}}_{n,l}^{(1,1)}|T_1(\xi_m)|\underline{\mathbf{0}}\rangle,
\tag{C.11}
$$

and defining $\underline{\mathbf{h}}'' = \{h_1'',...,h_N''\} = \underline{\mathbf{h}}_{n,l}^{(1,1)}$, we get

$$
\langle \underline{\mathbf{h}}_{n,l}^{(1,1)}|T_1(\xi_l^{(1)})|\underline{\mathbf{0}}\rangle \underset{\text{UpC}}{=} \sum_{m=1}^{N}\langle \underline{\mathbf{h}}_{n,l,m}^{(1,1,h_m''+1)}|\underline{\mathbf{0}}\rangle = 0.
\tag{C.12}
$$

Let us now consider the induction, i.e. we assume that the orthogonality works when there are $m \geq 1$ values of $h_a = 2$ in $\langle \underline{\mathbf{h}}|$ and we want to prove it for the case of $m+1$ values of $h_a = 2$ in $\langle \underline{\mathbf{h}}|$. Up to a reordering of the index of the $\{\xi_a\}$, this is equivalent to prove that

$$
\langle h_1 = 2,...,h_{m+1} = 2, h_{l\geq m+2} \in \{0,1\}|T_1(\xi_{m+1})|\underline{\mathbf{0}}\rangle = 0.
\tag{C.13}
$$

Setting

$$
\underline{\mathbf{h}} = \{h_1 = 2,...,h_{m+1} = 2, h_{l\geq m+2} \in \{0,1\}\},
\tag{C.14}
$$

and once again using the development by interpolation formula, we get

$$
\langle \underline{\mathbf{h}}|T_1(\xi_{m+1})|\underline{\mathbf{0}}\rangle \underset{\text{UpC}}{=} t_1\langle \underline{\mathbf{h}}|\underline{\mathbf{0}}\rangle + \langle \underline{\mathbf{h}}|\sum_{l=m+2}^{N}T_1(\xi_l)|\underline{\mathbf{0}}\rangle + \sum_{l=1}^{m+1}\langle \underline{\mathbf{h}}|T_1(\xi_l^{(1)})|\underline{\mathbf{0}}\rangle
\tag{C.15}
$$

$$
\underset{\text{UpC}}{=} \sum_{l=1}^{m+1}\langle \underline{\mathbf{h}}_l^{(1)}|T_2(\xi_l)|\underline{\mathbf{0}}\rangle,
\tag{C.16}
$$

so expanding $T_2(\xi_l)$:

$$
\langle \underline{\mathbf{h}}_l^{(1)}|T_2(\xi_l)|\underline{\mathbf{0}}\rangle \underset{\text{UpC}}{=} t_2\langle \underline{\mathbf{h}}_l^{(1)}|\underline{\mathbf{0}}\rangle + \langle \underline{\mathbf{h}}_l^{(1)}|\sum_{r=1}^{N}T_2(\xi_r^{(\delta_{h_r',1}+\delta_{h_r',2})})|\underline{\mathbf{0}}\rangle,
\tag{C.17}
$$

where $h_r'$ are the elements of $\underline{\mathbf{h}}' = \{h_1',...,h_N'\} \equiv \underline{\mathbf{h}}_l^{(1)}$. Then, we can use the rewriting

$$
\left(\delta_{h_r',1}+\delta_{h_r',2}\right)\langle \underline{\mathbf{h}}'|T_2(\xi_r^{(1)})|\underline{\mathbf{0}}\rangle = \left(\delta_{h_r',1}+\delta_{h_r',2}\right)c_r^{\delta_{h_r',2}}\langle \underline{\mathbf{h}}_{l,r}^{(1,h_r'-1)}|\underline{\mathbf{0}}\rangle = 0.
\tag{C.18}
$$

Indeed, $\underline{\mathbf{h}}_{l,r}^{(1,h_r'-1)} \neq \underline{\mathbf{0}}$ holds even for $r = l$, as $\underline{\mathbf{h}}_{l,l}^{(1,h_l'-1)} = \underline{\mathbf{h}}_l^{(0)}$ has at least one element equal to 2 being by assumption $m \geq 1$. Then, we get

$$
\langle \underline{\mathbf{h}}_l^{(1)}|T_2(\xi_l)|\underline{\mathbf{0}}\rangle \underset{\text{UpC}}{=} \langle \underline{\mathbf{h}}_l^{(1)}|\sum_{r=m+2}^{N}\delta_{h_r',0}T_2(\xi_r)|\underline{\mathbf{0}}\rangle = \sum_{r=m+2}^{N}\delta_{h_r',0}c_r\langle \underline{\mathbf{h}}_{l,r}^{(1,1)}|T_1(\xi_r^{(1)})|\underline{\mathbf{0}}\rangle,
\tag{C.19}
$$

and finally:

$$\langle \underline{\mathbf{h}}_{l,r}^{(1,1)} | T_1(\xi_r^{(1)}) | \underline{\mathbf{0}} \rangle \underset{\text{UpC}}{=} t_1 \langle \underline{\mathbf{h}}_{l,r}^{(1,1)} | \underline{\mathbf{0}} \rangle + \langle \underline{\mathbf{h}}_{l,r}^{(1,1)} | \sum_{s=1}^{N} T_1(\xi_s) | \underline{\mathbf{0}} \rangle, \tag{C.20}$$

which is zero by the induction. So we have proven the orthogonality:

$$\langle \underline{\mathbf{h}} | \underline{\mathbf{0}}_j^{(2)} \rangle = 0, \quad \text{for any } \underline{\mathbf{h}} \neq \underline{\mathbf{0}}_j^{(2)}. \tag{C.21}$$

**ii) The general $\underline{\mathbf{k}} \in \{0,2\}^N$** We now perform the induction over the number of $k_a = 2$ in $\underline{\mathbf{k}} \in \{0,2\}^N$. The orthogonality is assumed to work when there are $m$ values of $k_a = 2$ in $\underline{\mathbf{k}}$, while the others being all 0, and we want to prove it for the case of $m+1$ values of $k_a = 2$ in $\underline{\mathbf{k}}$.

Let us start proving the following

**Lemma C.1.** *Let $\underline{\mathbf{k}} \in \{0,2\}^N$ with*

$$\sum_{a=1}^{N} \delta_{k_a,2} = m, \tag{C.22}$$

*and $\overset{ab}{\underline{\mathbf{h}}}$ a N-tuple in $\{0,1,2\}$ such that $h_a \neq k_a$ and $h_b \neq k_b$ if $a \neq b$, while $h_a = 1 \neq k_a = 0$ if $a = b$. The following recursive formula holds for any fixed $c \in \{1, ..., N\}$ :*

$$\langle \overset{ab}{\underline{\mathbf{h}}} | T_1\big(\xi_c^{(\delta_{h_c,0}+\delta_{h_c,1})}\big) | \underline{\mathbf{k}} \rangle \underset{\text{UpC}}{=} \sum_{r=1}^{N} \delta_{h_r,2} \sum_{s=1,s\neq r}^{N} \delta_{h_s,0} c_s \langle \overset{ab}{\underline{\mathbf{h}}}_{r,s}^{(1,1)} | T_1(\xi_s^{(1)}) | \underline{\mathbf{k}} \rangle. \tag{C.23}$$

*Proof.* Let us use this first interpolation formula:

$$\langle \overset{ab}{\underline{\mathbf{h}}} | T_1\big(\xi_c^{(\delta_{h_c,0}+\delta_{h_c,1})}\big) | \underline{\mathbf{k}} \rangle \underset{\text{UpC}}{=} t_1 \langle \overset{ab}{\underline{\mathbf{h}}} | \underline{\mathbf{k}} \rangle + \langle \overset{ab}{\underline{\mathbf{h}}} | \sum_{r=1}^{N} T_1\big(\xi_r^{(\delta_{h_r,2})}\big) | \underline{\mathbf{k}} \rangle$$

$$\underset{\text{UpC}}{=} \sum_{r=1}^{N} \delta_{h_r,2} \langle \overset{ab}{\underline{\mathbf{h}}}_r^{(1)} | T_2(\xi_r) | \underline{\mathbf{k}} \rangle, \tag{C.24}$$

as by the orthogonality assumption it holds:

$$\langle \overset{ab}{\underline{\mathbf{h}}} | \underline{\mathbf{k}} \rangle = 0, \tag{C.25}$$

as well as

$$\big(\delta_{h_r,0} + \delta_{h_r,1}\big) \langle \overset{ab}{\underline{\mathbf{h}}} | T_1(\xi_r) | \underline{\mathbf{k}} \rangle = \big(\delta_{h_r,0} c_r + \delta_{h_r,1}\big) \langle \overset{ab}{\underline{\mathbf{h}}}_r^{(h_r+1)} | \underline{\mathbf{k}} \rangle = 0, \quad \forall r \in \{1, ..., N\}, \tag{C.26}$$

being $\overset{ab}{\underline{\mathbf{h}}}_r^{(h_r+1)} \neq \underline{\mathbf{k}}$ under the condition $\big(\delta_{h_r,0} + \delta_{h_r,1}\big) = 1$. This is easily the case for $b \neq a$ as $\overset{ab}{\underline{\mathbf{h}}}_r^{(h_r+1)}$ keeps at least one $h_j \neq k_j$, for $j = a$ or $j = b$, independently from the choice of $r$. In the case $a = b$ it holds $\overset{ab}{\underline{\mathbf{h}}}_r^{(h_r+1)} = \underline{\mathbf{h}}_{a,r}^{(1,h_r+1)}$ so that for $r \neq a$ it still holds $h_a = 1 \neq k_a = 0$. Finally, in the case $r = b = a$ it holds $\overset{ab}{\underline{\mathbf{h}}}_r^{(h_r+1)} = \underline{\mathbf{h}}_a^{(2)} \neq \underline{\mathbf{k}}$.

Now, let us use the following second interpolation formula to develop the terms on the r.h.s. of (C.24) :

$$\langle \overset{ab}{\underline{\mathbf{h}}}_r^{(1)} | T_2(\xi_r) | \underline{\mathbf{k}} \rangle \underset{\text{UpC}}{=} t_2 \langle \overset{ab}{\underline{\mathbf{h}}}_r^{(1)} | \underline{\mathbf{k}} \rangle + \langle \overset{ab}{\underline{\mathbf{h}}}_r^{(1)} | \sum_{s=1}^{N} T_2\big(\xi_s^{(\delta_{h'_s,1}+\delta_{h'_s,2})}\big) | \underline{\mathbf{k}} \rangle$$

$$\underset{\text{UpC}}{=} \sum_{s=1,s\neq r}^{N} \delta_{h_s,0} c_s \langle \overset{ab}{\underline{\mathbf{h}}}_{r,s}^{(1,1)} | T_1(\xi_s^{(1)}) | \underline{\mathbf{k}} \rangle, \tag{C.27}$$

where we have defined $\{h_1', ..., h_N'\} \equiv \underset{r}{\overset{ab(1)}{\underline{\mathbf{h}}}}$. Indeed, by the orthogonality condition it holds:

$$\langle \underset{r}{\overset{ab(1)}{\underline{\mathbf{h}}}} \, | \underline{\mathbf{k}} \rangle = 0, \tag{C.28}$$

and

$$\left(\delta_{h_s',1} + \delta_{h_s',2}\right) \langle \underset{r}{\overset{ab(1)}{\underline{\mathbf{h}}}} \, | T_2(\xi_s^{(1)}) | \underline{\mathbf{k}} \rangle = \left(\delta_{h_s',1} + \delta_{h_s',2}\right) c_s^{\delta_{h_s',2}} \langle \underset{r,s}{\overset{ab(1,h_s'-1)}{\underline{\mathbf{h}}}} \, | \underline{\mathbf{k}} \rangle = 0, \quad \forall s \in \{1, ..., N\}, \tag{C.29}$$

being $\underset{r,s}{\overset{ab(1,h_s'-1)}{\underline{\mathbf{h}}}} \neq \underline{\mathbf{k}}$ under the condition $\left(\delta_{h_s',1} + \delta_{h_s',2}\right) = 1$. Indeed, this is easily the case for $s \neq r$ as $\underset{r,s}{\overset{ab(1,h_s'-1)}{\underline{\mathbf{h}}}}$ keeps $h_r' = 1 \neq k_r \in \{0,2\}$, independently from the choice of $s$. In the case $s = r$, it holds $\underset{r,s}{\overset{ab(1,h_s'-1)}{\underline{\mathbf{h}}}} = \underset{r}{\overset{ab(0)}{\underline{\mathbf{h}}}}$ so that for $b \neq a$ $\underset{r}{\overset{ab(0)}{\underline{\mathbf{h}}}}$ keeps at least one $h_j \neq k_j$, for $j = a$ or $j = b$. Finally, in the case $s = r$ and $a = b$ it holds $r \neq a$ and so $\underset{r}{\overset{ab(0)}{\underline{\mathbf{h}}}} \neq \underline{\mathbf{k}}$ as by our hypothesis on $\overset{aa}{\underline{\mathbf{h}}}$ we have $h_a = 1$ while by (C.24) it must hold $h_r = 2$.

Putting together the results of these interpolation developments, we get our recursion formula as a consequence of the orthogonality assumed for $m$ values of $k_j = 2$ in $\underline{\mathbf{k}}$. $\qquad\square$

Note that the above lemma gives a recursive formula. The terms on the r.h.s. of equation (C.23) are of the same type as the starting one on the l.h.s. , and for any $r, s$ such $\delta_{h_r,2} = \delta_{h_s,0} = 1$, the $\underset{r,s}{\overset{ab}{\underline{\mathbf{h}}}}^{(1,1)}$ surely satisfies the condition to have at least two different elements w.r.t. the given $\underline{\mathbf{k}} \in \{0,2\}^N$. Indeed, $\underset{r,s}{\overset{ab}{\underline{\mathbf{h}}}}^{(1,1)}$ contains a couple of elements equal to 1. Hence it is possible to apply the same recursion formula to the terms on the r.h.s. of equation (C.23).

The previous lemma implies the following:

**Corollary C.1.** *Let $\underline{\mathbf{k}} \in \{0,2\}^N$ with*

$$\sum_{a=1}^{N} \delta_{k_a,2} = m, \tag{C.30}$$

*and $\overset{ab}{\underline{\mathbf{h}}}$ such that $h_a \neq k_a$ and $h_b \neq k_b$ if $a \neq b$, while $h_a = 1 \neq k_a = 0$ if $a = b$. The following orthogonality conditions hold for any fixed $c \in \{1, ..., N\}$ :*

$$\langle \overset{ab}{\underline{\mathbf{h}}} | T_1\left(\xi_c^{(\delta_{h_c,0}+\delta_{h_c,1})}\right) | \underline{\mathbf{k}} \rangle = 0. \tag{C.31}$$

*Proof.* Note that if $\overset{ab}{\underline{\mathbf{h}}}$ does not contain $h = 2$ or $h = 0$, the orthogonality is proven just by applying once the recursion formula. Otherwise, the recursion generate the $\underset{r,s}{\overset{ab}{\underline{\mathbf{h}}}}^{(1,1)}$ where we have reduced by one the number of $h = 2$, reduced by one the number of $h = 0$ and increased by two the number of $h = 1$. This amounts to change $\underline{\mathbf{h}}$ in $\underline{\mathbf{h}}'$, with $h_{a \neq r,s}' = h_a$ but $h_r = 2 \to h_r' = 1$ and $h_s = 0 \to h_s' = 1$. Then, if $\underset{r,s}{\overset{ab}{\underline{\mathbf{h}}}}^{(1,1)}$ does not contain $h = 2$ or $h = 0$, the orthogonality is proven just by applying the recursion formula one more time. Otherwise, we continue to apply (C.23) until we arrive to the condition that there are no $h = 2$ or $h = 0$ in the index of the SoV co-vectors involved. This proves the above corollary. $\qquad\square$

We are now in position to perform the induction over the number $m$ of $k_a = 2$ for the orthogonality.

Up to a reordering in the indices of the $\{\xi_a\}$, this is equivalent to prove:

$$\langle \underline{\mathbf{h}}|T_1(\xi_{m+1})|\underline{\mathbf{k}}\rangle = 0, \quad \text{for any } \underline{\mathbf{h}} \neq \underline{\mathbf{k}}_{m+1}^{(2)}, \tag{C.32}$$

where we have defined:

$$\underline{\mathbf{k}} = \{k_1 = 2, ..., k_m = 2, k_{l \geq m+1} = 0\}. \tag{C.33}$$

The only case that we have to consider is

$$\underline{\mathbf{h}} \neq \underline{\mathbf{k}}_{m+1}^{(2)} \text{ with } h_1 = 2, ..., h_{m+1} = 2. \tag{C.34}$$

Indeed, if this is not the case we can write:

$$\langle \underline{\mathbf{h}}|T_1(\xi_{m+1})|\underline{\mathbf{k}}\rangle = \langle \underline{\mathbf{h}}|T_1(\xi_{l<m+1})|\underline{\mathbf{k}}_{l,m+1}^{(0,2)}\rangle, \tag{C.35}$$

and we can directly apply the corresponding $T_1(\xi_{l \leq m+1})$ on the left vector $\langle \underline{\mathbf{h}}|$, increasing by one the associated $h_{l \leq m+1} \leq 1$. Then, using the orthogonality assumed for $m$ values of $k_j = 2$ in $\underline{\mathbf{k}}$, we get zero, i.e. for $h_{l \leq m+1} \leq 1$ it holds[19]:

$$\langle \underline{\mathbf{h}}|T_1(\xi_{m+1})|\underline{\mathbf{k}}\rangle = \langle \underline{\mathbf{h}}_l^{(h_l+1)}|\underline{\mathbf{k}}_{l,m+1}^{(0,2)}\rangle = 0. \tag{C.36}$$

So it is sufficient to consider the tuples $\underline{\mathbf{h}}$ of the form (C.34). But then $\underline{\mathbf{h}}$ has at least two elements different from the given $\underline{\mathbf{k}} \in \{0,2\}^{\mathsf{N}}$. Indeed, from $\underline{\mathbf{h}} \neq \underline{\mathbf{k}}_{m+1}^{(2)}$ it follows that there exists at least one $j \in \{m+2, ..., \mathsf{N}\}$ such that $h_j \neq k_j = 0$, and by the definitions (C.34) and (C.33) of $\underline{\mathbf{h}}$ and $\underline{\mathbf{k}}$, it holds $h_{m+1} = 2 \neq k_{m+1} = 0$. So we get our proof of the orthogonality induction being:

$$\langle \underline{\mathbf{h}}|T_1(\xi_{m+1})|\underline{\mathbf{k}}\rangle = 0, \tag{C.37}$$

as consequence of (C.31). Note that the proven orthogonality also implies that the above lemma and corollary indeed hold for any $m \leq \mathsf{N}$.

### C.1.2 Second step: the case $|\underline{\mathbf{k}}\rangle$ with $k_a = 1$, $k_{b \neq a} \in \{0, 2\}$

Let us make the orthogonality proof in the case where $\underline{\mathbf{k}}$ contains only one $a \in \{1, ..., \mathsf{N}\}$, such that $k_a = 1$ while $k_b \in \{0, 2\}$ for any $b \neq a \in \{1, ..., \mathsf{N}\}$, i.e. let us show that it holds:

$$\langle \underline{\mathbf{h}}|T_2(\xi_a)|\underline{\mathbf{k}}_a^{(0)}\rangle = 0, \quad \forall \underline{\mathbf{h}} \neq \underline{\mathbf{k}} \text{ with } \underline{\mathbf{h}} \in \{0,1,2\}^{\mathsf{N}}. \tag{C.38}$$

In the case $h_a = 0$, it rewrites

$$\langle \underline{\mathbf{h}}|T_2(\xi_a)|\underline{\mathbf{k}}_a^{(0)}\rangle = c_a \langle \underline{\mathbf{h}}_a^{(1)}|T_1(\xi_a^{(1)})|\underline{\mathbf{k}}_a^{(0)}\rangle = 0, \tag{C.39}$$

and this follows by (C.31), observing that $\underline{\mathbf{k}}_a^{(0)} \in \{0,2\}^{\mathsf{N}}$.

In the case $h_a = 1$ or $h_a = 2$, we first implement the interpolation development of $T_2(\xi_a)$:

$$\langle \underline{\mathbf{h}}|T_2(\xi_a)|\underline{\mathbf{k}}_a^{(0)}\rangle \underset{\text{UpC}}{=} t_2 \langle \underline{\mathbf{h}}|\underline{\mathbf{k}}_a^{(0)}\rangle + \langle \underline{\mathbf{h}}|\sum_{s=1}^{\mathsf{N}} T_2(\xi_s^{(\delta_{h_s,1}+\delta_{h_s,2})})|\underline{\mathbf{k}}_a^{(0)}\rangle$$

$$\underset{\text{UpC}}{=} \sum_{s=1, s \neq a}^{\mathsf{N}} \delta_{h_s,0} c_s \langle \underline{\mathbf{h}}_s^{(1)}|T_1(\xi_s^{(1)})|\underline{\mathbf{k}}_a^{(0)}\rangle. \tag{C.40}$$

---

[19]Note that the above discussion also implies the orthogonality $\langle \underline{\mathbf{h}}|\underline{\mathbf{2}}\rangle = 0$ for any $\underline{\mathbf{h}} \neq \underline{\mathbf{2}}$.

Indeed, we have:

$$\langle\underline{\mathbf{h}}|\underline{\mathbf{k}}_a^{(0)}\rangle = 0, \tag{C.41}$$

and

$$\left(\delta_{h_s,1} + \delta_{h_s,2}\right)\langle\underline{\mathbf{h}}|\mathsf{T}_2(\xi_s^{(1)})|\underline{\mathbf{k}}_a^{(0)}\rangle = (\delta_{h_s,1} + \delta_{h_s,2}c_s^{\delta_{h_s,2}})\langle\underline{\mathbf{h}}_s^{(h_s-1)}|\underline{\mathbf{k}}_a^{(0)}\rangle = 0, \tag{C.42}$$

as it holds $\underline{\mathbf{h}}_s^{(h_s-1)} \neq \underline{\mathbf{k}}_a^{(0)}$ for $s \neq a$ being $h_a \in \{1,2\}$, while for $s = a$ it still holds $\underline{\mathbf{h}}_a^{(h_a-1)} \neq \underline{\mathbf{k}}_a^{(0)}$ evidently for $h_a = 2$ but also for $h_a = 1$. Indeed, in this last case, the condition $\underline{\mathbf{h}} \neq \underline{\mathbf{k}}$ is explicitly written as $\underline{\mathbf{h}}_a^{(1)} \neq \underline{\mathbf{k}}_a^{(1)}$, which is equivalent to $\underline{\mathbf{h}}_a^{(0)} \neq \underline{\mathbf{k}}_a^{(0)}$. Now our orthogonality condition follows just remarking that $\underline{\mathbf{h}}_s^{(1)}$ and $\underline{\mathbf{k}}_a^{(0)}$ have different $a$ and $s(\neq a)$ elements, so that it holds

$$\langle\underline{\mathbf{h}}_s^{(1)}|\mathsf{T}_1(\xi_s^{(1)})|\underline{\mathbf{k}}_a^{(0)}\rangle = 0, \tag{C.43}$$

by applying (C.31).

### C.1.3  Third step: orthogonality by induction on the number of $k_i = 1$ in $|\underline{\mathbf{k}}\rangle$

Let us now prove our final orthogonality statement for the general case of $m+1$ indices $k_a = 1$ in $\underline{\mathbf{k}}$ by induction. Up to a reordering of the indices, this is equivalent to ask that given the N-tuple

$$\underline{\mathbf{k}}: \quad k_1 = k_2 = \cdots = k_m = k_{m+1} = 1, \ k_{l \geq m+2} \in \{0,2\}, \tag{C.44}$$

(we are fixing $\mathbf{1}_{\underline{\mathbf{k}}} = \{1,\dots,m+1\}$ and so $n_{\underline{\mathbf{k}}}$ is the integer part of $(m+1)/2$) then the covector $\langle\underline{\mathbf{h}}|$ is orthogonal to $|\underline{\mathbf{k}}\rangle$, i.e. $\langle\underline{\mathbf{h}}|\underline{\mathbf{k}}\rangle = 0$, if and only if:

$$\underline{\mathbf{h}} \in \{0,1,2\}^{\mathsf{N}} \quad \text{such that} \quad \sum_{r=0}^{n_{\underline{\mathbf{k}}}} \sum_{\substack{\alpha\cup\beta\cup\gamma=\mathbf{1}_{\underline{\mathbf{k}}}, \\ \alpha,\beta,\gamma \text{ disjoint}, \ \#\alpha=\#\beta=r}} \delta_{\underline{\mathbf{h}},\underline{\mathbf{k}}_{\alpha,\beta}^{(0,2)}} = 0. \tag{C.45}$$

The above condition on $\underline{\mathbf{h}}$ is denoted by

$$\underline{\mathbf{h}} \underset{\text{(C.45)}}{\neq} \underline{\mathbf{k}}. \tag{C.46}$$

It simply says that for any choice of the disjoint subsets $\alpha, \beta \subset \mathbf{1}_{\underline{\mathbf{k}}}$ with the same cardinality $0 \leq \#\alpha = \#\beta = r \leq n_{\underline{\mathbf{k}}}$, it must holds:

$$\underline{\mathbf{h}} \neq \underline{\mathbf{k}}_{\alpha,\beta}^{(0,2)}. \tag{C.47}$$

In the following, we assume that this orthogonality holds in the case where there are only $m$ values of $k_a = 1$, and prove it for $m+1$.

Let us start proving the following Lemma.

**Lemma C.2.** *Let $\underline{\mathbf{h}}$ be the generic element of $\{0,1,2\}^{\mathsf{N}}$ with $h_1 \neq 0$ and $h_r \neq 0$, satisfying (C.46) with $\underline{\mathbf{k}}$ of the form (C.44). The following recursive formula*

$$\langle\underline{\mathbf{h}}_{1,r}^{(h_1\neq0,h_r\neq0)}|\mathsf{T}_2(\xi_r)|\underline{\mathbf{k}}_1^{(0)}\rangle \underset{UpC}{=} \sum_{p=1}^{\mathsf{N}}\delta_{h_p,0}c_p\sum_{q=1}^{\mathsf{N}}\delta_{h_q,2}\langle\underline{\mathbf{h}}_{1,r,p,q}^{(h_1\neq0,h_r\neq0,1,1)}|\mathsf{T}_2(\xi_q)|\underline{\mathbf{k}}_a^{(0)}\rangle, \tag{C.48}$$

*holds for any fixed $r \in \{1,\dots,\mathsf{N}\}$, indifferently equal or different from 1.*

*Proof.* Let us make a first interpolation

$$
\begin{aligned}
\langle \underline{\mathbf{h}}_{1,r}^{(h_1\neq 0,h_r\neq 0)}|T_2(\xi_r)|\underline{\mathbf{k}}_1^{(0)}\rangle &\underset{\text{UpC}}{=} t_2\langle \underline{\mathbf{h}}_{1,r}^{(h_1\neq 0,h_r\neq 0)}|\underline{\mathbf{k}}_1^{(0)}\rangle + \langle \underline{\mathbf{h}}_{1,r}^{(h_1\neq 0,h_r\neq 0)}|\sum_{s=1}^{N} T_2\big(\xi_s^{(\delta_{h_s,1}+\delta_{h_s,2})}\big)|\underline{\mathbf{k}}_1^{(0)}\rangle \\
&\underset{\text{UpC}}{=} t_2\langle \underline{\mathbf{h}}_{1,r}^{(h_1\neq 0,h_r\neq 0)}|\underline{\mathbf{k}}_1^{(0)}\rangle + \sum_{s=1}^{N}\Big[c_s^{\delta_{h_s,2}}\big(\delta_{h_s,1}+\delta_{h_s,2}\big)\langle \underline{\mathbf{h}}_{1,r,s}^{(h_1\neq 0,h_r\neq 0,h_s-1)}| \\
&\quad + \delta_{h_s,0}c_s\langle \underline{\mathbf{h}}_{1,r,s}^{h_1\neq 0,h_r\neq 0,1}|\Big]T_1(\xi_s^{(1)})|\underline{\mathbf{k}}_1^{(0)}\rangle.
\end{aligned}
\tag{C.49}
$$

Now we have that it holds:

$$
\underline{\mathbf{h}}_{1,r}^{(h_1\neq 0,h_r\neq 0)} \underset{\text{(C.45)}}{\neq} \underline{\mathbf{k}}_1^{(0)},
\tag{C.50}
$$

$$
\underline{\mathbf{h}}_{1,r,s}^{(h_1\neq 0,h_r\neq 0,h_s-1)} \underset{\text{(C.45)}}{\neq} \underline{\mathbf{k}}_1^{(0)} \quad \text{for any } s \text{ such that } \delta_{h_s,1}+\delta_{h_s,2}=1.
\tag{C.51}
$$

Let us show the validity of (C.50) first. Clearly, we have $\underline{\mathbf{h}}_{1,r}^{(h_1\neq 0,h_r\neq 0)} \neq \underline{\mathbf{k}}_1^{(0)}$ as $h_1 \in \{1,2\} \neq k_1 = 0$. Moreover, taking $\underline{\mathbf{k}}_{1,a,b}^{(0,0,2)}$ for any $a \neq b \in \{2,...,m+1\}$, it holds $\underline{\mathbf{h}}_{1,r}^{(h_1\neq 0,h_r\neq 0)} \neq \underline{\mathbf{k}}_{1,a,b}^{(0,0,2)}$, because $h_1 \in \{1,2\} \neq k_1 = 0$. Similarly, if we take the generic $\alpha, \beta \subset \{2,...,m+1\}$ with $\alpha \cap \beta = \emptyset$ and $0 \leq \#\alpha = \#\beta = r \leq n_{\underline{\mathbf{k}}}$, it must holds:

$$
\underline{\mathbf{h}}_{1,r}^{(h_1\neq 0,h_1\neq 0)} \neq \underline{\mathbf{k}}_{1,\alpha,\beta}^{(0,2,...,2,0,...,0)},
\tag{C.52}
$$

which is (C.50). Note that our proof of (C.50) holds independently from the value of $r$ : it is valid both for $r \neq 1$ and for $r = 1$ where $\underline{\mathbf{h}}_{1,r}^{(h_1\neq 0,h_r\neq 0)} = \underline{\mathbf{h}}_1^{(h_1\neq 0)}$.

Let us now show (C.51). We have to distinguish the two cases $s = 1$ and $s \neq 1$. If $s = 1$ and $h_1 = 2$, we have $\underline{\mathbf{h}}_{1,r,s}^{(h_1\neq 0,h_r\neq 0,h_s-1)} = \underline{\mathbf{h}}_{r,1}^{(h_r\neq 0,1)}$, so the proof of (C.51) follows the same steps as the one for (C.50), independently from the value of $r$. If $s = 1$ and $h_1 = 1$, we have $\underline{\mathbf{h}}_{1,r,s}^{(h_1\neq 0,h_r\neq 0,h_s-1)} = \underline{\mathbf{h}}_{r,1}^{(h_r\neq 0,0)}$, and the following implication holds:

$$
\underline{\mathbf{h}}_{r,1}^{(h_r\neq 0,1)} \underset{\text{(C.45)}}{\neq} \underline{\mathbf{k}}_1^{(1)} \quad \Longrightarrow \quad \underline{\mathbf{h}}_{r,1}^{(h_r\neq 0,0)} \underset{\text{(C.45)}}{\neq} \underline{\mathbf{k}}_1^{(0)},
\tag{C.53}
$$

where the l.h.s. is our starting point assumption once we fix $h_1 = 1$, holds independently from the value of $r$. Note that we have used the notations $\underline{\mathbf{h}}_{r,1}^{(h_r\neq 0,1)}$ and $\underline{\mathbf{h}}_{r,1}^{(h_r\neq 0,0)}$, as these correctly reduce to $\underline{\mathbf{h}}_1^{(1)}$ and $\underline{\mathbf{h}}_1^{(0)}$ for $r = 1$, respectively.

Now, if $s \neq 1$, we have by definition $h_1 \neq k_1 = 0$ in $\underline{\mathbf{h}}_{1,r,s}^{(h_1\neq 0,h_r\neq 0,h_s-1)}$ so the proof of (C.51) follows once again the same lines as those of (C.50). This proves equation (C.51).

Returning to (C.49) and using the fact that in $\underline{\mathbf{k}}_1^{(0)}$ there are exactly $m$ entries with $k = 1$, equations (C.50), (C.51) and the assumed orthogonality give

$$
\langle \underline{\mathbf{h}}_{1,r}^{(h_1\neq 0,h_r\neq 0)}|\underline{\mathbf{k}}_1^{(0)}\rangle = 0 \quad \text{and} \quad \big(\delta_{h_s,1}+\delta_{h_s,2}\big)\langle \underline{\mathbf{h}}_{1,r,s}^{(h_1\neq 0,h_r\neq 0,h_s-1)}|\underline{\mathbf{k}}_1^{(0)}\rangle = 0,
\tag{C.54}
$$

and we are left with

$$
\langle \underline{\mathbf{h}}|T_2(\xi_1)|\underline{\mathbf{k}}_1^{(0)}\rangle \underset{\text{UpC}}{=} \sum_{p=1}^{N} \delta_{h_p,0}c_p\langle \underline{\mathbf{h}}_{1,r,p}^{(h_1\neq 0,h_r\neq 0,1)}|T_1(\xi_p^{(1)})|\underline{\mathbf{k}}_1^{(0)}\rangle.
\tag{C.55}
$$

Note that it similarly holds:

$$
\underline{\mathbf{h}}_{1,r,p}^{(h_1\neq 0,h_r\neq 0,1)} \underset{\text{(C.45)}}{\neq} \underline{\mathbf{k}}_1^{(0)},
\tag{C.56}
$$

as $h_1 = 1$ or $2$ does not coincide with $k_1 = 0$ and $p \neq 1$ and $r$, being associated to the condition $\delta_{h_p,0} = 1$.

Defining $\{h'_1, ..., h'_N\} \equiv \underline{\mathbf{h}}_{1,r,p}^{(h_1 \neq 0, h_r \neq 0, 1)}$, we can now perform a second interpolation:

$$\langle \underline{\mathbf{h}}_{1,r,p}^{(h_1 \neq 0, h_r \neq 0, 1)} | T_1(\xi_p^{(1)}) | \underline{\mathbf{k}}_1^{(0)} \rangle$$

$$\underset{\text{UpC}}{=} t_1 \langle \underline{\mathbf{h}}_{1,r,p}^{(h_1 \neq 0, h_r \neq 0, 1)} | \underline{\mathbf{k}}_1^{(0)} \rangle + \langle \underline{\mathbf{h}}_{1,r,p}^{(h_1 \neq 0, h_r \neq 0, 1)} | \sum_{q=1}^{N} T_1(\xi_q^{(\delta_{h'_q, 2})}) | \underline{\mathbf{k}}_1^{(0)} \rangle \tag{C.57}$$

$$\underset{\text{UpC}}{=} \sum_{q=1}^{N} \left( \delta_{h'_q, 0} + \delta_{h'_q, 1} \right) c_q^{\delta_{h'_q, 0}} \langle \underline{\mathbf{h}}_{1,r,p,q}^{(h_1 \neq 0, h_r \neq 0, 1, h'_q + 1)} | \underline{\mathbf{k}}_1^{(0)} \rangle + \sum_{q=1}^{N} \delta_{h'_q, 2} \langle \underline{\mathbf{h}}_{1,r,p,q}^{(h_1 \neq 0, h_r \neq 0, 1, 1)} | T_2(\xi_q) | \underline{\mathbf{k}}_1^{(0)} \rangle \tag{C.58}$$

$$\underset{\text{UpC}}{=} \sum_{r=1}^{N} \delta_{h_q, 2} \langle \underline{\mathbf{h}}_{1,r,p,q}^{(h_1 \neq 0, h_r \neq 0, 1, 1)} | T_2(\xi_q) | \underline{\mathbf{k}}_1^{(0)} \rangle, \tag{C.59}$$

where (C.59) follows as it holds:

$$\underline{\mathbf{h}}_{1,r,p,q}^{(h_1 \neq 0, h_r \neq 0, 1, h'_q + 1)} \underset{\text{(C.45)}}{\neq} \underline{\mathbf{k}}_1^{(0)}, \quad \text{for any } q \text{ such that } \left( \delta_{h'_q, 0} + \delta_{h'_q, 1} \right) = 1, \tag{C.60}$$

while we have suppressed the prime notation in the last line of (C.59), as $h'_q = 2$ iff. $h_q = 2$. Indeed, $q = 1$ is possible iff. $h_1 = 1$ and then $\underline{\mathbf{h}}_{1,r,p,q}^{(h_1 \neq 0, h_r \neq 0, 1, h'_q + 1)} = \underline{\mathbf{h}}_{r,p,1}^{(h_r \neq 0, 1, 2)}$. Then the component 1 of $\underline{\mathbf{h}}_{r,p,1}^{(h_r \neq 0, 1, 2)}$ is $2 \neq k_1 = 0$, as $p \neq 1, r$, so we can argue the proof of (C.60) as done for the proof of (C.50). Instead, if $q \neq 1$, $h_1$ is not modified in $\underline{\mathbf{h}}_{1,r,p,q}^{(h_1 \neq 0, h_r \neq 0, 1, h'_q + 1)}$ so it stays $h_1 \neq k_1 = 0$, and once again we can argue the proof of (C.60) as done for the proof of (C.50). Collecting the results of the two interpolation expansions, we get the wanted formula (C.48) of the Lemma.

Let us now remark that from the fact that $\underline{\mathbf{h}}_{1,r}^{(h_1 \neq 0, h_r \neq 0)}$ satisfies (C.46) with $\underline{\mathbf{k}}$ of the form (C.44), then $\underline{\mathbf{h}}_{1,r,p,q}^{(h_1 \neq 0, h_r \neq 0, 1, 1)}$ satisfies (C.46) with the same $\underline{\mathbf{k}}$. Moreover, we have that $\underline{\mathbf{h}}_{1,r,p,q}^{(h_1 \neq 0, h_r \neq 0, 1, 1)}$ satisfies (C.46) with $\underline{\mathbf{k}}_1^{(0)}$, as it stays true that the component one of $\underline{\mathbf{h}}_{1,r,p,q}^{(h_1 \neq 0, h_r \neq 0, 1, 1)}$ is non-zero, independently from the value of $p \in \{2, ..., N\} \backslash \{r\}$ and of $q \in \{1, ..., N\} \backslash \{p\}$. Then, all the terms $\langle \underline{\mathbf{h}}_{1,r,p,q}^{(h_1 \neq 0, h_r \neq 0, 1, 1)} | T_2(\xi_q) | \underline{\mathbf{k}}_1^{(0)} \rangle$ on the r.h.s. of (C.48) can be expanded once again according to the same formula (C.48), as $\underline{\mathbf{h}}_{1,r,p,q}^{(h_1 \neq 0, h_r \neq 0, 1, 1)}$ behaves exactly like a $\underline{\mathbf{h}}_{1,r'}^{(h_1 \neq 0, 1)}$ with $r' = q$ and $h_{r'} = 1 \neq 0$. This ensure that this is a recursive formula. $\square$

The previous lemma implies the following:

**Corollary C.2.** *Under the same assumptions on $\underline{\mathbf{h}}_{1,r}^{(h_1 \neq 0, h_r \neq 0)}$ and $\underline{\mathbf{k}}$ as in the previous lemma, the following orthogonality condition holds*

$$\langle \underline{\mathbf{h}}_{1,r}^{(h_1 \neq 0, h_r \neq 0)} | T_2(\xi_r) | \underline{\mathbf{k}}_1^{(0)} \rangle = 0, \tag{C.61}$$

*for any fixed $r \in \{1, ..., N\}$.*

*Proof.* If $\underline{\mathbf{h}}_{1,r}^{(h_1 \neq 0, h_r \neq 0)}$ does not contain $h = 2$ or $h = 0$, this is proven by applying once the recursion formula. Otherwise, a first application of the recursion generates the $\underline{\mathbf{h}}_{1,r,p,q}^{(h_1 \neq 0, h_r \neq 0, 1, 1)}$ where we have reduced by one unit the number of $h = 2$ and the number of $h = 0$, while we have increased by two unit the number of $h = 1$, transforming $\underline{\mathbf{h}}$ like $h_p = 0 \to h'_p = 1$ and $h_q = 2 \to h'_q = 1$. Then, if $\underline{\mathbf{h}}_{1,r,p,q}^{(h_1 \neq 0, h_r \neq 0, 1, 1)}$ does not contain $h = 2$ or $h = 0$, the orthogonality

is proven just by applying once again the recursion formula. Otherwise, we can continue to apply it over and over until there are no $h = 2$ or $h = 0$ in the index of the SoV co-vectors involved. This proves the above Corollary. □

Let us now perform the induction step over the number $m$ of $k_a = 1$ in the vector $|\underline{\mathbf{k}}\rangle$. Let $\underline{\mathbf{h}}$ be the generic element of $\{0, 1, 2\}^N$ satisfying (C.46) with a fixed $\underline{\mathbf{k}}$ of the form (C.44). If $h_1 \neq 0$, then the orthogonality condition reads

$$0 = \langle \underline{\mathbf{h}} | \underline{\mathbf{k}} \rangle = \langle \underline{\mathbf{h}}_1^{(h_1 \neq 0)} | T_2(\xi_1) | \underline{\mathbf{k}}_1^{(0)} \rangle, \tag{C.62}$$

which follows by a direct application of the above corollary. If $h_1 = 0$, it holds

$$\langle \underline{\mathbf{h}}_1^{(0)} | \underline{\mathbf{k}} \rangle = \langle \underline{\mathbf{h}}_1^{(0)} | T_2(\xi_1) | \underline{\mathbf{k}}_1^{(0)} \rangle = c_1 \langle \underline{\mathbf{h}}_1^{(1)} | T_1(\xi_1^{(1)}) | \underline{\mathbf{k}}_1^{(0)} \rangle,$$

and so we use the following interpolation

$$\langle \underline{\mathbf{h}}_1^{(1)} | T_1(\xi_1^{(1)}) | \underline{\mathbf{k}}_1^{(0)} \rangle \underset{\text{UpC}}{=} t_1 \langle \underline{\mathbf{h}}_1^{(1)} | \underline{\mathbf{k}}_1^{(0)} \rangle + \langle \underline{\mathbf{h}}_1^{(1)} | \sum_{s=1}^N T_1(\xi_s^{(\delta_{h'_s,2})}) | \underline{\mathbf{k}}_1^{(0)} \rangle$$

$$\underset{\text{UpC}}{=} \sum_{s=1}^N (\delta_{h'_s,0} + \delta_{h'_s,1}) c_s^{\delta_{h'_s,0}} \langle \underline{\mathbf{h}}_{1,s}^{(1,h'_s+1)} | \underline{\mathbf{k}}_1^{(0)} \rangle + \sum_{s=2}^N \delta_{h'_s,2} \langle \underline{\mathbf{h}}_{1,s}^{(1,1)} | T_2(\xi_s) | \underline{\mathbf{k}}_1^{(0)} \rangle, \tag{C.63}$$

where we have defined $\{h'_1, ..., h'_N\} = \underline{\mathbf{h}}_1^{(1)}$. From the assumed orthogonality (i.e. the induction hypothesis) we get

$$\langle \underline{\mathbf{h}}_1^{(0)} | \underline{\mathbf{k}} \rangle = \langle \underline{\mathbf{h}}_1^{(0)} | T_2(\xi_1) | \underline{\mathbf{k}}_1^{(0)} \rangle \underset{\text{UpC}}{=} c_1 \sum_{s=1}^N \delta_{h_s,2} \langle \underline{\mathbf{h}}_{1,s}^{(1,1)} | T_2(\xi_s) | \underline{\mathbf{k}}_1^{(0)} \rangle, \tag{C.64}$$

being

$$\underline{\mathbf{h}}_{1,s}^{(1,h'_s+1)} \underset{(C.45)}{\neq} \underline{\mathbf{k}}_1^{(0)}, \quad \text{for any } s \text{ such that } \delta_{h'_s,0} + \delta_{h'_s,1} = 1. \tag{C.65}$$

Indeed, for $s = 1$ it holds $h'_1 = 1$ and so $h'_1 + 1 = 2 \neq k_1 = 0$, so we can argue the proof of (C.65) as done for the proof of (C.50). While for $s \neq 1$ it stays $h'_1 = 1$ so we have $h'_1 \neq k_1 = 0$, and once again the proof of (C.65) is done as that of (C.50).

It remains to observe that the terms at the r.h.s. of (C.64) satisfy the requirements of the previous corollary. This completes the proof by the induction of the pseudo-orthogonality (3.14).

Note that the proven orthogonality also implies that the above lemma and corollary indeed hold for any $m \leq N - 1$.

## C.2 Non-zero SoV co-vector/vector couplings

### C.2.1 Nondiagonal elements from diagonal ones

The orthogonality conditions implied in the formula (3.14) of the Theorem 3.1 have been proven in the previous subsection. Here, we complete the proof of this formula for the non-zero matrix elements $\langle \underline{\mathbf{h}} | \underline{\mathbf{k}} \rangle$ with their expressions in terms of the diagonal ones $\langle \underline{\mathbf{k}} | \underline{\mathbf{k}} \rangle$ and the power dependence w.r.t. $c = \det K$.

More precisely, let us assume that there are $m$ $k = 1$ in $|\underline{\mathbf{k}}\rangle$, let us say

$$k_{\pi_1} = k_{\pi_2} = \cdots = k_{\pi_m} = 1, \tag{C.66}$$

then we want to show that it holds

$$\langle \underline{\mathbf{h}} | \underline{\mathbf{k}}^{(1,...,1)}_{\pi_1,...,\pi_m} \rangle = c^{r+1} C^{\underline{\mathbf{k}}}_{\underline{\mathbf{h}}} \langle \underline{\mathbf{k}} | \underline{\mathbf{k}} \rangle, \tag{C.67}$$

with $C^{\underline{\mathbf{k}}}_{\underline{\mathbf{h}}}$ non-zero and independent w.r.t. c for[20] $\underline{\mathbf{h}}_{\pi_1,...,\pi_m} = \underline{\mathbf{k}}_{\pi_1,...,\pi_m} \in \{0,2\}^{N-m}$ and

$$(h_{\pi_{2a-1}}, h_{\pi_{2a}}) = (0,2), \quad \forall a \in \{1,...,r+1\} \quad \text{and} \quad h_{\pi_s} = 1, \quad \forall s \in \{2r+3,...,m\}. \tag{C.68}$$

Moreover, the next lemmas completely characterize the coefficients $C^{\underline{\mathbf{k}}}_{\underline{\mathbf{h}}}$ in terms of solutions to a derived recursion relations.

Up to a reordering in the indices of the $\xi_i$, the generic case of $r+1$ (0,2) couples in $\langle \underline{\mathbf{h}} |$, corresponding to $r+1$ (1,1) in $|\underline{\mathbf{k}} \rangle$, is equivalent to compute $\langle \underline{\mathbf{h}}^{(0,2,\underline{\mathbf{p}})}_{1,2,3,...,2r} | \underline{\mathbf{h}}^{(1,1,\underline{\mathbf{q}})}_{1,2,3,...,2r+2} \rangle$ in terms of $\langle \underline{\mathbf{h}}^{(1,1,\underline{\mathbf{q}})}_{1,2,3,...,2r} | \underline{\mathbf{h}}^{(1,1,\underline{\mathbf{q}})}_{1,2,3,...,2r+2} \rangle$, where:

$$\underline{\mathbf{p}} = \{p_1,...,p_{2r}\} \text{ with } p_{2a-i} = 2(1-i), \quad \forall a \in \{1,...,r\}, i \in \{0,1\}, \tag{C.69}$$

$$\underline{\mathbf{q}} = \{q_1,...,q_{2r}\} \text{ with } q_{2a-i} = 1, \quad \forall a \in \{1,...,r\}, i \in \{0,1\}, \tag{C.70}$$

while $\underline{\mathbf{h}}_{1,2,3,...,2r+2} \in \{0,1,2\}^{N-2(r+1)}$. Then the following lemma holds:

**Lemma C.3.** *Under the previous definition of the $\underline{\mathbf{p}}$ and $\underline{\mathbf{q}}$, the following expansion holds:*

$$C^{\underline{\mathbf{h}}^{(1,1,\underline{\mathbf{q}})}_{1,2,3,...,2r+2}}_{\underline{\mathbf{h}}^{(0,2,\underline{\mathbf{p}})}_{1,2,3,...,2r+2}} =$$

$$= \frac{c^{-r} q - \det M^{(I)}(\xi_1)}{\langle \underline{\mathbf{h}}^{(1,1,\underline{\mathbf{q}})}_{1,2,3,...,2r+2} | \underline{\mathbf{h}}^{(1,1,\underline{\mathbf{q}})}_{1,2,3,...,2r+2} \rangle} \left[ \prod_{a \neq 2, a=1}^{N} \frac{(\xi_1^{(1)} - \xi_a^{(\delta_{h'_a,2})})}{(\xi_2^{(1)} - \xi_a^{(\delta_{h'_a,2})})} \langle \underline{\mathbf{h}}^{(1,1,\underline{\mathbf{p}})}_{1,2,3,...,2r+2} | T_2(\xi_2) | \underline{\mathbf{h}}^{(0,1,\underline{\mathbf{q}})}_{1,2,3,...,2r+2} \rangle \right.$$

$$\left. + \sum_{j=1}^{r} \prod_{a \neq 2j+2, a=1}^{N} \frac{(\xi_1^{(1)} - \xi_a^{(\delta_{h'_a,2})})}{(\xi_{2j+2}^{(1)} - \xi_a^{(\delta_{h'_a,2})})} \langle \underline{\mathbf{h}}^{(1,2,\underline{\mathbf{p}}^{(1)}_{2j})}_{1,2,3,...,2r+2} | T_2(\xi_{2j+2}) | \underline{\mathbf{h}}^{(0,1,\underline{\mathbf{q}})}_{1,2,3,...,2r+2} \rangle \right], \tag{C.71}$$

*where we have denoted*

$$\underline{\mathbf{h}}' \equiv \{h'_1,...,h'_N\} = \underline{\mathbf{h}}^{(1,1,\underline{\mathbf{p}})}_{1,2,3,...,2r+2}, \tag{C.72}$$

*and for $r = 0$ we get:*

$$C^{\underline{\mathbf{h}}^{(1,1)}_{1,2}}_{\underline{\mathbf{h}}^{(0,2)}_{1,2}} = \frac{d(\xi_2 - \eta)}{d(\xi_1 - \eta)} \frac{q - \det M^{(I)}(\xi_1)}{\eta^{-2}(\xi_1 - \xi_2 + \eta)^2} \prod_{a \geq 3}^{N} \frac{(\xi_1^{(1)} - \xi_a^{(\delta_{h_a,2})})(\xi_2 - \xi_a^{(1-\delta_{h_a,0})})}{(\xi_2^{(1)} - \xi_a^{(\delta_{h_a,2})})(\xi_1 - \xi_a^{(1-\delta_{h_a,0})})}. \tag{C.73}$$

*Proof.* By the definition of the coefficients $C^{\underline{\mathbf{k}}}_{\underline{\mathbf{h}}}$ we have:

$$C^{\underline{\mathbf{h}}^{(1,1,\underline{\mathbf{q}})}_{1,2,3,...,2r+2}}_{\underline{\mathbf{h}}^{(0,2,\underline{\mathbf{p}})}_{1,2,3,...,2r+2}} = \frac{\langle \underline{\mathbf{h}}^{(0,2,\underline{\mathbf{p}})}_{1,2,3,...,2r} | \underline{\mathbf{h}}^{(1,1,\underline{\mathbf{q}})}_{1,2,3,...,2r+2} \rangle}{c^{r+1} \langle \underline{\mathbf{h}}^{(1,1,\underline{\mathbf{q}})}_{1,2,3,...,2r+2} | \underline{\mathbf{h}}^{(1,1,\underline{\mathbf{q}})}_{1,2,3,...,2r+2} \rangle}, \tag{C.74}$$

then formula (C.71) follows by the following identity

$$\langle \underline{\mathbf{h}}^{(0,2,\underline{\mathbf{p}})}_{1,2,3,...,2r} | \underline{\mathbf{h}}^{(1,1,\underline{\mathbf{q}})}_{1,2,3,...,2r+2} \rangle = c_1 \langle \underline{\mathbf{h}}^{(1,1,\underline{\mathbf{p}})}_{1,2,3,...,2r+2} | T_1(\xi_1^{(1)}) T_1(\xi_2) | \underline{\mathbf{h}}^{(0,1,\underline{\mathbf{q}})}_{1,2,3,...,2r+2} \rangle, \tag{C.75}$$

---

[20]Where we have introduced the notation $\underline{\mathbf{x}}_{r_1,...,r_m}$ without the upper index values to indicate the $N - m$-tuple obtained from the generic N-tuple $\underline{\mathbf{x}}$ removing the entries $\{r_1,...,r_m\} \subset \{1,...,N\}$.

once we make an interpolation expansion of $T_1(\xi_1^{(1)})$. More in detail, up to the coefficients, we use the interpolation identity

$$T_1(\xi_1^{(1)}) \underset{\text{UpC}}{=} t_1 + T_1(\xi_1) + \sum_{s \geq 2} T_1(\xi_s^{(\delta_{h_s,2})}), \qquad (C.76)$$

from which it follows

$$\langle \underline{\mathbf{h}}_{1,2,\ldots,2r+2}^{(1,1,\underline{\mathbf{p}})} | T_1(\xi_1^{(1)}) T_1(\xi_2) | \underline{\mathbf{h}}_{1,2,\ldots,2r+2}^{(0,1,\underline{\mathbf{q}})} \rangle =$$

$$\underset{\text{UpC}}{=} t_1 \langle \underline{\mathbf{h}}_{1,2,\ldots,2r+2}^{(1,2,\underline{\mathbf{p}})} | \underline{\mathbf{h}}_{1,2,\ldots,2r+2}^{(0,1,\underline{\mathbf{q}})} \rangle + \langle \underline{\mathbf{h}}_{1,2,\ldots,2r+2}^{(2,2,\underline{\mathbf{p}})} | \underline{\mathbf{h}}_{1,2,\ldots,2r+2}^{(0,1,\underline{\mathbf{q}})} \rangle + \langle \underline{\mathbf{h}}_{1,2,\ldots,2r+2}^{(1,1,\underline{\mathbf{p}})} | T_2(\xi_2) | \underline{\mathbf{h}}_{1,2,\ldots,2r+2}^{(0,1,\underline{\mathbf{q}})} \rangle$$

$$+ \sum_{a=1}^{r} \sum_{i=0}^{1} \langle \underline{\mathbf{h}}_{1,2,\ldots,2r+2}^{(1,2,\underline{\mathbf{p}})} | T_1(\xi_{2a+2-i}^{(1-i)}) | \underline{\mathbf{h}}_{1,2,\ldots,2r+2}^{(0,1,\underline{\mathbf{q}})} \rangle + \sum_{s=1+2(r+1)}^{N} \langle \underline{\mathbf{h}}_{1,2,\ldots,2r+2}^{(1,2,\underline{\mathbf{p}})} | T_1(\xi_s^{(\delta_{h_s,2})}) | \underline{\mathbf{h}}_{1,2,\ldots,2r+2}^{(0,1,\underline{\mathbf{q}})} \rangle$$

$$\qquad (C.77)$$

$$\underset{\text{UpC}}{=} \langle \underline{\mathbf{h}}_{1,2,\ldots,2r+2}^{(1,1,\underline{\mathbf{p}})} | T_2(\xi_2) | \underline{\mathbf{h}}_{1,2,\ldots,2r+2}^{(0,1,\underline{\mathbf{q}})} \rangle + \sum_{j=1}^{r} \langle \underline{\mathbf{h}}_{1,2,\ldots,2r+2}^{(1,2,\underline{\mathbf{p}}_{2j}^{(1)})} | T_2(\xi_{2j+2}) | \underline{\mathbf{h}}_{1,2,\ldots,2r+2}^{(0,1,\underline{\mathbf{q}})} \rangle. \qquad (C.78)$$

Indeed, from the previous orthogonality conditions, we have:

$$\langle \underline{\mathbf{h}}_{1,2,3,\ldots,2r+2}^{(1,2,\underline{\mathbf{p}})} | \underline{\mathbf{h}}_{1,2,3,\ldots,2r+2}^{(0,1,\underline{\mathbf{q}})} \rangle = 0, \quad \langle \underline{\mathbf{h}}_{1,2,3,\ldots,2r+2}^{(2,2,\underline{\mathbf{p}})} | \underline{\mathbf{h}}_{1,2,3,\ldots,2r+2}^{(0,1,\underline{\mathbf{q}})} \rangle = 0, \qquad (C.79)$$

and

$$\langle \underline{\mathbf{h}}_{1,2,3,\ldots,2r+2}^{(1,2,\underline{\mathbf{p}})} | T_1(\xi_{2a+1}) | \underline{\mathbf{h}}_{1,2,3,\ldots,2r+2}^{(0,1,\underline{\mathbf{q}})} \rangle = c_{2a+1} \langle \underline{\mathbf{h}}_{1,2,3,\ldots,2r+2}^{(1,2,\underline{\mathbf{p}}_{2a-1}^{(1)})} | \underline{\mathbf{h}}_{1,2,3,\ldots,2r+2}^{(0,1,\underline{\mathbf{q}})} \rangle = 0, \qquad (C.80)$$

for any $1 \leq a \leq r$. Also, for $s \geq 2r + 3$ and $h_s = 0, 1$, we have:

$$\langle \underline{\mathbf{h}}_{1,2,3,\ldots,2r+2}^{(1,2,\underline{\mathbf{p}})} | T_1(\xi_s) | \underline{\mathbf{h}}_{1,2,3,\ldots,2r+2}^{(0,1,\underline{\mathbf{q}})} \rangle = c_s^{\delta_{h_s,0}} \langle \underline{\mathbf{h}}_{1,2,3,\ldots,2r+2,s}^{(1,2,\underline{\mathbf{p}},h_s+1)} | \underline{\mathbf{h}}_{1,2,3,\ldots,2r+2,s}^{(0,1,\underline{\mathbf{q}},h_s)} \rangle = 0, \qquad (C.81)$$

as well as for $s \geq 2r + 3$ and $h_s = 2$ we have:

$$\langle \underline{\mathbf{h}}_{1,2,3,\ldots,2r+2}^{(1,2,\underline{\mathbf{p}})} | T_1(\xi_s^{(1)}) | \underline{\mathbf{h}}_{1,2,3,\ldots,2r+2}^{(0,1,\underline{\mathbf{q}})} \rangle = \langle \underline{\mathbf{h}}_{1,2,3,\ldots,2r+2,s}^{(1,2,\underline{\mathbf{p}},2)} | \underline{\mathbf{h}}_{1,2,3,\ldots,2r+2,s}^{(0,1,\underline{\mathbf{q}},1)} \rangle = 0. \qquad (C.82)$$

So we are left only with the terms written in (C.78) and our formula (C.71) follows once we reintroduce the missing interpolation coefficients of the formula (C.76).

Let us now compute explicitly the case with only one couple of $(0, 2)$, i.e. the case $r = 0$. Formula (C.71) reads:

$$C_{\underline{\mathbf{h}}_{1,2}^{(0,2)}}^{\underline{\mathbf{h}}_{1,2}^{(1,1)}} = q - \det M^{(I)}(\xi_1) \prod_{a \neq 2, a=1}^{N} \frac{(\xi_1^{(1)} - \xi_a^{(\delta_{h_a,2})})}{(\xi_2^{(1)} - \xi_a^{(\delta_{h_a,2})})} \frac{\langle \mathbf{h}_{1,2}^{(1,1)} | T_2(\xi_2) | \mathbf{h}_{1,2}^{(0,1)} \rangle}{\langle \underline{\mathbf{h}}_{1,2}^{(1,1)} | \underline{\mathbf{h}}_{1,2}^{(1,1)} \rangle}, \qquad (C.83)$$

then by using the following, up to the coefficients, interpolation identity:

$$T_2(\xi_2) \underset{\text{UpC}}{=} t_2 + T_2(\xi_1) + \sum_{s \geq 2} T_2(\xi_s^{(1-\delta_{h_s,0})}), \qquad (C.84)$$

we get:

$$\langle \underline{\mathbf{h}}_{1,2}^{(1,1)} | T_2(\xi_2) | \underline{\mathbf{h}}_{1,2}^{(0,1)} \rangle \underset{\text{UpC}}{=} \langle \underline{\mathbf{h}}_{1,2}^{(1,1)} | \underline{\mathbf{h}}_{1,2}^{(1,1)} \rangle, \qquad (C.85)$$

as by the orthogonality conditions, proven in the previous subsection, it holds:

$$\langle \underline{\mathbf{h}}_{1,2}^{(1,1)} | T_2(\xi_s^{(1-\delta_{h_s,0})}) | \underline{\mathbf{h}}_{1,2}^{(0,1)} \rangle = 0, \text{ for any } s \geq 2. \qquad (C.86)$$

Indeed, we have:

$$\langle \underline{\mathbf{h}}_{1,2}^{(1,1)}|T_2(\xi_s^{(1-\delta_{h_s,0})})|\underline{\mathbf{h}}_{1,2}^{(0,1)}\rangle = \begin{cases} \langle \underline{\mathbf{h}}_{1,2}^{(1,1)}|\underline{\mathbf{h}}_{1,2,s}^{(0,1,1)}\rangle = 0 \text{ if } h_s = 0, \\ c_s^{\delta_{h_s,2}}\langle \underline{\mathbf{h}}_{1,2,s}^{(1,1,h_s-1)}|\underline{\mathbf{h}}_{1,2}^{(0,1)}\rangle = 0 \text{ if } h_s = 1,2. \end{cases} \quad \text{(C.87)}$$

Then, reintroducing the missing interpolation coefficients in front to $T_2^{(K)}(\xi_1)$ in (C.84) we get our result (C.73). $\qquad\square$

Note that any term in the sum in (C.78), associated to a fixed $j \in \{1,...,r\}$, is formally identical to the first term of (C.78) up to the exchange of indices 2 and $2j+2$ in $\xi_h$.

The following lemma gives a recursive formula to compute the matrix elements on the right hand side of (C.71). To simplify the notations, the lemma is formulated explicitly for the first matrix element but it can be used similarly for the others matrix elements $\langle \underline{\mathbf{h}}_{1,2,3,...,2r+2}^{(1,2,\underline{\mathbf{p}}_{2j}^{(1)})}|T_2(\xi_{2j+2})|\underline{\mathbf{h}}_{1,2,3,...,2r+2}^{(0,1,\underline{\mathbf{q}})}\rangle$, by exchanging the indices $2 \leftrightarrow 2j+2$ in the $\xi_h$, for every term involving the $j$ index.

**Lemma C.4.** *Under the previous definition of the $\underline{\mathbf{p}}$ and $\underline{\mathbf{q}}$, then for $r \geq 1$ the following recursion formulae hold*

$$\langle \underline{\mathbf{h}}_{1,2,3,...,2r+2}^{(1,1,\underline{\mathbf{p}})}|T_2(\xi_2)|\underline{\mathbf{h}}_{1,2,3,...,2r+2}^{(0,1,\underline{\mathbf{q}})}\rangle = c_3 r_{1,2} \sum_{s=1}^{r} s_{1,2,3,2s}\langle \underline{\mathbf{h}}_{1,2,3,...,2r+2}^{(1,1,\underline{\mathbf{p}}_{1,2s}^{(1,1)})}|T_2(\xi_{2s+2})|\underline{\mathbf{h}}_{1,2,3,...,2r+2}^{(1,1,\underline{\mathbf{q}}_1^{(0)})}\rangle$$
$$+ \sum_{a=1}^{r} c_{2a+1} r_{2a+1,2} \left( \sum_{b=1}^{r} s_{1,2,2a+1,2b}\langle \underline{\mathbf{h}}_{1,2,3,...,2r+2}^{(1,1,\underline{\mathbf{p}}_{2a-1,2b}^{(1,1)})}|T_2(\xi_{2b+2})|\underline{\mathbf{h}}_{1,2,3,...,2r+2}^{(0,1,\underline{\mathbf{q}})}\rangle \right),$$

$$\text{(C.88)}$$

*where:*

$$r_{2a+1,2} = \frac{d(\xi_2^{(1)})}{d(\xi_{2a+1}^{(1)})} \prod_{n=0}^{r} \frac{\xi_2 - \xi_{2n+2}^{(1)}}{\xi_{2a+1} - \xi_{2n+2}^{(1)}} \prod_{\substack{n=0 \\ n \neq a}}^{r} \frac{\xi_2 - \xi_{2n+1}}{\xi_{2a+1} - \xi_{2n+1}} \prod_{2r+3 \leq j \leq N} \frac{\xi_2 - \xi_j^{(1-\delta_{h_j,0})}}{\xi_{2a+1} - \xi_j^{(1-\delta_{h_j,0})}}, \quad \text{(C.89)}$$

*and*

$$s_{1,2,2a+1,2b} = \prod_{i=1}^{2} \frac{\xi_{2a+1}^{(1)} - \xi_i}{\xi_{2b+2}^{(1)} - \xi_i} \prod_{\substack{n=1 \\ n \neq b}}^{r} \frac{\xi_{2a+1} - \xi_{2n+2}}{\xi_{2b+2} - \xi_{2n+2}} \prod_{n=1}^{r} \frac{\xi_{2a+1}^{(1)} - \xi_{2n+1}}{\xi_{2b+2}^{(1)} - \xi_{2n+1}} \prod_{2r+3 \leq j \leq N} \frac{\xi_{2a+1}^{(1)} - \xi_j^{(\delta_{h_j,2})}}{\xi_{2b+2}^{(1)} - \xi_j^{(\delta_{h_j,2})}}, \quad \text{(C.90)}$$

*with the following initial condition for $r = 0$:*

$$\langle \underline{\mathbf{h}}_{1,2}^{(1,1)}|T_2(\xi_2)|\underline{\mathbf{h}}_{1,2}^{(0,1)}\rangle = \langle \underline{\mathbf{h}}_{1,2}^{(1,1)}|\underline{\mathbf{h}}_{1,2}^{(1,1)}\rangle \frac{d(\xi_2-\eta)}{d(\xi_1-\eta)} \frac{\eta}{(\xi_1-\xi_2+\eta)^2} \prod_{a\geq 3}^{N} \frac{\xi_2 - \xi_a^{(1-\delta_{h_a,0})}}{\xi_1 - \xi_a^{(1-\delta_{h_a,0})}}. \quad \text{(C.91)}$$

*Proof.* Using the interpolation formula (C.84), we get

$$
\begin{aligned}
\langle \underline{\mathbf{h}}_{1,2,3,\ldots,2r+2}^{(1,1,\underline{\mathbf{p}})} | T_2(\xi_2) | \underline{\mathbf{h}}_{1,2,3,\ldots,2r+2}^{(0,1,\underline{\mathbf{q}})} \rangle \\
\underset{\text{UpC}}{=} \; t_2 \langle \underline{\mathbf{h}}_{1,2,3,\ldots,2r+2}^{(1,1,\underline{\mathbf{p}})} | \underline{\mathbf{h}}_{1,2,3,\ldots,2r+2}^{(0,1,\underline{\mathbf{q}})} \rangle \\
+ \langle \underline{\mathbf{h}}_{1,2,3,\ldots,2r+2}^{(1,1,\underline{\mathbf{p}})} | \underline{\mathbf{h}}_{1,2,3,\ldots,2r+2}^{(1,1,\underline{\mathbf{q}})} \rangle + \langle \underline{\mathbf{h}}_{1,2,3,\ldots,2r+2}^{(1,0,\underline{\mathbf{p}})} | \underline{\mathbf{h}}_{1,2,3,\ldots,2r+2}^{(0,1,\underline{\mathbf{q}})} \rangle \\
+ \sum_{a=1}^{r} \sum_{i=0}^{1} \langle \underline{\mathbf{h}}_{1,2,3,\ldots,2r+2}^{(1,1,\underline{\mathbf{p}})} | T_2(\xi_{2a+2-i}^{(1-i)}) | \underline{\mathbf{h}}_{1,2,3,\ldots,2r+2}^{(0,1,\underline{\mathbf{q}})} \rangle \\
+ \sum_{s=1+2(r+1)}^{N} \langle \underline{\mathbf{h}}_{1,2,3,\ldots,2r+2}^{(1,1,\underline{\mathbf{p}})} | T_2(\xi_s^{(\delta_{h_s,1}+\delta_{h_s,2})}) | \underline{\mathbf{h}}_{1,2,3,\ldots,2r+2}^{(0,1,\underline{\mathbf{q}})} \rangle \quad \text{(C.92)} \\
\underset{\text{UpC}}{=} \; \langle \underline{\mathbf{h}}_{1,2,3,\ldots,2r+2}^{(1,1,\underline{\mathbf{p}})} | \underline{\mathbf{h}}_{1,2,3,\ldots,2r+2}^{(1,1,\underline{\mathbf{q}})} \rangle + \sum_{a=1}^{r} c_{2a+1} \langle \underline{\mathbf{h}}_{1,2,3,\ldots,2r+2}^{(1,1,\underline{\mathbf{p}}_{2a-1}^{(1)})} | T_1(\xi_{2a+1}^{(1)}) | \underline{\mathbf{h}}_{1,2,3,\ldots,2r+2}^{(0,1,\underline{\mathbf{q}})} \rangle.
\end{aligned}
$$
$$\text{(C.93)}$$

Indeed,

$$
\langle \underline{\mathbf{h}}_{1,2,3,\ldots,2r+2}^{(1,1,\underline{\mathbf{p}})} | \underline{\mathbf{h}}_{1,2,3,\ldots,2r+2}^{(0,1,\underline{\mathbf{q}})} \rangle = 0, \quad \langle \underline{\mathbf{h}}_{1,2,3,\ldots,2r+2}^{(1,0,\underline{\mathbf{p}})} | \underline{\mathbf{h}}_{1,2,3,\ldots,2r+2}^{(0,1,\underline{\mathbf{q}})} \rangle = 0, \quad \text{(C.94)}
$$

and

$$
\langle \underline{\mathbf{h}}_{1,2,3,\ldots,2r+2}^{(1,1,\underline{\mathbf{p}})} | T_2(\xi_{2a+2}^{(1)}) | \underline{\mathbf{h}}_{1,2,3,\ldots,2r+2}^{(0,1,\underline{\mathbf{q}})} \rangle = c_{2a+2} \langle \underline{\mathbf{h}}_{1,2,3,\ldots,2r+2}^{(1,1,\underline{\mathbf{p}}_{2a}^{(1)})} | \underline{\mathbf{h}}_{1,2,3,\ldots,2r+2}^{(0,1,\underline{\mathbf{q}})} \rangle = 0, \quad \text{(C.95)}
$$

for any $1 \le a \le r$. Also, for $s \ge 2r+3$ and $h_s = 1, 2$, we have:

$$
\langle \underline{\mathbf{h}}_{1,2,3,\ldots,2r+2}^{(1,1,\underline{\mathbf{p}})} | T_2(\xi_s^{(1)}) | \underline{\mathbf{h}}_{1,2,3,\ldots,2r+2}^{(0,1,\underline{\mathbf{q}})} \rangle = c_s^{\delta_{h_s,2}} \langle \underline{\mathbf{h}}_{1,2,3,\ldots,2r+2,s}^{(1,1,\underline{\mathbf{p}},h_s-1)} | \underline{\mathbf{h}}_{1,2,3,\ldots,2r+2,s}^{(0,1,\underline{\mathbf{q}},h_s)} \rangle = 0, \quad \text{(C.96)}
$$

while for $s \ge 2r+3$ and $h_s = 0$, we have:

$$
\langle \underline{\mathbf{h}}_{1,2,3,\ldots,2r+2}^{(1,1,\underline{\mathbf{p}})} | T_2(\xi_s) | \underline{\mathbf{h}}_{1,2,3,\ldots,2r+2}^{(0,1,\underline{\mathbf{q}})} \rangle = \langle \underline{\mathbf{h}}_{1,2,3,\ldots,2r+2,s}^{(1,1,\underline{\mathbf{p}},0)} | \underline{\mathbf{h}}_{1,2,3,\ldots,2r+2,s}^{(0,1,\underline{\mathbf{q}},1)} \rangle = 0. \quad \text{(C.97)}
$$

So we are left only with the following terms for $a \in \{1,\ldots,r\}$ which read:

$$
\langle \underline{\mathbf{h}}_{1,2,3,\ldots,2r+2}^{(1,1,\underline{\mathbf{p}})} | T_2(\xi_{2a+1}) | \underline{\mathbf{h}}_{1,2,3,\ldots,2r+2}^{(0,1,\underline{\mathbf{q}})} \rangle = c_{2a+1} \langle \underline{\mathbf{h}}_{1,2,3,\ldots,2r+2}^{(1,1,\underline{\mathbf{p}}_{2a-1}^{(1)})} | T_1(\xi_{2a+1}^{(1)}) | \underline{\mathbf{h}}_{1,2,3,\ldots,2r+2}^{(0,1,\underline{\mathbf{q}})} \rangle. \quad \text{(C.98)}
$$

Now we can use the interpolation formula (C.76) and we get

$$
\begin{aligned}
\langle \underline{\mathbf{h}}_{1,2,3,\ldots,2r+2}^{(1,1,\underline{\mathbf{p}}_{2a-1}^{(1)})} | T_1(\xi_{2a+1}^{(1)}) | \underline{\mathbf{h}}_{1,2,3,\ldots,2r+2}^{(0,1,\underline{\mathbf{q}})} \rangle \underset{\text{UpC}}{=} t_1 \langle \underline{\mathbf{h}}_{1,2,3,\ldots,2r+2}^{(1,1,\underline{\mathbf{p}}_{2a-1}^{(1)})} | \underline{\mathbf{h}}_{1,2,3,\ldots,2r+2}^{(0,1,\underline{\mathbf{q}})} \rangle \\
+ \langle \underline{\mathbf{h}}_{1,2,3,\ldots,2r+2}^{(2,1,\underline{\mathbf{p}}_{2a-1}^{(1)})} | \underline{\mathbf{h}}_{1,2,3,\ldots,2r+2}^{(0,1,\underline{\mathbf{q}})} \rangle + \langle \underline{\mathbf{h}}_{1,2,3,\ldots,2r+2}^{(1,2,\underline{\mathbf{p}}_{2a-1}^{(1)})} | \underline{\mathbf{h}}_{1,2,3,\ldots,2r+2}^{(0,1,\underline{\mathbf{q}})} \rangle \\
+ \sum_{b=1}^{r} c_{2b+1}^{1-\delta_{b,a}} \langle \underline{\mathbf{h}}_{1,2,3,\ldots,2r+2}^{(1,1,\underline{\mathbf{p}}_{2a-1}^{(1)})} + \underline{\mathbf{e}}_{2b+1} | \underline{\mathbf{h}}_{1,2,3,\ldots,2r+2}^{(0,1,\underline{\mathbf{q}})} \rangle \\
+ \sum_{b=1}^{r} \langle \underline{\mathbf{h}}_{1,2,3,\ldots,2r+2}^{(1,1,\underline{\mathbf{p}}_{2a-1,2b}^{(1,0)})} | T_1(\xi_{2b+2}^{(1)}) | \underline{\mathbf{h}}_{1,2,3,\ldots,2r+2}^{(0,1,\underline{\mathbf{q}})} \rangle \\
+ \sum_{s=1+2(r+1)}^{N} \langle \underline{\mathbf{h}}_{1,2,3,\ldots,2r+2}^{(1,1,\underline{\mathbf{p}}_{2a-1}^{(1)})} | T_1(\xi_s^{(\delta_{h_s,2})}) | \underline{\mathbf{h}}_{1,2,3,\ldots,2r+2}^{(0,1,\underline{\mathbf{q}})} \rangle \quad \text{(C.99)} \\
\underset{\text{UpC}}{=} \sum_{b=1}^{r} \langle \underline{\mathbf{h}}_{1,2,3,\ldots,2r+2}^{(1,1,\underline{\mathbf{p}}_{2a-1,2b}^{(1,1)})} | T_2(\xi_{2b+2}) | \underline{\mathbf{h}}_{1,2,3,\ldots,2r+2}^{(0,1,\underline{\mathbf{q}})} \rangle. \quad \text{(C.100)}
\end{aligned}
$$

Indeed, by the orthogonality it holds:

$$\langle \underline{\mathbf{h}}^{(1,1,\underline{\mathbf{p}}^{(1)}_{2a-1})}_{1,2,3,\dots,2r+2} | \underline{\mathbf{h}}^{(0,1,\underline{\mathbf{q}})}_{1,2,3,\dots,2r+2} \rangle = 0, \quad \langle \underline{\mathbf{h}}^{(2,1,\underline{\mathbf{p}}^{(1)}_{2a-1})}_{1,2,3,\dots,2r+2} | \underline{\mathbf{h}}^{(0,1,\underline{\mathbf{q}})}_{1,2,3,\dots,2r+2} \rangle = 0 \tag{C.101}$$

$$\langle \underline{\mathbf{h}}^{(1,2,\underline{\mathbf{p}}^{(1)}_{2a-1})}_{1,2,3,\dots,2r+2} | \underline{\mathbf{h}}^{(0,1,\underline{\mathbf{q}})}_{1,2,3,\dots,2r+2} \rangle = 0, \quad \langle \underline{\mathbf{h}}^{(1,1,\underline{\mathbf{p}}^{(1)}_{2a-1})}_{1,2,3,\dots,2r+2} + \underline{\mathbf{e}}_{2b+1} | \underline{\mathbf{h}}^{(0,1,\underline{\mathbf{q}})}_{1,2,3,\dots,2r+2} \rangle = 0, \tag{C.102}$$

while for $h_s = 2$ it holds:

$$\langle \underline{\mathbf{h}}^{(1,1,\underline{\mathbf{p}}^{(1)}_{2a-1},2)}_{1,2,3,\dots,2r+2,s} | \mathsf{T}_1(\xi^{(1)}_s) | \underline{\mathbf{h}}^{(0,1,\underline{\mathbf{q}})}_{1,2,3,\dots,2r+2} \rangle = \langle \underline{\mathbf{h}}^{(1,1,\underline{\mathbf{p}}^{(1)}_{2a-1},2)}_{1,2,3,\dots,2r+2,s} | \underline{\mathbf{h}}^{(0,1,\underline{\mathbf{q}},1)}_{1,2,3,\dots,2r+2,s} \rangle = 0, \tag{C.103}$$

as well as for $h_s = 0, 1$ it holds:

$$\langle \underline{\mathbf{h}}^{(1,1,\underline{\mathbf{p}}^{(1)}_{2a-1})}_{1,2,3,\dots,2r+2} | \mathsf{T}_1(\xi_s) | \underline{\mathbf{h}}^{(0,1,\underline{\mathbf{q}})}_{1,2,3,\dots,2r+2} \rangle = \langle \underline{\mathbf{h}}^{(1,1,\underline{\mathbf{p}}^{(1)}_{2a-1},h_s+1)}_{1,2,3,\dots,2r+2,s} | \underline{\mathbf{h}}^{(0,1,\underline{\mathbf{q}},h_s)}_{1,2,3,\dots,2r+2,s} \rangle = 0. \tag{C.104}$$

Therefore we obtain the following mixed recursion formula

$$\langle \underline{\mathbf{h}}^{(1,1,\underline{\mathbf{p}})}_{1,2,3,\dots,2r+2} | \mathsf{T}_2(\xi_2) | \underline{\mathbf{h}}^{(0,1,\underline{\mathbf{q}})}_{1,2,3,\dots,2r+2} \rangle \underset{\mathrm{UpC}}{=} \langle \underline{\mathbf{h}}^{(1,1,\underline{\mathbf{p}})}_{1,2,3,\dots,2r+2} | \underline{\mathbf{h}}^{(1,1,\underline{\mathbf{q}})}_{1,2,3,\dots,2r+2} \rangle$$
$$+ \sum_{a=1}^{r} \sum_{b=1}^{r} c_{2a+1} \langle \underline{\mathbf{h}}^{(1,1,\underline{\mathbf{p}}^{(1,1)}_{2a-1,2b})}_{1,2,3,\dots,2r+2} | \mathsf{T}_2(\xi_{2b+2}) | \underline{\mathbf{h}}^{(0,1,\underline{\mathbf{q}})}_{1,2,3,\dots,2r+2} \rangle. \tag{C.105}$$

Indeed, all the matrix elements $\langle \underline{\mathbf{h}}^{(1,1,\underline{\mathbf{p}}^{(1,1)}_{2a-1,2b})}_{1,2,3,\dots,2r+2} | \mathsf{T}_2(\xi_{2b+2}) | \underline{\mathbf{h}}^{(0,1,\underline{\mathbf{q}})}_{1,2,3,\dots,2r+2} \rangle$ on the r.h.s. of (C.105) have $(r-1)$-couples of $(0,2)$, i.e. one less w.r.t. the first matrix element on the r.h.s. of (C.78) $\langle \underline{\mathbf{h}}^{(1,1,\underline{\mathbf{p}})}_{1,2,3,\dots,2r+2} | \mathsf{T}_2(\xi_2) | \underline{\mathbf{h}}^{(0,1,\underline{\mathbf{q}})}_{1,2,3,\dots,2r+2} \rangle$. Moreover, the matrix element $\langle \underline{\mathbf{h}}^{(1,1,\underline{\mathbf{p}})}_{1,2,3,\dots,2r+2} | \underline{\mathbf{h}}^{(1,1,\underline{\mathbf{q}})}_{1,2,3,\dots,2r+2} \rangle$ contains one couple less of $(0,2)$ that the starting matrix element $\langle \underline{\mathbf{h}}^{(0,2,\underline{\mathbf{p}})}_{1,2,3,\dots,2r+2} | \underline{\mathbf{h}}^{(1,1,\underline{\mathbf{q}})}_{1,2,3,\dots,2r+2} \rangle$, i.e. $r$-couples of $(0,2)$. Up to a reordering in the indices, $\langle \underline{\mathbf{h}}^{(1,1,\underline{\mathbf{q}})}_{1,2,3,\dots,2r+2} | \underline{\mathbf{h}}^{(1,1,\underline{\mathbf{q}})}_{1,2,3,\dots,2r+2} \rangle$ can be developed just as done in (C.71), generating matrix elements with $(r-1)$-couples of $(0,2)$. In total, we have that

$$\langle \underline{\mathbf{h}}^{(1,1,\underline{\mathbf{p}})}_{1,2,3,\dots,2r+2} | \underline{\mathbf{h}}^{(1,1,\underline{\mathbf{q}})}_{1,2,3,\dots,2r+2} \rangle \underset{\mathrm{UpC}}{=} c_3 \sum_{j=2}^{r+1} \langle \underline{\mathbf{h}}^{(1,1,\underline{\mathbf{p}}^{(1,1)}_{1,2j})}_{1,2,3,\dots,2r+2} | \mathsf{T}_2(\xi_{2j}) | \underline{\mathbf{h}}^{(1,1,\underline{\mathbf{q}}^{(0)}_1)}_{1,2,3,\dots,2r+2} \rangle, \tag{C.106}$$

and by substituting it in (C.105) we get the recursion formula (C.88), up to the coefficients.

Now that we have identified the non-zero contributions in the used interpolation formulae, we can easily compute the missing coefficients presented in (C.88). From (C.92), the non-zero contributions of $\mathsf{T}_2(\xi_2)$ read:

$$\sum_{a=0}^{r} \frac{d(\xi^{(1)}_2)}{d(\xi^{(1)}_{2a+1})} \prod_{b \neq 2a+1} \frac{\xi_2 - \xi^{(1-\delta_{\tilde{h}_b,0})}_b}{\xi_{2a+1} - \xi^{(1-\delta_{\tilde{h}_b,0})}_b} \mathsf{T}_2(\xi_{2a+1}), \tag{C.107}$$

where $\tilde{h}_0 = 1$ and $\tilde{h}_b$ is the $b$ element of $\underline{\mathbf{h}}^{(1,1,\underline{\mathbf{p}})}_{1,2,3,\dots,2r+2}$ for any $b \geq 2$. Similarly, from (C.99), the non-zero contributions of $\mathsf{T}_1(\xi^{(1)}_{2a+1})$ read:

$$\sum_{b=1}^{r} \prod_{c \neq 2b+2} \frac{\xi^{(1)}_{2a+1} - \xi^{(\delta_{h_c,2})}_c}{\xi^{(1)}_{2b+2} - \xi^{(\delta_{h_c,2})}_c} \mathsf{T}_1(\xi^{(1)}_{2b+2}), \tag{C.108}$$

where $h_b$ is the $b$ element of $\underline{\mathbf{h}}_{1,2,3,...,2r+2}^{(1,1,\underline{\mathbf{p}})}$ for any $b \geq 1$. Finally, from (C.106), the non-zero contributions of $\mathsf{T}_1(\xi_2^{(1)})$ read:

$$\sum_{b=1}^{r} \prod_{c \neq 2b+2} \frac{\xi_3^{(1)} - \xi_c^{(\delta_{h_c,2})}}{\xi_{2b+2}^{(1)} - \xi_c^{(\delta_{h_c,2})}} \mathsf{T}_1(\xi_{2b+2}^{(1)}), \tag{C.109}$$

where $h_b$ is the $b$ element of $\underline{\mathbf{h}}_{1,2,3,...,2r+2}^{(1,1,\underline{\mathbf{p}})}$ for any $b \geq 1$. From these expansions, it is simple to verify that the recursion holds as written in the lemma.

Finally, the initial condition (C.91) for the recursion just coincides with the identity (C.85), proven in the previous lemma, by reintroducing the missing interpolation coefficients in front to $T_2^{(K)}(\xi_1)$ in (C.84). $\qquad\square$

It is worth remarking that in the recursion formula (C.88) the common part $\underline{\mathbf{h}}_{1,2,3,...,2r+2}$ of the SoV co-vectors and vectors are left unchanged by the recursion, i.e. the recursion acts only on the $(0,2)$ couples.

Moreover, thanks to Lemma C.3 the solution of these recursions formulae lead to the determination of the coefficient $C_{\underline{\mathbf{h}}}^{\underline{\mathbf{k}}}$, as defined in (C.67). Here, we do not solve these recursions but we use the previous lemmas to complete the proof of the Theorem 3.1 by proving the independence of the $C_{\underline{\mathbf{h}}}^{\underline{\mathbf{k}}}$ w.r.t. c. We have just to remark that at the right hand side of (C.88) we have matrix elements of $\mathsf{T}_2(\xi_{2h})$ with $(r-1)$-couples of $(0,2)$ in the co-vector corresponding to $(r-1)$-couples of $(1,1)$ in the vector. The same statement holds true adapting (C.88) for the development of the others matrix elements $\langle \underline{\mathbf{h}}_{1,2,3,...,2r+2}^{(1,2,\underline{\mathbf{p}}^{(1)})} | \mathsf{T}_2(\xi_{2j+2}) | \underline{\mathbf{h}}_{1,2,3,...,2r+2}^{(0,1,\underline{\mathbf{q}})} \rangle$. Hence, applying $(r-1)$-times the same recursion formulae to all the non-zero matrix elements generated in this first step of the recursion, we end up exactly in the same diagonal matrix element $\langle \underline{\mathbf{h}}_{1,2,3,...,2r+2}^{(1,1,\underline{\mathbf{q}})} | \underline{\mathbf{h}}_{1,2,3,...,2r+2}^{(1,1,\underline{\mathbf{q}})} \rangle$, proving the following proportionality:

$$\langle \underline{\mathbf{h}}_{1,2,3,...,2r+2}^{(0,2,\underline{\mathbf{p}})} | \underline{\mathbf{h}}_{1,2,3,...,2r+2}^{(1,1,\underline{\mathbf{q}})} \rangle \propto \mathsf{c}^{r+1} \langle \underline{\mathbf{h}}_{1,2,3,...,2r+2}^{(1,1,\underline{\mathbf{q}})} | \underline{\mathbf{h}}_{1,2,3,...,2r+2}^{(1,1,\underline{\mathbf{q}})} \rangle, \tag{C.110}$$

as any time that we make a recursion we generate exactly a power one of c. The proportionality coefficient $C_{\underline{\mathbf{h}}}^{\underline{\mathbf{k}}}$ must then be independent with respect to c as the full dependence in c is already made explicit in the previous formula.

### C.2.2 Computation of diagonal elements

Here we give a proof of the form of the diagonal coupling between SoV co-vectors and vectors. It is independent from the proof of the same result, but in the special case $\det K = 0$, that is given in the main body of the paper, see Theorem 4.1.

We follow the standard procedure used to prove the "Sklyanin measure" [37,39], by using the usual interpolation formulae of the transfer matrices.

i) We have that
$$\langle \underline{\mathbf{h}}_a^{(1)} | T_2^{(K)}(\xi_a^{(1)}) | \underline{\mathbf{h}}_a^{(0)} \rangle = \langle \underline{\mathbf{h}}_a^{(0)} | \underline{\mathbf{h}}_a^{(0)} \rangle. \tag{C.111}$$

Computing the action of $T_2^{(K)}(\xi_a^{(1)})$ by interpolating in the right points

$$T_2^{(K)}(\xi_a^{(1)}) = d(\xi_a^{(2)}) \left( T_{2,\underline{\mathbf{z}}(\underline{\mathbf{h}})}^{(K,\infty)}(\xi_a^{(1)}) + \sum_{b=1}^{\mathsf{N}} g_{b,\underline{\mathbf{z}}(\underline{\mathbf{h}})}^{(2)}(\xi_a^{(1)}) T_2^{(K)}(\xi_b^{(\delta_{h_b,1}+\delta_{h_b,2})}) \right), \tag{C.112}$$

where we recall the definitions

$$\underline{z}(\underline{h}) = \{\delta_{h_1,1} + \delta_{h_1,2}, ..., \delta_{h_N,1} + \delta_{h_N,2}\}, \tag{C.113}$$

$$g_{a,\underline{h}}^{(m)}(\lambda) = \prod_{b \neq a, b=1}^{N} \frac{\lambda - \xi_b^{(h_b)}}{\xi_a^{(h_a)} - \xi_b^{(h_b)}} \prod_{b=1}^{(m-1)N} \frac{1}{\xi_a^{(h_a)} - \xi_b^{(-1)}}, \tag{C.114}$$

we get

$$\langle \underline{h}_a^{(0)} | \underline{h}_a^{(0)} \rangle = d(\xi_a^{(2)})(T_{2,\underline{z}(\underline{h})}^{(K,\infty)}(\xi_a^{(1)}) \langle \underline{h}_a^{(1)} | \underline{h}_a^{(0)} \rangle + g_{a,\underline{z}(\underline{h})}^{(2)}(\xi_a^{(1)}) \langle \underline{h}_a^{(1)} | \underline{h}_a^{(1)} \rangle \tag{C.115}$$

$$+ \sum_{b=1, b \neq a}^{N} g_{b,\underline{z}(\underline{h})}^{(2)}(\xi_a^{(1)}) \langle \underline{h}_a^{(1)} | T_2^{(K)}(\xi_b^{(\delta_{h_b,1} + \delta_{h_b,2})}) | \underline{h}_a^{(0)} \rangle). \tag{C.116}$$

Now, we can use the following identities:

$$\langle \underline{h}_a^{(1)} | T_2^{(K)}(\xi_b^{(\delta_{h_b,1} + \delta_{h_b,2})}) | \underline{h}_a^{(0)} \rangle = \begin{cases} \langle \underline{h}_{a,b}^{(1,h_b-1)} | \underline{h}_{a,b}^{(0,h_b)} \rangle & \text{if } h_b \in \{1,2\}, \\ \langle \underline{h}_{a,b}^{(1,0)} | \underline{h}_{a,b}^{(0,1)} \rangle & \text{if } h_b = 0, \end{cases} \tag{C.117}$$

and then being

$$\underline{h}_a^{(1)} \underset{(C.45)}{\neq} \underline{h}_a^{(0)}, \quad \underline{h}_{a,b}^{(1,0)} \underset{(C.45)}{\neq} \underline{h}_{a,b}^{(0,1)}, \tag{C.118}$$

$$\underline{h}_{a,b}^{(1,h_b-1)} \underset{(C.45)}{\neq} \underline{h}_{a,b}^{(0,h_b)} \quad \text{if } h_b \in \{1,2\}, \tag{C.119}$$

the orthogonality conditions implies the identity:

$$\langle \underline{h}_a^{(0)} | \underline{h}_a^{(0)} \rangle = d(\xi_a^{(2)}) g_{a,\underline{z}(\underline{h})}^{(2)}(\xi_a^{(1)}) \langle \underline{h}_a^{(1)} | \underline{h}_a^{(1)} \rangle, \tag{C.120}$$

or equivalently:

$$\frac{\langle \underline{h}_a^{(0)} | \underline{h}_a^{(0)} \rangle}{\langle \underline{h}_a^{(1)} | \underline{h}_a^{(1)} \rangle} = \frac{d(\xi_a^{(2)})}{d(\xi_a^{(1)})} \prod_{n \neq a, n=1}^{N} \frac{\xi_a^{(1)} - \xi_n^{(\delta_{h_n,1} + \delta_{h_n,2})}}{\xi_a - \xi_n^{(\delta_{h_n,1} + \delta_{h_n,2})}}. \tag{C.121}$$

ii) Similarly, we have

$$\langle \underline{h}_a^{(1)} | T_1^{(K)}(\xi_a) | \underline{h}_a^{(2)} \rangle = \langle \underline{h}_a^{(2)} | \underline{h}_a^{(2)} \rangle. \tag{C.122}$$

Computing the action of $T_1^{(K)}(\xi_a)$ by interpolating in the right points

$$T_1^{(K)}(\lambda) = T_{1,\underline{y}(\underline{h})}^{(K,\infty)}(\xi_a) + \sum_{a=1}^{N} g_{a,\underline{y}(\underline{h})}^{(1)}(\xi_a) T_1^{(K)}(\xi_a^{(\delta_{h_a,2})}), \tag{C.123}$$

where we recall the definitions

$$\underline{y}(\underline{h}) = \{\delta_{h_1,2}, ..., \delta_{h_N,2}\}, \tag{C.124}$$

we get

$$\langle \underline{h}_a^{(2)} | \underline{h}_a^{(2)} \rangle = T_{1,\underline{y}(\underline{h})}^{(K,\infty)}(\xi_a) \langle \underline{h}_a^{(1)} | \underline{h}_a^{(2)} \rangle + g_{a,\underline{y}(\underline{h})}^{(1)}(\xi_a) \langle \underline{h}_a^{(1)} | \underline{h}_a^{(1)} \rangle \tag{C.125}$$

$$+ \sum_{b=1, b \neq a}^{N} g_{b,\underline{y}(\underline{h})}^{(1)}(\xi_a) \langle \underline{h}_a^{(1)} | T_1^{(K)}(\xi_b^{(\delta_{h_b,2})}) | \underline{h}_a^{(2)} \rangle. \tag{C.126}$$

Now, using the following identities:

$$\langle \underline{\mathbf{h}}_a^{(1)}|T_1^{(K)}(\xi_b^{(\delta_{h_b,2})})|\underline{\mathbf{h}}_a^{(2)}\rangle = \begin{cases} \langle \underline{\mathbf{h}}_{a,b}^{(1,h_b+1)}|\underline{\mathbf{h}}_{a,b}^{(2,h_b)}\rangle & \text{if } h_b \in \{0,1\}, \\ \langle \underline{\mathbf{h}}_{a,b}^{(1,2)}|\underline{\mathbf{h}}_{a,b}^{(2,1)}\rangle & \text{if } h_b = 2, \end{cases} \tag{C.127}$$

and then being

$$\underline{\mathbf{h}}_a^{(1)} \underset{(C.45)}{\neq} \underline{\mathbf{h}}_a^{(2)}, \quad \underline{\mathbf{h}}_{a,b}^{(1,2)} \underset{(C.45)}{\neq} \underline{\mathbf{h}}_{a,b}^{(2,1)}, \tag{C.128}$$

$$\underline{\mathbf{h}}_{a,b}^{(1,h_b+1)} \underset{(C.45)}{\neq} \underline{\mathbf{h}}_{a,b}^{(2,h_b)} \quad \text{if } h_b \in \{1,2\}, \tag{C.129}$$

the orthogonality conditions implies the identity:

$$\langle \underline{\mathbf{h}}_a^{(2)}|\underline{\mathbf{h}}_a^{(2)}\rangle = g_{a,\mathbf{y}(\underline{\mathbf{h}})}^{(2)}(\xi_a)\langle \underline{\mathbf{h}}_a^{(1)}|\underline{\mathbf{h}}_a^{(1)}\rangle, \tag{C.130}$$

or equivalently:

$$\frac{\langle \underline{\mathbf{h}}_a^{(2)}|\underline{\mathbf{h}}_a^{(2)}\rangle}{\langle \underline{\mathbf{h}}_a^{(1)}|\underline{\mathbf{h}}_a^{(1)}\rangle} = \prod_{n\neq a,n=1}^{\mathsf{N}} \frac{\xi_a - \xi_n^{(\delta_{h_n,2})}}{\xi_a^{(1)} - \xi_n^{(\delta_{h_n,2})}}. \tag{C.131}$$

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
