# Peer review of "On Scalar Products in Higher Rank Quantum Separation of Variables"

_SciPost Physics, doi:SciPost Phys. 9, 086 (2020)_

## Round 2 · Referee Report · Anonymous (Referee 1) · 2020-10-5

Strengths

  1. Important results are obtained in the field of the SoV method.
  2. The paper is clearly written.

Weaknesses

No obvious weaknesses.

Report

This paper is devoted to the Separation of Variables (SoV) method. The authors use the approach developed in paper [1] and apply it to integrable models with a gl_3invariant R-matrix. The main goal of this work is to construct such separated bases of co-vectors and vectors that would be convenient for calculating scalar products. The authors first construct a system of pseudo-orthogonal co-vector/vector SoV bases. Then, using the freedom in generating family of conserved charges, the authors construct orthogonal systems. These results are used to calculate the scalar product of separated states.

I am sure that the results obtained are very important and deserve to be published. The article is rather technical, but it is very clearly written. The formulas and their transformations are provided with detailed comments.

I have two comments. The first one is optional. I missed at least a small final section, despite the fact that the main results are listed in detail in Introduction. I still think that summarizing the results and describing possible perspectives would be useful. Especially considering that the article is quite technical.

Second remark: a typo in Acknowledgements (L.V. is supported by...)

To summarize, I recommend publishing the paper without re-reviewing.

Requested changes

A typo in Acknowledgements. Other changes are optional.

---

## Round 2 · Referee Report · Anonymous (Referee 2) · 2020-10-7

Report

In this paper an improved version of Skyanin’s SoV procedure for constructing a basis of eigenstates of the Bethe algebra is worked out in the case of a gl(3). The eigenvectors in the improved approach are obtained by repeated action of twisted transfer matrices on a reference state. Explicit formulas are given for the scalar products of the separate bra and ket states. The reported results are correct, original and significant. The paper is carefully written but it is structured so that the reader should go through the whole text in order to understand and use the results. A short summary of the most important formulas would be helpful. This of course does not diminish the qualities of the paper. I recommend publication of the manuscript it its present form.

---

## Round 3 · Author Response

Dear Editor,
We would like to thank the referees for their comments and remarks. Following their suggestion, we have added a final section "conclusions and perspectives" to summarize the main results of the paper and to give perspectives on future developments. We also improved the links to the main formulae in our description of results in the end of introduction.
Best regards,
J. M. Maillet, G. Niccoli, L. Vignoli

---

## Round 3 · List of Changes

We have implemented the following changes in this version : - we have added links to the main formulae in our description of results in the end of introduction. - we have added a final section "conclusions and perspectives" to summarize the main results of the paper and to give perspectives on future developments. - a few typos fixed, in particular in the Acknowledgements. - some new links to published versions included in references when available.

---

## Editorial Decision

published